# FLOW-DISENTANGLED FEATURE IMPORTANCE

**Xingshu Chen** [†]
School of Mathematics
Sun Yat-sen University
Guangzhou, China

**Yifeng Guo** [†]
Department of Computational Biology
St. Jude Children's Research Hospital
TN, USA

**Jin-Hong Du** [‡]
Department of Statistics and Actuarial Science & Institute of Data Science
The University of Hong Kong
Hong Kong SAR, China

## ABSTRACT

Quantifying feature importance with valid statistical uncertainty is central to interpretable machine learning, yet classical model-agnostic methods often fail under feature correlation, producing unreliable attributions and compromising inference. Statistical approaches that address correlation through feature decorrelation have shown promise but remain restricted to $\ell_2$ loss, limiting their applicability across diverse machine learning tasks. We introduce Flow-Disentangled Feature Importance (FDFI), a model-agnostic framework that resolves these limitations by combining principled statistical inference with computational flexibility. FDFI leverages flow matching to learn flexible disentanglement maps that not only handle arbitrary feature distributions but also provide an interpretable pathway for understanding how importance is attributed through the data's correlation structure. The framework generalizes the decorrelation-based attribution to general differentiable loss functions, enabling statistically valid importance assessment for black-box predictors across regression and classification. We establish statistical inference theory, deriving semiparametric efficiency of FDFI estimators, which enables valid confidence intervals and hypothesis testing with Type I error control. Experiments demonstrate that FDFI achieves substantially higher statistical power than removal-based and conditional permutation approaches, while maintaining robust and interpretable attributions even under severe interdependence. These findings hold across synthetic benchmarks and a broad collection of real datasets spanning diverse domains.

## 1 INTRODUCTION

Quantifying the importance of input features is fundamental to model interpretability and scientific discovery (Murdoch et al., 2019). However, standard model-agnostic methods falter when features are correlated. Removal-based approaches, such as Leave-One-Covariate-Out (LOCO) (Lei et al., 2018) and resample-based approaches such as Conditional Permutation Importance (CPI) (Strobl et al., 2008), can produce ambiguous attributions because they cannot cleanly isolate the unique predictive contribution of a single variable from that of its statistical dependents. This confounding effect of multicollinearity undermines the reliability of explanations derived from complex models (Williamson et al., 2021; Verdinelli & Wasserman, 2024a). Further, many attribution methods provide only point estimates, lacking *uncertainty quantification* necessary for valid statistical inference, such as constructing confidence intervals or performing hypothesis testing (Chamma et al., 2023).

To address the challenge of correlation, Disentangled Feature Importance (DFI) was recently proposed as a principled framework for attribution under dependence (Du et al., 2025). The core idea is to first learn a transformation that maps the original correlated features into a latent space where they

---

[†]Equal contribution
[‡]Corresponding author (`jinhongd@hku.hk`)

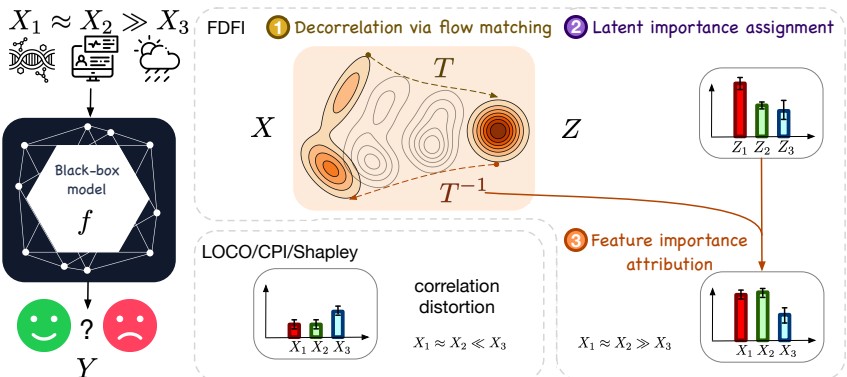

**Figure 1:** Overview of FDFI. A black-box model takes correlated features $X$ as input to predict $Y$. Conventional attribution methods (e.g., LOCO, CPI, Shapley) underestimate the importance of correlated features. DFI addresses this issue for Gaussian features under squared-error loss. The proposed FDFI framework generalizes DFI by replacing the linear optimal transport with flexible flow matching and extending to general losses and black-box models. Operationally, FDFI addresses this by (1) decorrelating $X$ into latent variables $Z$ via flow matching, (2) assigning importance scores in the disentangled latent space, and (3) attributing them back to the original features with uncertainty quantification. The latent importance reveals the intrinsic predictive variability, and the disentangled map enables interpretable attributions under correlations.

become statistically independent (Genizi, 1993). By measuring importance in this disentangled representation and then mapping the scores back to the original feature space, DFI effectively isolates the unique signal attributable to each feature. While powerful, the original DFI framework has two key limitations: (i) its reliance on an optimal transport (OT) map to perform the disentanglement can be computationally intensive and less flexible for complex, high-dimensional distributions beyond Gaussianity, and (ii) its formulation is restricted to importance scores evaluated with the $\ell_2$ loss.

In this work, we build upon this foundation to introduce Flow-Disentangled Feature Importance (FDFI), a significant generalization and enhancement of the DFI framework. An overview of the FDFI framework is provided in Figure 1. We replace the restrictive OT map with a more powerful and flexible transformation learned via flow matching, a state-of-the-art technique from generative modeling (Lipman et al., 2022). This allows our method to learn complex, nonlinear disentanglement maps between arbitrary feature distributions. Furthermore, we extend the DFI formulation to accommodate general differentiable loss functions, making it applicable to a broader range of tasks, including regression and classification. FDFI thus provides a unified framework that not only delivers robust *feature importance* under feature dependency but also enables valid statistical inference and *uncertainty quantification* for these importance scores. Moreover, the generative nature of the flow-based map provides a transparent mechanism for understanding how importance is attributed back through the data's *correlation structure*, providing reliable feature attribution.

### 1.1 SUMMARY OF CONTRIBUTIONS

**A general feature importance framework.** We analyze the relationship between three foundational feature importance measures under general differentiable loss functions (Theorem 2.1). We establish their formal equivalence under the $\ell_2$ loss (Lemma 2.2), a result that underscores their shared fundamental limitation: a vulnerability to correlation distortion when features are dependent. To address this, we propose a new framework, Flow-Disentangled Feature Importance (FDFI), which makes two key advances: (i) it replaces restrictive Gaussian transport map (Du et al., 2025; Genizi, 1993) with a more flexible and powerful transformation learned via flow matching (Lipman et al., 2022), enabling it to handle arbitrary feature distributions; and (ii) it extends the attribution framework beyond the $\ell_2$ loss to any differentiable loss function (4), broadening its applicability to classification and other machine learning tasks.

**Semiparametric statistical inference.** We establish valid statistical inference to quantify the uncertainty of estimated importance based on semiparametric efficiency theory. We derive the efficient influence functions and formally prove the asymptotic normality of our FDFI estimators for both the latent importance scores (Theorem 3.1) and the final original feature importance scores (Propo-

sition 3.2). This theoretical foundation provides a principled basis for valid statistical inference, enabling the construction of confidence intervals and hypothesis testing.

**Extensive empirical validation.** We conduct extensive experiments on both synthetic and real-world data, covering regression and classification tasks. Synthetic experiments systematically vary correlation strength, sample size, and data-generating processes (Section 4.1 and appendix E.1). Across nine real-world datasets spanning multiple domains and tasks (Section 4.3 and appendix E.2), FDFI produces robust and clinically interpretable importance profiles, consistently outperforming existing methods in identifying influential features under complex dependency structures.

## 1.2 RELATED WORK

**Model-agnostic feature attributions.** A central goal of explainable AI is to develop model-agnostic methods for quantifying feature importance. Permutation approaches (Breiman, 2001; Janitza et al., 2018) remove information by randomly shuffling a variable, but were originally introduced as algorithmic heuristics without statistical guarantees. One refinement is Conditional Permutation Importance (CPI) (Strobl et al., 2008; Hooker et al., 2021) that improves robustness through conditional resampling, and recent works (Chamma et al., 2023; Reyero-Lobo et al., 2026) further explore the statistical inference problem. In parallel, Leave-One-Covariate-Out (LOCO) defines importance as the change in predictive risk when a variable is removed, which naturally supports statistical inference (Lei et al., 2018; Rinaldo et al., 2019; Verdinelli & Wasserman, 2024b). In contrast, Shapley-value methods, originating from cooperative game theory (Shapley, 1953) and popularized in machine learning through SHAP (Lundberg & Lee, 2017), provide an axiomatic framework that ensures fairness and additivity; however, exact computation and statistical inference remain challenging. Subsequent work shows that SHAP values can be expressed as weighted averages of LOCO estimands under squared-error loss, thereby linking the two paradigms (Williamson & Feng, 2020). Despite these methodological advances, all approaches remain sensitive to collinearity, often underestimate the importance of correlated features (Verdinelli & Wasserman, 2024a).

**Feature importance under correlated predictors.** A growing body of work has sought to address the correlation distortion that undermines classical feature importance measures (Iooss & Prieur, 2019; Williamson & Feng, 2020; Williamson et al., 2021; Verdinelli & Wasserman, 2024b). Beyond conditional resampling strategies, a complementary line of work leverages knockoff constructions and conditional randomization tests to control the false discovery rate of variable selection (Candes et al., 2018; Gimenez et al., 2019; Mason & Fei, 2025); however, this differs from our objective of quantifying the marginal importance of features. A conceptually distinct direction is Disentangled Feature Importance (DFI) (Du et al., 2025), as a nonparametric extension of classic $R^2$-decomposition under linear models with correlated features (Genizi, 1993). It maps correlated predictors into an independent latent space via OT, computes importance in this disentangled representation, and attributes the results back to the original variables. However, DFI is less flexible for mapping between complex high-dimensional distributions and is also restricted to the square loss.

## 2 PRELIMINARIES

### 2.1 FOUNDATIONAL FEATURE IMPORTANCE MEASURES

Let $(X, Y) \in \mathbb{X} \times \mathbb{Y} \subseteq \mathbb{R}^d \times \mathbb{R}$ be a random vector representing features and a target variable. For a loss function $\ell : \mathbb{Y} \times \mathbb{Y} \to \mathbb{R}_+$, the risk of the model is defined as $R(f; X, Y) = \mathbb{E}[\ell(Y, f(X))]$. For any $j \in \{1, \ldots, d\}$, we denote by $X_{-j} = (X_1, \ldots, X_{j-1}, X_{j+1}, \ldots, X_d)$ the feature vector except $X_j$ and $X^{(j)} = (X_1, \ldots, X_{j-1}, \widetilde{X}_j, X_{j+1}, \ldots, X_d)$ the feature vector with $j$th feature replaced by $\widetilde{X}_j \sim p(X_j \mid X_{-j})$ independent of $Y$ and $X_j$ conditional on $X_{-j}$. For a given loss function $\ell$, we study the behavior of a prediction model $f : \mathbb{X} \to \mathbb{Y}$. In particular, we focus on the analysis of the Bayes optimal predictors whose prediction values are defined as

$$f(x) = \underset{y \in \mathbb{Y}}{\operatorname{argmin}} \, \mathbb{E}[\ell(Y, y) \mid X = x] \quad \text{and} \quad f_{-j}(x_{-j}) = \underset{y \in \mathbb{Y}}{\operatorname{argmin}} \, \mathbb{E}[\ell(Y, y) \mid X_{-j} = x_{-j}].$$

There are various types of importance measures. Below, we restrict our analysis to three basic ones.

- Leave-One-Covariate-Out Importance (LOCO) (Lei et al., 2018) that measures the increase in risk when the model is retrained without feature $j$:

$$\phi_{X_j}^{\text{LOCO}} = R(f_{-j}; X_{-j}, Y) - R(f; X, Y).$$

  In particular, the Shapley value with prediction error as the value function can be expressed as a weighted average of LOCO estimators over all subsets (Verdinelli & Wasserman, 2024b).
- Conditional Permutation Importance (CPI) (Hooker et al., 2021; Reyero-Lobo et al., 2026) that measures the increase in error when a feature is replaced by a random draw from its conditional distribution:

$$\phi_{X_j}^{\text{CPI}} = \frac{1}{2}[R(f; X^{(j)}, Y) - R(f; X, Y)].$$

- Sobol-Conditional Permutation Importance (SCPI) (Reyero-Lobo et al., 2026) that measures the portion of the model's prediction variance attributable to a feature, conditional on other features:

$$\phi_{X_j}^{\text{SCPI}} = R(g_j; X_{-j}, Y) - R(f; X, Y),$$

  where $g_j(X_{-j}) = \mathbb{E}[f(X^{(j)}) \mid X_{-j}]$ is the condition mean of prediction through $f$ given $X_{-j}$.

For a general function, $f_{-j}(X_{-j})$ and $g_j(X_{-j})$ may not be identical. Their difference gives rise to the distinction between LOCO and CPI. We first establish an exact identity that decomposes the difference $|\phi_{X_j}^{\text{LOCO}} - \phi_{X_j}^{\text{CPI}}|$ into interpretable components, which requires Assumption A1.

**Assumption A1** (Loss function). *Let the loss function $\ell : \mathbb{R} \times \mathbb{R} \to \mathbb{R}_+$ be a function of the true label $y$ and the prediction $\widehat{y}$. We assume the function $\ell(y, \cdot)$ is convex and differentiable with respect to its second argument for any fixed $y$. Furthermore, $\ell(y, \cdot)$ is $M$-smooth, i.e., $|\partial^2 \ell(y, \widehat{y})/\partial \widehat{y}^2| \leq M$.*

This assumption imposes standard regularity conditions for analyzing the risk functional. Convexity and differentiability ensure that the Bayes predictors are well-defined minimizers of the expected loss. The smoothness condition controls the second-order variability and holds for loss functions (e.g., the $\ell_2$ loss $\ell(y, \widehat{y}) = (y - \widehat{y})^2$, the binary cross-entropy loss defined on the logit scale); though it is not needed for the FDFI framework we introduce in the next section. Under this assumption, an identity involving LOCO and CPI is given in the following theorem.

**Theorem 2.1** (Bound on LOCO and CPI Discrepancy). *Under Assumption A1, the difference between LOCO and CPI can be decomposed into two components:*

$$\phi_{X_j}^{\text{LOCO}} - \phi_{X_j}^{\text{CPI}} = E_{\text{MIE}} - E_{\text{approx}}, \tag{1}$$

*where $E_{\text{MIE}} := \frac{1}{2}(\phi_{X_j}^{\text{SCPI}} - J_g)$ is the model interaction effect and $E_{\text{approx}} := R(g_j) - R(f_{-j})$ is the approximation error, and $J_g := \mathbb{E}[\ell(Y, f(X^{(j)}))] - R(g_j)$ is the Jensen gap. The absolute difference is bounded by:*

$$|\phi_{X_j}^{\text{LOCO}} - \phi_{X_j}^{\text{CPI}}| \leq E_{\text{approx}} + |E_{\text{MIE}}|,$$

*where individual components can be further upper-bounded by*

$$E_{\text{approx}} \leq \frac{M}{2}\|g_j - f_{-j}\|_{L_2}^2, \qquad |E_{\text{MIE}}| \leq \frac{M}{2}\mathbb{E}\left[\mathbb{V}(f(X) \mid X_{-j})\right].$$

The approximation error $E_{\text{approx}}$ quantifies the suboptimality from using the averaged model $g_j$ instead of the actual optimal model $f_{-j}$, while the Jensen gap $J_g$ quantifies the risk difference due to the convexity of the loss function, based on Jensen's inequality. The term $E_{\text{MIE}}$ precisely captures the discrepancy that arises from using a general loss function instead of the $\ell_2$ loss. For the $\ell_2$ loss, where $M = 2$, one has $J_g = \phi_{X_j}^{\text{SCPI}} = \mathbb{E}[\mathbb{V}(f(X) \mid X_{-j})]$. Substituting these into the definition yields $E_{\text{MIE}} \equiv 0$. Since $E_{\text{approx}}$ is also zero for the $\ell_2$ loss (as $g_j = f_{-j}$), the bound is tight. These results are summarized in Lemma 2.2.

**Lemma 2.2** (Equivalence of LOCO, CPI, and SCPI for $\ell_2$ loss). *For the $\ell_2$ loss $\ell(y, \widehat{y}) = (y - \widehat{y})^2$, the Bayes optimal predictor $f$ satisfies $g_j = f_{-j}$. Further, the identity (1) in Theorem 2.1 equals zero, yielding an exact equivalence: $\phi_{X_j}^{\text{LOCO}} = \phi_{X_j}^{\text{CPI}} = \phi_{X_j}^{\text{SCPI}} = \mathbb{E}[\mathbb{V}(f(X) \mid X_{-j})]$.*

Despite their equivalence under the $\ell_2$ loss, all three measures share a fundamental limitation: they suffer from *correlation distortion* when predictors are collinear. These methods cannot disentangle the predictive contribution of a feature from the shared signal of its statistical dependents, causing the importance scores to be diluted or masked and yielding ambiguous attributions (Verdinelli & Wasserman, 2024a;b). For example, as shown by Du et al. (2025, Example 5), given a linear model $Y = X_1 + X_2 + \epsilon$ where features $X_1$ and $X_2$ are near-perfectly correlated ($X_1 \approx X_2$), the above methods would assign near-zero importance to $X_1$ (since removing it causes minimal performance drop, as $X_2$ retains all predictive information) and, symmetrically, near-zero importance to $X_2$. This contradicts the fact that $X_1$ and $X_2$ are both critical.

## 2.2 DISENTANGLED FEATURE IMPORTANCE

To overcome this limitation, disentangled feature importance (DFI) (Du et al., 2025) was introduced to attribute importance scores while accounting for correlation under an $\ell_2$ loss. More specifically, DFI seeks a Gaussian optimal transport map $T : \mathbb{R}^d \to \mathbb{R}^d$ such that $Z = T(X)$ has *independent* coordinates, ideally matching a simple reference distribution such as multivariate Gaussian $\mathcal{N}_d(0, I)$. Without loss of generality, we assume $Z_j$ has zero mean and unit variance for all $j = 1, \ldots, d$. Once the disentangled representation is obtained, the latent DFI score $\phi_{Z_j}^{\mathrm{DFI}}$ for $Z_j$ is defined as:

$$\phi_{Z_j}^{\mathrm{DFI}} := \mathbb{E}\left[\mathbb{V}(f(X) \mid Z_{-j})\right] = \mathbb{E}\left[\mathbb{V}(\eta(Z) \mid Z_{-j})\right], \tag{2}$$

where $\mu(x) = \mathbb{E}[Y \mid X = x]$ and $\eta(z) := \mu(T^{-1}(z))$ denote the regression functions in the latent and feature spaces, respectively. This measure aligns with both LOCO and CPI, except that it is defined in terms of the latent feature $Z$ rather than the raw feature $X$. The DFI score $\phi_{X_l}^{\mathrm{DFI}}$ for each original feature $X_l$ by transferring importance from the disentangled features $Z$ back to $X$ through the sensitivity of $Z_j$ with respect to $X_l$. Formally, the original DFI measure for the $\ell_2$ loss is

$$\phi_{X_l}^{\mathrm{DFI}} := \sum_{j=1}^{d} \mathbb{E}\left[\mathbb{V}(f(X) \mid Z_{-j})\left(\frac{\partial X_l}{\partial Z_j}\right)^2\right], \tag{3}$$

where $\partial X_l/\partial Z_j$ denotes the partial derivative of $X_l = e_l^\top T^{-1}(Z)$ with respect to $Z_j$. The inner term $\mathbb{V}(\eta(Z) \mid Z_{-j})$ is the first-order Sobol index of $Z_j$, representing the "intrinsic" predictive signal uniquely attributable to that disentangled direction. Multiplying by $(\partial X_l/\partial Z_j)^2$ gauges how strongly fluctuations in $Z_j$ are expressed through $X_l$; integrating over the data distribution, averages these local sensitivities into a global importance score. Thus, $\phi_{X_l}^{\mathrm{DFI}}$ quantifies how much of the irreducible signal carried by all latent directions is channelled through $X_l$.

While DFI provides an alternative strategy for attribution under dependence, its formulation presents two key limitations that motivate our work: (i) By defining latent importance via conditional variance (2), the framework is intrinsically restricted to the $\ell_2$ loss, precluding its application to classification tasks or models using general differentiable loss functions; and (ii) The reliance of DFI on optimal transport maps to learn the transformation $T$ can be computationally intensive and less flexible for mapping the complex, high-dimensional distributions encountered in modern machine learning.

# 3 FLOW-BASED DISENTANGLED FEATURE IMPORTANCE

## 3.1 MODEL-AGNOSTIC DFI WITH GENERAL LOSS FUNCTIONS

To extend DFI beyond the $\ell_2$ loss, we generalize its two core components: the latent importance measure and the attribution rule. First, we require a latent importance measure $\phi_{Z_j}$ that is well-defined for a general loss function $\ell$. For each disentangled feature $Z_j$, we define the latent FDFI score as the expected increase in risk upon resampling its values conditional on other latent features:

$$\phi_{Z_j}^{\mathrm{FDFI}} := \mathbb{E}[\omega(O; T)], \text{ where } \omega(O; T) = \frac{1}{2}\left[\ell(Y, f(T^{-1}(Z^{(j)}))) - \ell(Y, f(T^{-1}(Z)))\right].$$

Here, $O = (X, Y)$ is the observation, $Z^{(j)}$ is the latent vector with its $j$-th coordinate $Z_j^{(j)}$ resampled from $p(Z_j)$, and $T$ is the nuisance transport map.

Second, we generalize the aggregation rule in (3). The original formulation weights the intrinsic signal of $Z_j$ quantified by the conditional variance $\mathbb{V}(f(X) \mid Z_{-j})$ by a sensitivity term. The squared Jacobian is based on a first-order approximation of the geometric influence of the latent variables on the original features. This sensitivity term is a property of the transport map $T$ itself, independent of the specific loss function. For a general loss, the analogous measure of intrinsic signal is the conditional expected increase in loss $\mathbb{E}[\omega(O; T) \mid Z_{-j}]$. By substituting this term for the conditional variance, we obtain a natural generalization:

$$\phi_{X_l}^{\text{FDFI}} := \sum_{j=1}^{d} \mathbb{E}\left[ \mathbb{E}\left[ \omega(O; T) \mid Z_{-j}\right] \left(\frac{\partial X_l}{\partial Z_j}\right)^2\right]. \tag{4}$$

This definition is a principled extension, and it recovers DFI (3) under $\ell_2$ loss; see Appendix A.3.

## 3.2 DISENTANGLED TRANSFORMATION WITH PROBABILISTIC FLOW

To overcome the limitation of (Gaussian) optimal transport, we utilize flow matching to learns a vector field that transports between a target distribution $X$ into a simple source distribution $Z$ (Lipman et al., 2022). Suppose that $\rho_0$ and $\rho_1$ are the densities for the source and target distributions, respectively. This amounts to constructing $U_t : \mathbb{R}^d \to \mathbb{R}^d$ such that if $u \sim \rho_0$, then $U_t(u) \sim \rho_t$ for some density satisfying $\rho_{t=0} = \rho_0$, $\rho_{t=1} = \rho_1$. In particular, a velocity field $v_t$ is used to construct the *flow* $U_t(u)$ of the ordinary differential equation:

$$\frac{\mathrm{d}}{\mathrm{d}t} U_t(u) = v_t(U_t(u)), \quad U_0(u) = u.$$

*Flow matching* (Lipman et al., 2022; Liu et al., 2022) was proposed to efficiently learn a regression-based vector field along a predefined probability path. The central idea is to define an interpolation between $U_0 \sim \rho_0$ and $U_1 \sim \rho_1$, typically in the form $U_t = (1-t)U_0 + tU_1$, $t \in [0, 1]$. The velocity field for flow matching Lipman et al. (2022; 2024) is defined as

$$v_t \in \arg\min_{w_t \in L_2(\rho_t)} \mathcal{L}(w_t \mid U_0, U_1), \text{ where } \mathcal{L}(w_t \mid U_0, U_1) := \int_0^1 \mathbb{E}\left[\left\|w_t(U_t) - U_1 + U_0\right\|^2\right] \mathrm{d}t.$$

The above optimization problem admits a unique minimizer and can be formulated as $v_t(u) = \mathbb{E}[U_1 - U_0 \mid U_t = u]$ (Lipman et al., 2022; Liu et al., 2022; Hertrich et al., 2025), which further guarantees a unique map $U_t$ by Theorem B.1. Hence, we can define FDFI (4) relative to this unique flow map $T := U_1$ (not necessarily an optimal transport map) that transforms the original feature $X$ to the latent feature $Z$, rather than discovering a single ground-truth importance. Let $\widehat{U}_t$ be the flow map obtained by solving the above ODE with $v_t$ replaced with $\widehat{v}_t$. Then, the estimated transport map can be represented by $\widehat{T} := \widehat{U}_1$. In particular, since the disentangled flow does not require labels, we can utilize independent, and potentially much larger, auxiliary unlabeled data to estimate $T$.

## 3.3 STATISTICAL ESTIMATION AND INFERENCE

After we obtain $\widehat{T}$ from auxiliary samples, we construct an estimator for $\phi_{Z_j}^{\text{FDFI}}$ using a set of $n$ i.i.d. samples $\{O_i\}_{i=1}^{n}$. For each data point $O_i = (X_i, Y_i)$, let $\widehat{Z}_i = \widehat{T}(X_i)$. The estimator is defined as:

$$\widehat{\phi}_{Z_j}^{\text{FDFI}} := \frac{1}{n} \sum_{i=1}^{n} \left[ \left( \frac{1}{2M} \sum_{k=1}^{M} \left[ \ell(Y_i, f(\widehat{T}^{-1}(\widehat{Z}_i^{(j,k)}))) - \ell(Y_i, f(\widehat{T}^{-1}(\widehat{Z}_i))) \right] \right) \right], \tag{5}$$

where $\{\widehat{Z}_i^{(j,k)}\}_{k=1}^{M}$ is generated by resampling the $j$-th coordinate of $\widehat{Z}_i$. We analyze this as a cross-fit estimator for the parameter $\phi_{Z_j}^{\text{FDFI}}$ defined in (2).

Before stating the main result, we briefly summarize the standing assumptions presented in Appendix C.1. Assumption A2 imposes basic identifiability and regularity conditions ensuring that the latent representation and associated flow are well-defined. Assumption A3 collects smoothness and integrability requirements on the velocity field $v_t$ that guarantee existence and stability of the flow $T$ and its inverse. Assumption A4 (i)–(iii) encode the loss differentiability and standard complexity/rate conditions on the nuisance estimators, which together ensure the pathwise differentiability of $\phi_{Z_j}^{\text{FDFI}}(\mathbb{P})$ and that the remainder of the cross-fit estimator is $o_{\mathbb{P}}(n^{-1/2})$.

**Theorem 3.1** (Asymptotic efficiency of latent FDFI). *Assume that Assumptions A2–A4 (i)-(iii) hold. If the nuisance estimator satisfies $\sqrt{\int_0^1 \|v_t - \widehat{v}_t\|_{L_2}^2 \mathrm{d}t} = o_{\mathbb{P}}(n^{-1/4})$, then the cross-fit estimator $\widehat{\phi}_{Z_j}^{\mathrm{FDFI}}$ given in Algorithm D.1 is asymptotically linear. It satisfies the expansion:*

$$\widehat{\phi}_{Z_j}^{\mathrm{FDFI}} - \phi_{Z_j}^{\mathrm{FDFI}}(\mathbb{P}) = (\mathbb{P}_n - \mathbb{P})\{\varphi_{Z_j}(O; \mathbb{P})\} + o_{\mathbb{P}}(n^{-1/2}),$$

*where the efficient influence function (EIF) $\varphi_{Z_j}(O; \mathbb{P})$ is given by:*

$$\varphi_{Z_j}(O; \mathbb{P}) := \omega(O; T) - \phi_{Z_j}^{\mathrm{FDFI}}(\mathbb{P}). \tag{6}$$

*Consequently, $\sqrt{n}(\widehat{\phi}_{Z_j}^{\mathrm{FDFI}} - \phi_{Z_j}^{\mathrm{FDFI}}) \xrightarrow{\mathrm{d}} \mathcal{N}(0, \mathbb{V}\{\varphi_{Z_j}(O; \mathbb{P})\})$ under the alternative $\mathcal{H}_{1j} : \phi_{Z_j}^{\mathrm{FDFI}}(\mathbb{P}) \neq 0$.*

Theorem 3.1 establishes semiparametric efficiency under the Neyman orthogonality condition that the estimand $\phi_{Z_j}^{\mathrm{FDFI}}$ is locally insensitive to first-order errors in the estimation of the nuisance transport map $T$. Consequently, the EIF (6) is identical to the EIF one would obtain if the actual map $T$ were known a priori. This permits us to use a flexible nonparametrically estimated $\widehat{T}$ (or equiavlently the velocity field $\widehat{v}_t$ through flow matching), which may converge at a rate slower than $n^{-1/2}$, and still achieve a $\sqrt{n}$-consistent, asymptotically normal, and efficient estimator for the latent importance $\phi_{Z_j}^{\mathrm{FDFI}}$. Without this property, the EIF would contain a complex correction term accounting for the influence of estimating $T$, as shown in Appendix C.2.

To construct Wald-based confidence intervals for $\phi_{Z_j}^{\mathrm{FDFI}}$, we estimate the asymptotic covariance $\mathbb{V}\{\varphi_{Z_j}(O; \mathbb{P})\}$ by plugging in consistent estimators of $T$ in (6) and evaluating the sample variance over independent observations. If $\phi_{Z_j}^{\mathrm{FDFI}} = 0$, the observation contribution to the asymptotic variance will be zero, which leads to additional complications that we discuss further in Appendix D.3.2.

A cross-fit estimator can also be used to estimate the importance $\phi_{X_l}^{\mathrm{FDFI}}$ defined in (4) for raw features:

$$\widehat{\phi}_{X_l}^{\mathrm{FDFI}} := \sum_{j=1}^d \frac{1}{n} \sum_{i=1}^n \left[ \frac{1}{2M} \sum_{k=1}^M \left[ \ell(Y_i, f(\widehat{T}^{-1}(\widehat{Z}_i^{(j,k)}))) - \ell(Y_i, f(\widehat{T}^{-1}(\widehat{Z}_i))) \right] \widehat{H}_{jl}(\widehat{Z}_i) \right], \tag{7}$$

where $\widehat{H}_{jl}(Z) = [\nabla \widehat{T}^{-1}(Z)]_{jl}^2$ is the square of estimated Jacobian of $X_l$ with respect to $Z_j$. The statistical properties of this estimator are analyzed in the following proposition. In addition to the conditions of Theorem 3.1, Assumption A4 (iv) is a mild strengthening of the complexity requirement that ensures uniform convergence of the plug-in estimator of the flow estimator $\widehat{T}$.

**Proposition 3.2** (Statistical inference of $\phi_{X_l}$). *Assume conditions in Theorem 3.1 and Assumption A4 (iv) hold. Then, the estimator $\widehat{\phi}_{X_l}^{\mathrm{FDFI}}$ is asymptotically normal under $\mathcal{H}_{1l} : \phi_{X_l}^{\mathrm{FDFI}}(\mathbb{P}) \neq 0$:*

$$\sqrt{n}(\widehat{\phi}_{X_l}^{\mathrm{FDFI}} - \phi_{X_l}^{\mathrm{FDFI}}(\mathbb{P})) \xrightarrow{\mathrm{d}} \mathcal{N}(0, \mathbb{V}\{\varphi_{X_l}(O; \mathbb{P})\}),$$

*where $\varphi_{X_l}(O; \mathbb{P}) = \sum_{j=1}^d \varphi_{jl}(O; \mathbb{P})$ is the EIF for $\phi_{X_l}^{\mathrm{FDFI}}(\mathbb{P})$ and $\varphi_{jl}$ is the EIF for the component $\phi_{jl} = \mathbb{E}[\omega_j(O; T) H_{jl}(X)]$, defined as $\varphi_{jl}(O; \mathbb{P}) := (\omega_j(O; T) H_{jl}(X) - \phi_{jl}) + \mathrm{Cov}(\omega_j(O; T), \mathrm{IF}_{H_{jl}}(O; \cdot))$, where $\mathrm{IF}_{H_{jl}}$ is the influence function of $\widehat{H}_{jl}$.*

Proposition 3.2 provides inferential tools for the final FDFI scores. The EIF $\varphi_{jl}$ for component attribution from $Z_j$ to $X_l$ induces a more complex structure than its latent counterpart in Theorem 3.1. It consists of two parts: (i) a first-order approximation term, $\omega_j(O; T) H_{jl}(X) - \phi_{jl}$, which represents the pointwise importance score centered by its mean, and (ii) a second-order correction term, $\mathrm{Cov}(\omega_j(O; T), \mathrm{IF}_{H_{jl}}(O; \cdot))$, which arises because the squared Jacobian term $\widehat{H}_{jl}$ is also estimated from data. The form of covariance reveals that the bias is driven by the covariance between point-wise importance scores and the point-wise influence on the map's estimated geometry. Since $\widehat{H}_{jl}$ is estimated on the auxiliary sample of size $m$, its influence function $\mathrm{IF}_{H_{jl}}$ is of order $\mathcal{O}_{\mathbb{P}}(m^{-1/2})$.

When $m$ is large, the correction term is negligible, and approximate EIF can be used for practical inference (see Appendix D.3.1). Algorithm 1 outlines a statistical inference procedure using FDFI.

---

**Algorithm 1 FDFI (brief).** See Algorithm D.1 for full pseudocode and implementation details.

---

**Require:** Labeled data $\mathcal{D}_{\text{est}} = \{(X_i, Y_i)\}_{i=1}^n$; black-box predictor $f$ and loss $\ell$; independent unlabeled covariates $\mathcal{D}_X = \{\widetilde{X}_i\}_{i=1}^m$; flow training routine $\mathcal{M}$; and other hyperparameters.

**Ensure:** Latent importance scores $\{\widehat{\phi}_{Z_j}\}_{j=1}^d$ and original feature scores $\{\widehat{\phi}_{X_l}\}_{l=1}^d$, with p-values.

1: **Step 1: Disentangled representation.** Train the flow on $\mathcal{D}_X$ to obtain a transport map $\widehat{T} = \mathcal{M}(\mathcal{D}_X)$ such that $Z = \widehat{T}(X)$ has approximately independent coordinates. For each labeled point $X_i$, set $\widehat{Z}_i := \widehat{T}(X_i)$, $J_i := \nabla \widehat{T}^{-1}(z)\big|_{z=\widehat{Z}_i}$, and $\widehat{H}_{jl}(\widehat{Z}_i) := (J_i)_{lj}^2$ for $j, l \in \{1, \ldots, d\}$.

2: **Step 2: Point-wise latent perturbations.** For each observation index $i \in \{1, \ldots, n\}$ and latent coordinate $j \in \{1, \ldots, d\}$, construct perturbed latent vectors $\{\widehat{Z}_i^{(j,k)}\}_{k=1}^M$ by resampling only the $j$-th coordinate of $\widehat{Z}_i$ from the reference $p_{Z_j}$, keeping all other coordinates fixed. Define the point-wise latent score $\Omega_{ij} := \frac{1}{2M} \sum_{k=1}^M \left\{ \ell\big(Y_i, f(\widehat{T}^{-1}(\widehat{Z}_i^{(j,k)}))\big) - \ell\big(Y_i, f(X_i)\big) \right\}$.

3: **Step 3: Attribution to original features.** For each observation $i$ and original feature index $l \in \{1, \ldots, d\}$, aggregate latent scores: $\Psi_{il} := \sum_{j=1}^d \Omega_{ij} \widehat{H}_{jl}(\widehat{Z}_i)$.

4: **Step 4: Latent importance and inference.** For each latent coordinate $j$, compute $\widehat{\phi}_{Z_j} := \frac{1}{n} \sum_{i=1}^n \Omega_{ij}$ in (5) and one-sided p-values based on estimated EIF components.

5: **Step 5: Original feature importance and inference.** For each original feature index $l$, compute $\widehat{\phi}_{X_l} := \frac{1}{n} \sum_{i=1}^n \Psi_{il}$ in (7) and one-sided p-values based on its approximate EIF.

6: **return** Importance scores and p-values.

---

## 4 EXPERIMENTS

### 4.1 THE IMPACT OF DISENTANGLEMENT

We evaluate the performance of LOCO, CPI, DFI, FDFI (SCPI), and FDFI (CPI) on identifying important raw features $X_l$'s across varying feature correlation strengths and sample sizes through simulations. DFI uses the Gaussian optimal transport map, while the two variants of FDFI estimators given in (14) and (15) use flow matching. The simulated data is generated from a nonlinear response model $y = \arctan(X_0 + X_1)\, \mathbb{1}_{\{X_2 > 0\}} + \sin(X_3 X_4)\, \mathbb{1}_{\{X_2 < 0\}} + \epsilon$ with $X \sim \mathcal{N}_{50}(0, \Sigma(\rho))$ and $\epsilon \sim \mathcal{N}(0, 1)$. The covariance matrix $\Sigma(\rho) := I_5 \otimes \Sigma_\rho \in \mathbb{R}^{50 \times 50}$ consists of 5 blocks of the equicorrelated matrix $\Sigma_\rho \in \mathbb{R}^{10 \times 10}$ with correlation coefficient $\rho$, i.e., $(\Sigma_\rho)_{ij} = 1$ if $i = j$ and $\rho$ otherwise. The $d = 50$ features are partitioned into three non-overlapping subsets $\{1, \ldots, d\} = \cup_{\ell=1}^3 \mathcal{C}_\ell$. Based on the response model, $\mathcal{C}_1 = \{0, \ldots, 4\}$ contains the *active features* that directly generate $y$. The set $\mathcal{C}_2 = \{5, \ldots, 9\}$ contains features from the first correlation block that are *correlated nulls*, i.e., dependent on $\mathcal{C}_1$ but with no direct predictive effect. The set $\mathcal{C}_3 = \{10, \ldots, 49\}$ contains the remaining *independent null features* from the other four blocks.

We evaluate all methods based on four key metrics: (1) AUC (Area Under the ROC Curve) on $\mathcal{C}_1 \cup \mathcal{C}_3$, (2) Power on $\mathcal{C}_1$, (3) Type I error on $\mathcal{C}_3$, and (4) Power on $\mathcal{C}_1 \cup \mathcal{C}_2$. AUC is computed by treating the estimated feature importance scores as prediction values and the ground-truth feature labels (informative = 1, null = 0) as binary outcomes. For statistical inference, we test $H_{0j}$ : feature $X_j$ is not important versus $H_{1j}$ : feature $X_j$ is important for each feature $j$. Based on a p-value $P_j$, statistical power is defined as the probability of rejecting the null when $H_{1j}$ is true, and type-I error is the probability of incorrectly rejecting $H_{0j}$ when it is true, i.e., declaring a null feature significant:

$$\text{Power} = \mathbb{P}(P_j \leq \alpha \mid H_{1j} \text{ is true}), \qquad \text{Type I Error} = \mathbb{P}(P_j \leq \alpha \mid H_{0j} \text{ is true}).$$

In all experiments, we present results using random forests (Breiman, 2001) as the regressor/classifier for clarity, while additional comparisons with alternative predictors (Lasso and neural networks) and with more complex dependency structures are provided in Appendices E.1.1–E.1.3.

As shown in Figure 2, all methods control the Type-I error at the nominal 5% level on the independent null features ($\mathcal{C}_3$) as expected. However, FDFI and DFI consistently outperform LOCO and CPI, attaining higher AUC and statistical power. Their performance is notably robust to increasing correlation, whereas LOCO and CPI's performance degrades significantly. Though both FDFI variants (SCPI and CPI) are theoretically equivalent under the $\ell_2$ loss (Lemma 2.2), the CPI variant demonstrates superior finite-sample power in low-sample or low-correlation regimes; we thus select

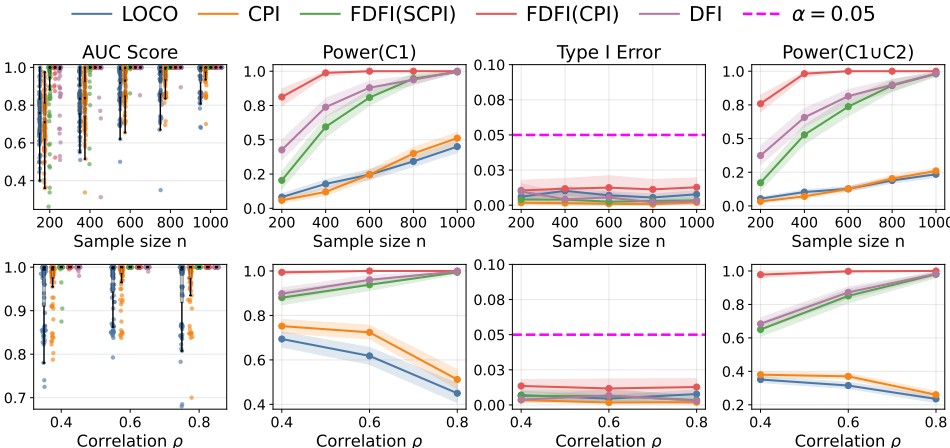

**Figure 2:** Simulation results of Section 4.1. Top: varying sample size with fixed $\rho = 0.8$. Bottom: varying correlation with fixed $n = 1000$. We report AUC, power, and type-I error for each method over 100 runs. Shaded regions denote 95% bootstrap confidence intervals, and the dashed line indicates the nominal type-I error level ($\alpha = 0.05$).

FDFI (CPI) as the representative method for subsequent experiments. These results demonstrate the effectiveness of disentanglement for assessing feature importance.

## 4.2 VALIDATION ON RNA-SEQUENCING DATASETS

To further validate FDFI's performance with high-dimensional and complex correlated features, we evaluate it on two RNA-seq datasets: (i) the TCGA-PANCAN-HiSeq bulk RNA-seq dataset ($n = 801, d = 20531$) for classifying five tumor types (BRCA, KIRC, COAD, LUAD, PRAD) (Weinstein et al., 2013); and (ii) a human single-cell RNA-seq dataset ($n = 632, d = 23257$) distinguishing neoplastic cells from tumor core versus periphery (Darmanis et al., 2017). In practice, a common ad-hoc approach for mitigating multicollinearity is to apply CPI or LOCO after hierarchical clustering. We select highly variable genes and compare the average prediction accuracy across datasets for important features selected by FDFI, DFI, and the ad-hoc approach, using two-fold splits and reporting the mean accuracy for each dataset; see Appendix E.2.5 for further details. As shown in Figure E10, FDFI consistently outperforms both DFI and the ad hoc method, highlighting its superiority in identifying more predictive and biologically representative gene sets; see Appendix E.2.5 for specific biomarkers identified by FDFI and their clinical relevance.

## 4.3 CASE STUDY ON CARDIOTOCOGRAPHY DATASET

In clinical applications, feature dependency is ubiquitous, and decisions are high-stakes, making reliable interpretability essential for clinical decision support. To evaluate FDFI's practical utility, we focus here on the Cardiotocography (CTG) dataset ($n = 2126, d = 21$) for a case study, which uses fetal heart rate (FHR) and uterine contraction (UC) features to classify fetal state (normal $y = 0$, non-normal $y = 1$) (Campos & Bernardes, 2000). We evaluate feature importance with the binary cross-entropy loss: $l(y, \widehat{y}) = -y \log(\widehat{y}) - (1-y) \log(1-\widehat{y})$. Additional large-scale real data studies, covering diverse domains and high-dimensional settings, are provided in Appendix E.2.

On the CTG dataset, nonlinear feature correlations cause LOCO and CPI to identify only a few essential features, whereas FDFI demonstrates substantially higher statistical power (Figure E7). Figure 3(a) visualizes the FDFI attribution pipeline: latent importance scores (left bar plot) are mapped via the squared Jacobian heatmap to produce the final FDFI scores. The heatmap $(\partial X_l / \partial Z_j)^2$ reveals the first-order influences among features, capturing a strong block-diagonal relationship among the FHR histogram features *LB*, *Mean*, *Mode*, and *Median*, which shows how the predictive importance from underlying latent features is distributed across correlated features.

As illustrated in Figure 3(b), the ad-hoc approach clusters features using Spearman correlation and selects a medoid to represent each cluster. However, choosing the top-$k$ features with FDFI or DFI achieves higher predictive accuracy than relying on cluster representatives from the ad-hoc method

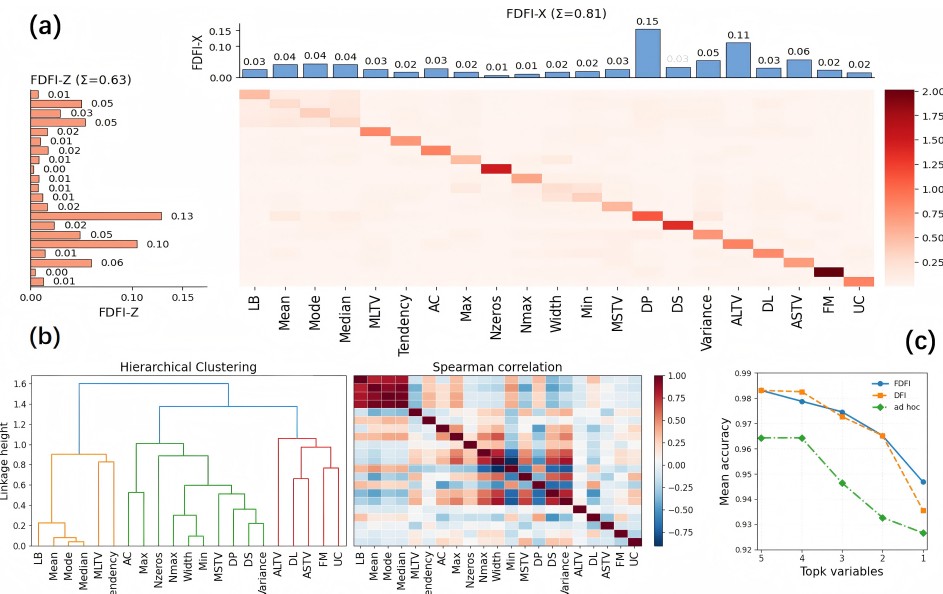

**Figure 3:** Data analysis of the CTG dataset. (a) FDFI estimation. Barplots show the estimated latent importance scores $\phi_{Z_j}^{\text{FDFI}}$ (left) and original importance scores $\phi_{X_l}^{\text{FDFI}}$ (top). The heatmap shows the squared Jacobian weight $(\partial X_l/\partial Z_j)^2$. (b) Hierarchical clustering results based on Spearman correlation. (c) Prediction accuracy with selected important features for FDFI, DFI, and an ad hoc method that applies CPI on the cluster-representative features.

(Figure 3(c)). Moreover, FDFI consistently outperforms DFI, underscoring the benefit of its flexible flow-based mapping over DFI's more restrictive Gaussian assumption.

## 5 CONCLUSION

We introduce a model-agnostic framework, FDFI, that uses flow matching to assist feature importance attribution under general differentiable loss functions. We establish the semiparametric efficiency of our FDFI estimators and provide valid statistical inference. Empirically, FDFI resolves the correlation distortion problem (Verdinelli & Wasserman, 2024b), successfully recovering established and correlated diagnostic features that classical methods overlook.

Beyond the empirical evaluations above, FDFI also enables several practical downstream workflows that benefit from statistically principled importance scores under dependence. First, by ranking features in a way that is robust to correlation, FDFI can be used for feature pruning and model compression, to preserve predictive performance while reducing dimensionality, inference cost, and model complexity (Han et al., 2015; Nelson et al., 2022; Ranek et al., 2024). Second, in data collection and experimental design, FDFI highlights variables or modalities with the highest marginal utility to prioritize operations under resource constraints (Wang et al., 2024). Third, by mitigating spurious attributions that arise from correlated covariates, FDFI improves model debugging and the detection of implausible signals, unlike traditional importance measures that can assign non-zero relevance to purely null variables (e.g., Chen et al., 2022). In summary, FDFI reliably guides what to measure, keep, or discard in high-stakes or resource-limited decision-making pipelines.

Lastly, FDFI is inherently a *local* sensitivity measure in a learned latent space and is most appropriate when the flow $\widehat{T}$ and predictor $f \circ \widehat{T}^{-1}$ are locally smooth and well-approximated by first-order expansions. In highly non-smooth or purely combinatorial regimes (e.g., parity/XOR-type rules with near piecewise-constant predictors), infinitesimal latent perturbations can fail to probe the truly influential directions. Developing extensions of flow-disentangled importance reliable under such non-smooth, combinatorial interaction structures is an interesting direction for future work.

**Acknoledgement.** The corresponding author gratefully acknowledges support from the Institute for Data Science (IDS) at the University of Hong Kong through the HKU100 award. Part of the computations was performed using research computing facilities offered by Information Technology Services, the University of Hong Kong.

**Ethics statement.** The authors acknowledge and adhere to the ICLR Code of Ethics. This research utilizes publicly available, anonymized biomedical datasets for empirical validation, which are standard in the machine learning literature, and their use in this context, to the best of our knowledge, presents no new risks to patient privacy. We do not foresee any direct negative societal impacts stemming from our methodological contribution. Furthermore, in line with ICLR policy, our use of Large Language Models as writing assistants has been disclosed in the Appendix.

**Reproducibility statement.** We have taken several steps to ensure the reproducibility of our work. The main text provides clear descriptions of the proposed method, experimental setup, and evaluation protocols. Complete proofs of the theoretical results and detailed explanations of assumptions are included in the Appendix. For experiments, we describe datasets and hyperparameter settings in the Appendix. The code for reproducing results can be found at https://github.com/ParadiseforAndaChen/FLOW-DISENTANGLED-FEATURE-IMPORTANCE/ and the FDFI package can be found at https://github.com/jaydu1/FDFI.

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

# Appendix

This serves as an appendix to the main paper. Below, we provide an outline for the appendix along with a summary of the notation used in the main paper and the appendix.

**Organization.** The content of the appendix is organized as follows.

| | Appendix | Content |
|---|---|---|
| Appendix A | Appendix A.1 | Proof of Theorem 2.1. |
| | Appendix A.2 | Proof of Lemma 2.2. |
| | Appendix A.3 | Equivalence of FDFI and DFI under $\ell_2$ loss. |
| Appendix B | Appendix B.1 | Uniqueness of flow solution (Theorem B.1). |
| | Appendix B.2 | Estimation error of transport map (Lemma B.2). |
| Appendix C | Appendix C.1 | Assumptions. |
| | Appendix C.2 | Proof of Theorem C.1 and Theorem 3.1. |
| | Appendix C.3 | Proof of Proposition 3.2. |
| Appendix D | Appendix D.1 | The full FDFI algorithm and computational devices. |
| | Appendix D.2 | Technical details of flow matching for FDFI. |
| | Appendix D.3 | Technical details of statistical inference for FDFI. |
| | Appendix D.4 | Computational time comparison. |
| Appendix E | Appendix E.1 | Extra simulation studies. |
| | Appendix E.2 | Extra real data studies. |

**Notation.** An overview of some general notation used in the main paper and the appendix is as follows.

For a vector $X \in \mathbb{R}^d$, $X_{\mathcal{S}} \in \mathbb{R}^{|\mathcal{S}|}$ denotes a sub-vector of $X$ indexed by $\mathcal{S} \subseteq [d]$, and $X_{-j} := X_{\{1,\dots,j-1,j+1,\dots,d\}}$. In $\mathbb{R}^d$, the $j$-th standard basis vector is denoted by $e_j$ and the zero vector is denoted by $0_d$ or simply $0$ if the dimension is clear from the context. The cardinality of a set $\mathcal{S}$ is denoted by $|\mathcal{S}|$. The indicator function is denoted by $\mathbb{1}_{\{\cdot\}}$.

For a tuple of random vectors $O = (X, Y)$, the expectation and probability over the joint distribution $\mathbb{P}$ are denoted by $\mathbb{E}(\cdot)$ and $\mathbb{P}(\cdot)$, respectively. For (potentially random) measurable functions $f$, we denote expectations with respect to the data-generating distribution of $O$ alone by $\mathbb{P}f(O) = \int f d\mathbb{P}$, while $\mathbb{E}[f(O)]$ marginalizes out all randomness from both $O$ and any nuisance functions $f$ is dependent on. The empirical expectation over $n$ samples is denoted by $\mathbb{P}_n f(O) = \frac{1}{n}\sum_{i=1}^{n} f(O_i)$. The $L_2$ norm of a function $f$ is denoted by $\|f\|_{L_2}$.

For a statistical estimand $\phi$, we write $\phi(\mathbb{P})$ to emphasise its dependence on the underlying distribution $\mathbb{P}$. The population and empirical variances (covariances) are denoted by $\mathbb{V}$ and $\mathbb{V}_n$ ($\mathbb{C}$ and $\mathbb{C}_n$). The $d$-dimensional multivariate normal distribution with mean $\mu$ and covariance $\Sigma$ is denoted by $\mathcal{N}_d(\mu, \Sigma)$. For matrices $A_1, A_2, \dots, A_k$, the notation $\mathrm{diag}(A_1, A_2, \dots, A_k)$ represents a block diagonal matrix that combines all of the matrices.

We use "$o$" and "$\mathcal{O}$" to denote the little-o and big-O notations; "$o_{\mathbb{P}}$" and "$\mathcal{O}_{\mathbb{P}}$" are their probabilistic counterparts. For sequences $\{a_n\}$ and $\{b_n\}$, we write $a_n \lesssim b_n$ if $a_n = \mathcal{O}(b_n)$; and $a_n \asymp b_n$ if $a_n = \mathcal{O}(b_n)$ and $b_n = \mathcal{O}(a_n)$. Convergence in distribution is denoted by "$\xrightarrow{d}$".

**The Use of Large Language Models.** In preparing this manuscript, Large Language Models (LLMs) were used strictly as auxiliary tools for: (i) Language editing and polishing: improving clarity, grammar, and academic style of author-written text without changing technical content. (ii) Literature search assistance: surfacing potentially relevant references and recent work; all cited materials were independently verified and read by the authors. No parts of the scientific contribution (problem formulation, methodology, experiments, results, or analysis) were generated by LLMs. The authors take full responsibility for all contents of the paper, including any text refined with LLM support. LLMs are not eligible for authorship.

# A    COMPARISON BETWEEN LOCO- AND CPI-BASED DFI

## A.1    PROOF OF THEOREM 2.1

*Proof of Theorem 2.1.* The proof proceeds in three parts. First, we prove the exact identity. Second, we apply the triangle inequality. Third, we derive the bounds for the error terms $E_{\text{approx}}$ and $J_g$.

For simplicity, we write $R(f; X, Y)$ and $R(f_{-j}; X_{-j}, Y)$ as $R(f)$ and $R(f_{-j})$, respectively.

**Part 1: Proof of the Identity.** We start with the right-hand side of the identity and substitute the definitions of $\phi_{X_j}^{\text{SCPI}}$, $E_{\text{approx}}$, and $J_g$.

$$
\begin{aligned}
&\frac{1}{2}\phi_{X_j}^{\text{SCPI}} - \frac{1}{2}J_g - E_{\text{approx}} \\
&= \frac{1}{2}(R(g_j) - R(f)) - \frac{1}{2}(\mathbb{E}[\ell(Y, f(X^{(j)}))] - R(g_j)) - (R(g_j) - R(f_{-j})) \\
&= \left(\frac{1}{2}R(g_j) - R(g_j) + \frac{1}{2}R(g_j)\right) + R(f_{-j}) - \frac{1}{2}R(f) - \frac{1}{2}\mathbb{E}[\ell(Y, f(X^{(j)}))] \\
&= R(f_{-j}) - \frac{1}{2}R(f) - \frac{1}{2}\mathbb{E}[\ell(Y, f(X^{(j)}))] \\
&= (R(f_{-j}) - R(f)) - \frac{1}{2}(R(f; X^{(j)}, Y) - R(f; X)) \\
&= \phi_{X_j}^{\text{LOCO}} - \phi_{X_j}^{\text{CPI}}.
\end{aligned}
$$

This confirms the identity $\phi_{X_j}^{\text{LOCO}} - \phi_{X_j}^{\text{CPI}} = \frac{1}{2}(\phi_{X_j}^{\text{SCPI}} - J_g) - E_{\text{approx}}$.

**Part 2: Application of the Triangle Inequality.** Taking the absolute value of the identity and applying the triangle inequality yields:

$$
\begin{aligned}
|\phi_{X_j}^{\text{LOCO}} - \phi_{X_j}^{\text{CPI}}| &= |(\phi_{X_j}^{\text{SCPI}} - J_g)/2 - E_{\text{approx}}| \\
&\leq \frac{1}{2}|\phi_{X_j}^{\text{SCPI}} - J_g| + |-E_{\text{approx}}| \\
&= |E_{\text{MIE}}| + E_{\text{approx}}.
\end{aligned}
$$

The last step follows from the fact that $E_{\text{approx}} \geq 0$, because the function $f_{-j}$ is the Bayes optimal predictor, meaning it is the minimizer of the risk functional.

**Part 3: Bounding the Error Terms.** We now derive the bounds for $E_{\text{approx}}$ and $E_{\text{MIE}}$.

*(i) Bounding $E_{\text{approx}} = R(g_j) - R(f_{-j})$:* The risk $R(h) = \mathbb{E}[\ell(Y, h(X_{-j}))]$ is a convex functional of the predictor $h$. For a convex and $M$-smooth functional, the difference in value between any point and the minimizer is bounded. Because $R(h)$ is $M$-smooth with respect to the $L_2$ norm on functions, we have:

$$
R(g_j) - R(f_{-j}) \leq \frac{M}{2}\mathbb{E}[|g_j(X_{-j}) - f_{-j}(X_{-j})|^2] = \frac{M}{2}\|g_j - f_{-j}\|_{L_2}^2.
$$

This provides the bound for the approximation error.

*(ii) Bounding $J_g = \mathbb{E}[\ell(Y, f(X^{(j)}))] - R(g_j)$:* We can write $J_g$ using iterated expectation:

$$
J_g = \mathbb{E}_{Y, X_{-j}}\left[\mathbb{E}[\ell(Y, f(X^{(j)})) \mid X_{-j}] - \ell(Y, \mathbb{E}[f(X^{(j)}) \mid X_{-j}])\right] \tag{8}
$$

The inner term is the Jensen gap for the convex function $\ell(y, \cdot)$ and the random variable $f(X^{(j)})$ (where the randomness comes from $\widetilde{X}_j \sim P(\cdot \mid X_{-j})$). For an $M$-smooth convex function $\phi$, the Jensen gap is bounded by $\mathbb{E}[\phi(Z)] - \phi(\mathbb{E}[Z]) \leq \frac{M}{2}\mathbb{V}(Z)$. Applying this to the inner expectation (conditional on $Y, X_{-j}$):

$$
\mathbb{E}[\ell(Y, f(X^{(j)})) \mid X_{-j}] - \ell(Y, \mathbb{E}[f(X^{(j)}) \mid X_{-j}]) \leq \frac{M}{2}\mathbb{V}(f(X^{(j)} \mid X_{-j})).
$$

Taking the expectation of both sides with respect to $Y, X_{-j}$ gives the final bound:

$$
J_g \leq \mathbb{E}\left[\frac{M}{2}\mathbb{V}(f(X) \mid X_{-j})\right] = \frac{M}{2}\mathbb{E}[\mathbb{V}(f(X) \mid X_{-j})].
$$

*(iii) Bounding* $|E_{\mathrm{MIE}}|$*:* By the triangle inequality on the definition of $E_{\mathrm{MIE}}$:

$$|E_{\mathrm{MIE}}| = \frac{1}{2}|\phi_{X_j}^{\mathrm{SCPI}} - J_g| \leq \frac{1}{2}(|\phi_{X_j}^{\mathrm{SCPI}}| + J_g).$$

From step (ii), the bound for $J_g$ is: $J_g \leq \frac{M}{2}\mathbb{E}\left[\mathbb{V}(f(X) \mid X_{-j})\right]$. To bound $|\phi_{X_j}^{\mathrm{SCPI}}|$, we use a second-order Taylor expansion of $\ell(Y, g_j)$ around $f(X)$. By Assumption A1, for some $\xi_i$ on the line segment between $g_j(X_{-j})$ and $f(X)$, we have:

$$\ell(Y, g_j(X_{-j})) = \ell(Y, f(X)) + \ell'(Y, f(X))(g_j(X_{-j}) - f(X)) + \frac{1}{2}\ell''(\xi_i)(g_j(X_{-j}) - f(X))^2.$$

Taking the expectation and rearranging gives

$$\begin{aligned}
\phi_{X_j}^{\mathrm{SCPI}} &= R(g_j) - R(f) \\
&= \mathbb{E}[\ell(Y, g_j) - \ell(Y, f)] \\
&= \mathbb{E}[\ell'(Y, f(X))(g_j(X_{-j}) - f(X))] + \frac{1}{2}\mathbb{E}[\ell''(\xi_i)(g_j(X_{-j}) - f(X))^2].
\end{aligned}$$

Since $f$ is the Bayes optimal predictor, $\mathbb{E}[\ell'(Y, f(X)) \mid X] = 0$. The first term vanishes by iterated expectation: $\mathbb{E}[\mathbb{E}[\ell'(Y, f(X)) \mid X](g_j(X_{-j}) - f(X))] = 0$. This leaves:

$$|2\phi_{X_j}^{\mathrm{SCPI}}| = \left|\frac{1}{2}\mathbb{E}[\ell''(\xi_i)(g_j - f)^2]\right| \leq \frac{1}{2}\mathbb{E}[|\ell''(\xi_i)|(g_j - f)^2] \leq \frac{M}{2}\mathbb{E}[(g_j - f)^2].$$

Recognizing that $\mathbb{E}[(g_j(X_{-j}) - f(X))^2] = \mathbb{E}[\mathbb{V}(f(X) \mid X_{-j})]$, we get $|\phi_{X_j}^{\mathrm{SCPI}}| \leq \frac{M}{2}\mathbb{E}[\mathbb{V}(f(X) \mid X_{-j})]$.

Combining the bounds for $|\phi_{X_j}^{\mathrm{SCPI}}|$ and $J_g$:

$$\begin{aligned}
|E_{\mathrm{MIE}}| \leq \frac{1}{2}(|\phi_{X_j}^{\mathrm{SCPI}}| + J_g) &\leq \frac{M}{4}\mathbb{E}[\mathbb{V}(f(X) \mid X_{-j})] + \frac{M}{4}\mathbb{E}[\mathbb{V}(f(X) \mid X_{-j})] \\
&= \frac{M}{2}\mathbb{E}\left[\mathbb{V}(f(X) \mid X_{-j})\right].
\end{aligned}$$

Substituting the final bounds for $E_{\mathrm{approx}}$ and $|E_{\mathrm{MIE}}|$ into the result from Part 1 completes the proof.
$\qquad\square$

## A.2 PROOF OF LEMMA 2.2

*Proof of Lemma 2.2.* For the $\ell_2$ loss, we analyze each term of the identity $\phi_{X_j}^{\mathrm{LOCO}} - \phi_{X_j}^{\mathrm{CPI}} = (\phi_{X_j}^{\mathrm{SCPI}} - J_g)/2 - E_{\mathrm{approx}}$. Since $f_{-j}(X_{-j}) = \mathbb{E}[Y \mid X_{-j}]$ and $g_j(X_{-j}) = \mathbb{E}[\mathbb{E}[Y \mid X] \mid X_{-j}] = \mathbb{E}[Y \mid X_{-j}]$, we have $f_{-j} = g_j$. This implies the approximation error $E_{\mathrm{approx}} = R(g_j) - R(f_{-j}) = 0$ and by definition,

$$\phi_{X_j}^{\mathrm{LOCO}} = \phi_{X_j}^{\mathrm{SCPI}}. \tag{9}$$

For (8) under the $\ell_2$ loss, from the derivation in Du et al. (2025a, Lemma 2.2), we also have that

$$\phi_{X_j}^{\mathrm{SCPI}} = J_g = \mathbb{E}[\mathbb{V}(f(X) \mid X_{-j})]. \tag{10}$$

Substituting these into the identity gives $\phi_{X_j}^{\mathrm{LOCO}} - \phi_{X_j}^{\mathrm{CPI}} = \frac{1}{2}J_g - \frac{1}{2}J_g - 0 = 0$, i.e.,

$$\phi_{X_j}^{\mathrm{LOCO}} = \phi_{X_j}^{\mathrm{CPI}}. \tag{11}$$

Combining (9) and (11) completes the proof.
$\qquad\square$

## A.3 EQUIVALENCE OF FDFI AND DFI UNDER $\ell_2$ LOSS

Recall that the DFI and FDFI estimands for the original features are defined in (3) and (4), respectively.

For $\ell_2$ loss, the influence function component in (4) becomes

$$\omega(O; T) = \frac{1}{2}\left[\ell(Y, f(T^{-1}(Z^{(j)}))) - \ell(Y, f(T^{-1}(Z)))\right]$$
$$= \frac{1}{2}[(Y - f(T^{-1}(Z^{(j)})))^2 - (Y - f(T^{-1}(Z)))^2].$$

Taking expectation conditional on $Z_{-j}$, we further have

$$\mathbb{E}[\omega(O; T) \mid Z_{-j}] = \frac{1}{2}\mathbb{E}[(Y - f(T^{-1}(Z^{(j)})))^2 - (Y - f(T^{-1}(Z)))^2 \mid Z_{-j}]$$
$$= \frac{1}{2}\mathbb{E}[f(T^{-1}(Z^{(j)})))^2 - f(T^{-1}(Z))^2]$$
$$\quad - \mathbb{E}[Y(f(T^{-1}(Z^{(j)})) - f(T^{-1}(Z)), \mid Z_{-j}]$$
$$= \frac{1}{2}\mathbb{E}[[f(T^{-1}(Z^{(j)})) - f(T^{-1}(Z))]^2 \mid Z_{-j}]$$
$$\quad - \mathbb{E}[[f(T^{-1}(Z)) - Y][f(T^{-1}(Z^{(j)})) - f(T^{-1}(Z))], \mid Z_{-j}]$$
$$= \frac{1}{2}\mathbb{V}(f(T^{-1}(Z)) \mid Z_{-j}) - 0$$
$$= \frac{1}{2}\mathbb{V}(f(T^{-1}(Z)) \mid Z_{-j}),$$

which reduces to the component in (3). Therefore, the DFI and FDFI estimands coincide under $\ell_2$ loss.

# B  PROPERTIES OF PROBABILISTIC FLOW

## B.1  UNIQUENESS OF SOLUTIONS

**Assumption A2.** *Let $(U_0, U_1, U_t) \in \mathbb{R}^{3d}$ admit a joint density $p_t(u_0, u_1, u)$ for $t \in [0,1]$. Define*

$$\rho_t(u) := \int_{\mathbb{R}^{2d}} p_t(u_0, u_1, u)\, \mathrm{d}u_0 \mathrm{d}u_1,$$

$$m_t(u) := \int_{\mathbb{R}^{2d}} (u_1 - u_0) p_t(u_0, u_1, u)\, \mathrm{d}u_0 \mathrm{d}u_1,$$

*and*

$$v_t(u) := \frac{m_t(u)}{\rho_t(u)}.$$

*Assume that*

(i) *Continuity: $m_t$, $\rho_t$ are continuous on $[0,1] \times \mathbb{R}^d$, are $C^1$ in $u$, and $\nabla_u m_t$, $\nabla_u \rho_t$ are continuous on $[0,1] \times \mathbb{R}^d$;*

(ii) *Uniform lower bound and bounded score on compact supports: For every compact set $K \subset \mathbb{R}^d$, there exist $\beta_K > 0$ and $M_K < \infty$ such that*

$$\inf_{(t,u) \in [0,1] \times K} \rho_t(u) \geq \beta_K, \qquad \sup_{(t,u) \in [0,1] \times K} \|\nabla_u \log \rho_t(u)\| \leq M_K;$$

(iii) *Bounded $\nabla_u m_t$ on compact supports: For every compact set $K$, there exists $J_K < \infty$ with*

$$\sup_{(t,u) \in [0,1] \times K} \|\nabla_u m_t(u)\| \leq J_K;$$

(iv) *Conditional second moment with linear growth (uniform in $t$): There exist bounded continuous $c_0, c_1 : [0,1] \to [0, \infty)$ such that*

$$\mathbb{E}\big[\|U_1 - U_0\|^2 \mid U_t = u\big] \leq c_0(t) + c_1(t)\|u\|^2 \quad \text{for all } (t, u).$$

*$A := \sup_t \sqrt{c_0(t)} < \infty$ and $B := \sup_t \sqrt{c_1(t)} < \infty$.*

The conditions in Assumption A2 are standard in the theory of ordinary differential equations, Assumption A2(i)-(iii) ensure that the velocity field $v_t(u)$ is continuous and uniformly Lipschitz in $u$. This is precisely the regularity required by the Picard–Lindelöf theorem, which guarantees local existence and uniqueness of the ODE solution (Hartman, 2002; Amann, 2011). Assumption A2(iv) further imposes a linear growth bound, ruling out finite-time blow-up and thus ensuring that the solution extends globally to $[0,1]$ (Hartman, 2002; Amann, 2011). In the Gaussian mixture setting, Assumption A2(i)-(iii) hold since the density and score are smooth and bounded on compacts (Bishop & Nasrabadi, 2006; Hyvärinen, 2005). Moreover, the conditional expectation $\mathbb{E}[U_1 - U_0 \mid U_t = u]$ is a convex combination of affine functions in $u$, so that $v_t(u)$ remains smooth and $m_t(u) = \rho_t(u)v_t(u)$ has gradient $\nabla_u m_t(u)$ continuous and bounded on compacts (Anderson, 2003; Bishop & Nasrabadi, 2006). Assumption A2(iv) is verified in the final part of the proof.

**Theorem B.1** (Local and global uniqueness of $U_t$). *Under Assumption A2 (i)-(iii), the velocity field $v_t(u)$ is continuous on $[0,1] \times \mathbb{R}^d$ and, for every compact $K \subset \mathbb{R}^d$, is uniformly (in $t$) Lipschitz in $u$. Hence, for every initial $u \in \mathbb{R}^d$, there exists a $\tau > 0$ such that the ODE*

$$\frac{\mathrm{d}}{\mathrm{d}t} U_t(u) = v_t(U_t(u)), \quad U_0(u) = u,$$

*admits a locally unique solution on $[0, \tau)$ .*

*If, in addition, Assumption A2 (iv) holds, then the solution extends to the whole interval $[0,1]$, and is globally unique. Moreover, this condition is automatically satisfied when the source distribution $U_0$ is Gaussian and the target distribution $U_1$ follows a Gaussian mixture distribution.*

In the standard theory of ordinary differential equations (ODEs), local uniqueness of the solution follows immediately if one directly assumes that the velocity field $v_t(u)$ is continuous in $t$ and

locally Lipschitz in $u$. This is precisely the setting of the classical Picard–Lindelöf theorem. Under these conditions, the ODE admits a unique local solution. A further linear growth condition ensures global uniqueness.

In practice, when $v_t$ is parameterized by a neural network, Lipschitz continuity can be enforced by architectural choices: for instance, using $1$-Lipschitz activations such as ReLU or $\tanh$ together with spectral norm constraints on each weight matrix. In this case the Lipschitz constant of the network is upper bounded by the product of the spectral norms of the layers (Virmaux & Scaman, 2018; Bartlett et al., 2017; Gouk et al., 2021). Compared to directly assuming Lipschitz continuity of $v_t$, our Assumption A2 adopts a distributional perspective: it imposes regularity and boundedness on $\rho_t$ and $m_t$, which in turn imply that $v_t$ is Lipschitz continuous. The advantage of this formulation is that it does not rely on a particular parameterization of $v_t$ and can be verified for broad distribution families such as Gaussian mixtures.

*Proof of Theorem B.1.* The proof proceeds in six parts. The first five parts establish the proof of Theorem B.1, and the last part shows that the Gaussian and Gaussian mixture case automatically satisfies Assumption A2 (iv).

**Part 1: Continuity of $v_t$.** By the quotient rule (componentwise),

$$\nabla_u v_t(u) = \frac{\nabla_u m_t(u)}{\rho_t(u)} - v_t(u) \otimes \nabla_u \log \rho_t(u)$$

Assumption A2 (i) guarantees that $m_t(u)$ and $\rho_t(u)$ are continuous on $[0,1] \times \mathbb{R}^d$, they are differentiable in $u$, and their derivatives $\nabla_u m_t(u)$ and $\nabla_u \rho_t(u)$ are also continuous. Since $\rho_t(u) > 0$, the reciprocal $1/\rho_t(u)$ is continuous, and thus $\nabla_u m_t(u)/\rho_t(u)$ is continuous as well. Moreover, $\nabla_u \log \rho_t(u) = \nabla_u \rho_t(u)/\rho_t(u)$ is continuous by the same reasoning. Consequently, both terms on the right-hand side above are continuous, which shows that $\nabla_u v_t(u)$ is continuous. Together with $v_t(u) = m_t(u)/\rho_t(u)$ being continuous, we conclude that both $v_t$ and $\nabla_u v_t$ are continuous on $[0,1] \times \mathbb{R}^d$.

**Part 2: Uniform Lipschitzness of $v_t$.** Fix compact $K$ with radius $R_K := \sup_{u \in K} \|u\|$. From Assumption A2 (ii) and (iii),

$$\sup_{(t,u) \in [0,1] \times K} \left\| \frac{\nabla_u m_t(u)}{\rho_t(u)} \right\| \le \frac{J_K}{\beta_K}.$$

From Cauchy–Schwarz inequality and Assumption A2 (iv),

$$\|v_t(u)\| \le \sqrt{c_0(t)} + \sqrt{c_1(t)} \|u\| \le A + B \|u\|,$$

so $\sup_{(t,u) \in [0,1] \times K} \|v_t(u)\| \le A + B R_K$. Using $\|a \otimes b\| \le \|a\| \|b\|$ and Assumption A2 (ii),

$$\sup_{(t,u) \in [0,1] \times K} \|\nabla_u v_t(u)\| \le \frac{J_K}{\beta_K} + M_K (A + B R_K) =: L_K < \infty.$$

Thus, $\|v_t(u) - v_t(v)\| \le L_K \|u - v\|$ for all $t$ and $u, v \in K$.

**Part 3: Local existence and uniqueness of the ODE solution.** By **Part 1** and **Part 2**, $v_t(u)$ is continuous and uniformly Lipschitz in $u$ on compacts; Picard–Lindelöf theorem yields a unique solution on $[0, \tau)$.

**Part 4: Proof of a priori bound and extension.** Using $\|v_t(U_t)\| \le A + B\|U_t\|$,

$$\frac{\mathrm{d}}{\mathrm{d}t}\|U_t\| \le A + B\|U_t\| \;\Rightarrow\; \|U_t\| \le (\|u\| + At)e^{Bt} \le (\|u\| + A)e^{B} \quad (t \in [0,1]).$$

Hence, there is no finite-time blow-up, the solution extends to $[0,1]$.

**Part 5: Global uniqueness of the ODE solution.** Let $U_t$ and $V_t$ be two solutions with the same initial condition $U_0 = V_0$. By the linear-growth bound $\|v_t(z)\| \le A + B\|z\|$ and Grönwall's inequality, there exists $R < \infty$ such that $\|U_t\| \le R$ and $\|V_t\| \le R$ for all $t \in [0,1]$. Hence both trajectories remain in the common compact ball $K := \{z : \|z\| \le R\}$, on which the field is uniformly (in $t$) Lipschitz in $u$:

$$\|v_t(u) - v_t(v)\| \le L_K \|u - v\|, \qquad \forall t \in [0,1], \forall u, v \in K.$$

Writing the integral form of the ODE and subtracting,

$$U_t - V_t = \int_0^t \big(v_s(U_s) - v_s(V_s)\big)\,\mathrm{d}s,$$

we obtain, with $Z_t := U_t - V_t$,

$$\|Z_t\| \le \int_0^t \|v_s(U_s) - v_s(V_s)\|\,\mathrm{d}s \le \int_0^t L_K\,\|Z_s\|\,\mathrm{d}s.$$

By Grönwall's inequality in integral form, letting $q(t) := \|Z_t\|$ yields $q(t) \le q(0)\exp(L_K t) = 0$, so $q(t) \equiv 0$ and hence $U_t \equiv V_t$. Therefore the solution on $[0,1]$ is unique.

**Part 6: Proof of the Gaussian and Gaussian mixture case.** We first consider the case where $U_1$ follows a Gaussian distribution, and then extend the argument to show that the result also holds when $U_1$ follows a Gaussian mixture distribution. Note that a Gaussian distribution can be regarded as a special case of a Gaussian mixture distribution.

**Gaussian.** Consider $Y = [U_0; U_1] \in \mathbb{R}^{2d}$ such that $Y \sim \mathcal{N}(\mu, \Sigma)$ with mean and covariance:

$$\mu = \begin{pmatrix} \mu_0 \\ \mu_1 \end{pmatrix}, \qquad \Sigma = \begin{pmatrix} \Sigma_{00} & \Sigma_{01} \\ \Sigma_{10} & \Sigma_{11} \end{pmatrix} \succ 0.$$

Define $U_t := (1-t)U_0 + tU_1 = C_t Y$, $D := U_1 - U_0 = BY$, where $C_t = [(1-t)I_d,\ tI_d] \in \mathbb{R}^{d \times 2d}$ and $B = [-I_d,\ I_d] \in \mathbb{R}^{d \times 2d}$. By the Gaussian conditioning formula,

$$Y \mid U_t = u \ \sim\ \mathcal{N}(\mu + \Sigma C_t^\top \Lambda_t(u - C_t\mu),\ \Sigma - \Sigma C_t^\top \Lambda_t C_t \Sigma),$$

with $\Lambda_t := (C_t \Sigma C_t^\top)^{-1}$. Consequently,

$$D \mid U_t = u \ \sim\ \mathcal{N}(m(t,u),\ S(t)),$$

where

$$m(t,u) = \alpha(t) + M(t)u, \qquad S(t) = B\big(\Sigma - \Sigma C_t^\top \Lambda_t C_t \Sigma\big)B^\top,$$

with $\alpha(t) = B\mu - B\Sigma C_t^\top \Lambda_t C_t \mu$ and $M(t) = B\Sigma C_t^\top \Lambda_t$. Thus,

$$\mathbb{E}[\|D\|^2 \mid U_t = u] = \|m(t,u)\|^2 + \operatorname{tr} S(t).$$

Since $\|m(t,u)\|^2 \le 2\|\alpha(t)\|^2 + 2\|M(t)\|_{\mathrm{op}}^2 \|u\|^2$, we obtain

$$\mathbb{E}[\|D\|^2 \mid U_t = u] \le \underbrace{\big(2\|\alpha(t)\|^2 + \operatorname{tr} S(t)\big)}_{=:c_0(t)} + \underbrace{2\|M(t)\|_{\mathrm{op}}^2}_{=:c_1(t)} \|u\|^2.$$

Here $c_0(t), c_1(t)$ are continuous and bounded on $[0,1]$.

**Gaussian mixture.** Consider $U_1 \sim \sum_{k=1}^K \pi_k \mathcal{N}(\mu_{1k}, \Sigma_{1k})$, where $\pi_k > 0$, $\sum_k \pi_k = 1$, $\Sigma_{1k} \succ 0$, and $Z \in \{1, \ldots, K\}$ with $\mathbb{P}(Z = k) = \pi_k$. Conditioned on $Z = k$, $(U_0, U_1)$ is jointly Gaussian, and the bound from the previous **Gaussian** case yields

$$\mathbb{E}[\|D\|^2 \mid U_t = u, Z = k] \le c_{0k}(t) + c_{1k}(t)\|u\|^2.$$

Denote the posterior weight by $w_k(t,u) = \mathbb{P}(Z = k \mid U_t = u)$. Hence

$$\mathbb{E}[\|D\|^2 \mid U_t = u] = \sum_{k=1}^K w_k(t,u)\,\mathbb{E}[\|D\|^2 \mid U_t = u, Z = k]$$

$$\le \Big(\max_{1 \le k \le K} c_{0k}(t)\Big) + \Big(\max_{1 \le k \le K} c_{1k}(t)\Big)\|u\|^2.$$

This completes the proof. $\qquad\square$

## B.2 ESTIMATION ERROR

In this subsection, we characterize the estimation error of the resulting transport map in terms of the estimation accuracy of the probabilistic flow. We require the smoothness condition of the estimated velocity field associated with the learned flow.

**Assumption A3** (Smoothness of velocity field). *Assume that $\widehat{v}_t(u)$ is continuously differentiable in $(t, u)$ and Lipschitz in $u$ uniformily on $(t, u) \in [0, 1] \times \mathbb{R}^d$ with Lipschitz constant $K$.*

Under the uniqueness assumption and the above smoothness condition, we can show that the estimation error of the transport map can be controlled by the uniform (over the time horizon) estimation error of the velocity field. This suggests that if one can estimate the velocity field $v_t$ well enough, then $T$ can also be well approximated in $L_2$ norm.

**Lemma B.2** (Estimation error of transport map). *Under Assumptions A2 and A3, it holds that*

$$\|T - \widehat{T}\|_{L_2}^2 \leq e^{1+2K} \int_0^1 \|v_t - \widehat{v}_t\|_{L_2}^2 \mathrm{d}t.$$

*Proof of Lemma B.2.* By Assumptions A2 and A3, Albergo & Vanden-Eijnden (2023, Proposition 3) gives that

$$
\begin{aligned}
\|T - \widehat{T}\|_{L_2}^2 &= \int_{\mathbb{R}^d} \left| U_1(u) - \widehat{U}_1(u) \right|^2 \rho_0(u) \, \mathrm{d}u \\
&\leq e^{1+2K} \int_0^1 \int_{\mathbb{R}^d} \left| v_t(U_t(u)) - \widehat{v}_t(U_t(u)) \right|^2 \rho_0(u) \, \mathrm{d}u \, \mathrm{d}t \\
&= e^{1+2K} \int_0^1 \|v_t - \widehat{v}_t\|_{L_2}^2 \mathrm{d}t.
\end{aligned}
$$

$\square$

## C    SEMIPARAMETRIC THEORY

### C.1    ASSUMPTIONS

To formalize the asymptotic analysis of our estimator, we rely on the von Mises expansion, which is a form of Taylor series for functionals. This requires a precise notion of a functional derivative. We introduce two related concepts: the Gateaux derivative, which is directional, and the stronger Fréchet derivative, which ensures a uniform linear approximation.

The *Gateaux derivative* of the functional $\phi$ with respect to the function $T$ in the direction of another function $h$ is defined as:

$$\nabla\phi(T)[h] := \lim_{\epsilon \to 0} \frac{\phi(T + \epsilon h) - \phi(T)}{\epsilon}.$$

It describes how the functional behaves along a specific linear path. Gateaux differentiability is also referred to as pathwise differentiability, where the path specifies the direction in the function space. To control the remainder term of the Taylor expansion for any perturbation, not just along straight lines, a stronger condition of Fréchet differentiability is needed to guarantee that the functional can be well-approximated by a linear map in a neighborhood of $T$. Formally, let $V$ and $W$ be normed vector spaces, let $U \subset V$ be an open set, and let $f : U \to W$. The function $f$ is Fréchet differentiable at $x \in U$ if there exists a bounded linear operator $A : V \to W$ such that:

$$\lim_{\|h\|_V \to 0} \frac{\|f(x+h) - f(x) - Ah\|_W}{\|h\|_V} = 0$$

If such a bounded linear operator $A$ exists, it is unique and is called the *Fréchet derivative* of $f$ at $x$, denoted by $Df(x)$.

Within this framework, a pivotal concept is *Neyman orthogonality* (Chernozhukov et al., 2018). This condition is defined using the Gateaux derivative: the functional $\phi$ is Neyman-orthogonal with respect to the nuisance parameter $T$ if its Gateaux derivative is the zero functional, meaning it evaluates to zero for any valid perturbation function $h$:

$$\nabla\phi(T) = 0 \quad \Longleftrightarrow \quad \nabla\phi(T)[h] = 0, \ \forall h.$$

When this condition holds, the first-order bias from the estimation of the nuisance function $T$ vanishes, which simplifies the asymptotic analysis and leads to estimators with only higher-order bias.

For the remaining section, we will write $\phi_{Z_j}^{\text{FDFI}}$ and $\phi_{X_l}^{\text{FDFI}}$ as $\phi_{Z_j}$ and $\phi_{X_l}$, respectively, for notation simplicty. We require assumptions about the data generating process and nuisance function estimation.

**Assumption A4** (Regularity conditions). *For any $\mathbb{P} \in \mathcal{P}$, assume the following holds:*

   (i) *(Smoothness) The map $T \mapsto \omega(O; T)$ is twice Fréchet differentiable with respect to $T$. The loss function $\ell(y, \widehat{y})$ is differentiable with respect to its second argument, and the function $T$ is continuously differentiable.*
  (ii) *(Donsker Class) The class of functions $\{\omega(O; T') : \|T' - T\|_{L_2} < \delta\}$ for some $\delta > 0$ is a $\mathbb{P}$-Donsker class.*
 (iii) *(Neyman orthogonality) Gateaux derivative of $\phi_{Z_j}$ with respect to $T$ satisfies that $\nabla\phi_{Z_j}(T) = 0$.*
 (iv) *(Nuisance estimator influence function) The nuisance function estimator $\widehat{T}$, trained on an independent auxiliary dataset with size $m$, admits an efficient influence function representation. For any point $x$, its estimation error can be linearized as:*

$$\widehat{T}(x) - T(x) = \mathbb{P}_m^{\text{aux}}[\text{IF}_T(O; x)] + \mathcal{O}_{\mathbb{P}}(\|\widehat{T} - T\|_{L_2}^2),$$

*where $\mathbb{P}_m^{\text{aux}}$ is the empirical average over the auxiliary data, and $\mathbb{E}[\text{IF}_T(O; x)] = 0$.*

Assumption A4(i) is smoothness condition, and Assumption A4(ii)-(iii) are analogous to those necessary for the results in Chernozhukov et al. (2018); Williamson & Feng (2020).

Assumption A4(ii) ensures that empirical process terms of the form $(\mathbb{P}_n - \mathbb{P})(\omega(O; T') - \omega(O; T))$ are well-behaved. It can be weakened to proper bounded moment conditions with cross-fitting, as in double machine learning literature (Chernozhukov et al., 2018). However, we stick with the former assumption to simplify the exposition.

Neyman orthogonality condition (Assumption A4(iii)) is the conceptual equivalent of the core requirement in Williamson & Feng (2020), as both require the parameter of interest to be locally quadratic (i.e., first-order insensitive) to estimation errors in the nuisance function. The primary distinction arises from the nature of the nuisance parameter itself: our framework must manage the estimation of the complex generative transport map $T$, whereas related works typically analyze the estimation of the regression function $f$.

The Neyman orthogonality condition Assumption A4(iii) holds in important cases. For instance, for the $\ell_2$ loss, if the predictor $f(x)$ is the true conditional expectation, $f(x) = \mathbb{E}[Y|X = x]$, the Gateaux derivative of the parameter $\phi_{Z_j}$ with respect to $T$ is zero. This occurs because the derivative term involves $\mathbb{E}[Y - f(X) \mid X]$, which is zero by definition. In this scenario, our estimator is first-order insensitive to errors in estimating $T$. The practical implication is that even though $\widehat{T}$ is estimated nonparametrically and may converge slowly, the final estimator $\widehat{\phi}_{Z_j}$ remains $\sqrt{n}$-consistent and asymptotically normal under the alternative without requiring a correction term for the estimation of $T$. When the Neyman orthogonality condition does not hold, the first-order error from estimating the nuisance map $T$ no longer vanishes. To achieve a $\sqrt{n}$-consistent estimator for $\phi_{Z_j}$, we must explicitly correct for this bias using a one-step estimator can be used to correct for first-order bias, which is possible if the nuisance estimator $\widehat{T}$ itself admits a linear expansion, as in Assumption A4(iv).

## C.2 SEMIPARAMETRIC EFFICIENCY

Under Assumption A4 (i), (ii) and (iv), we first derive general EIF in Theorem C.1 without the Neyman orthogonality assumption. We will then specialize this result to prove Theorem 3.1.

**Theorem C.1** (General EIF for Latent FDFI). *Let the data be randomly partitioned into a main sample of size $n$ and an auxiliary sample of size $m$. Let $\widehat{T}$ be an estimator of the nuisance function $T$ computed on auxiliary samples. Under Assumption A4 (i), (ii) and (iv), the cross-fit estimator $\widehat{\phi}_{Z_j}(\mathbb{P}) = \mathbb{P}_n[\omega(O; \widehat{T})]$ for the parameter $\phi_{Z_j}(\mathbb{P}) = \mathbb{E}[\omega(O; T)]$ satisfies the following asymptotic expansion:*

$$\widehat{\phi}_{Z_j}(\mathbb{P}) - \phi_{Z_j}(\mathbb{P}) = (\mathbb{P}_n - \mathbb{P})\{\varphi_{\mathrm{main}}(O; \mathbb{P})\} + (\mathbb{P}_m - \mathbb{P})\{\varphi_{\mathrm{corr}}(O; \mathbb{P})\} + R_{n,m},$$

*where $\mathbb{P}_n$ and $\mathbb{P}_m$ are the empirical measures for the main and auxiliary samples, respectively. The efficient influence function $\varphi_{Z_j}(O; \mathbb{P})$ is given by:*

$$\varphi_{Z_j}(O; \mathbb{P}) = \varphi_{\mathrm{main}}(O; \mathbb{P}) + \varphi_{\mathrm{corr}}(O; \mathbb{P}),$$

*where the components of the efficient influence function are:*

$$\varphi_{\mathrm{main}}(O; \mathbb{P}) = \omega(O; T) - \phi_{Z_j}(\mathbb{P})$$
$$= \frac{1}{2}\left[\ell(f(T^{-1}(Z^{(j)})), Y) - \ell(f(T^{-1}(Z)), Y)\right] - \phi_{Z_j}(\mathbb{P}),$$
$$\varphi_{\mathrm{corr}}(O; \mathbb{P}) = \nabla\phi(T)[\mathrm{IF}_T(O; \cdot)]$$
$$= \frac{1}{2}\left(\mathbb{E}_{X^{(j)}}[D_X(X^{(j)})\mathrm{IF}_T(O)(X^{(j)})] - \mathbb{E}_X[D_X(X)\mathrm{IF}_T(O)(X)]\right).$$

*The remainder term $R_{n,m}$ is of order $\mathcal{O}_{\mathbb{P}}(n^{-1/2}\mathcal{E}_m + \mathcal{E}_m^2)$, where $\mathcal{E}_m = \|\widehat{T} - T\|_{L_2}$ is the $L_2$ error of the nuisance estimator computed on the auxiliary sample. Here, $X = T^{-1}(Z)$, $X^{(j)} = T^{-1}(Z^{(j)})$, $\mathrm{IF}_T(O)(\cdot)$ is the influence function of $\widehat{T}$ evaluated at observation $O$, and*

$$D_X(x) = \mathbb{E}_Y\left[-\frac{\ell'(f(x), Y)f'(x)}{T'(x)} \,\middle|\, X = x\right],$$

*with $\ell'(u, y)$ being the derivative of $\ell$ with respect to its first argument. The expectations $\mathbb{E}_X[\cdot]$ and $\mathbb{E}_{X^{(j)}}[\cdot]$ are taken over the marginal distributions of $X$ and $X^{(j)}$, respectively.*

The EIF consists of the "naive" influence function, $\omega(O; T) - \phi_{Z_j}(\mathbb{P})$, which would be correct if $T$ were known, plus a correction term. This correction term, $\nabla\phi(T)[\mathrm{IF}_T(O; \cdot)]$, directly accounts for

the first-order impact of using an estimate $\widehat{T}$ instead of the true $T$. When the Neyman orthogonality condition holds ($\nabla\phi(T) = 0$), this correction term vanishes, and we recover the simplified EIF from Theorem 3.1. Therefore, this theorem provides a complete characterization of the estimator's asymptotic behavior, nesting the orthogonal case as a special instance.

While the EIF itself is independent of sample sizes, the asymptotic variance of the cross-fit estimator depends on $n$ and $m$. For a cross-fit estimator, its variance is approximated by a sum of variances of the EIF components, scaled by the respective sample sizes:

$$\mathbb{V}(\widehat{\phi}_{Z_j}) \approx \frac{1}{n}\mathbb{V}(\varphi_{\text{main}}) + \frac{1}{m}\mathbb{V}(\varphi_{\text{corr}}),$$

when the remainder term $R_{n,m}$ is negligible.

### C.2.1 PROOF OF THEOREM C.1

*Proof of Theorem C.1.* Note that (5) is a sample average of $M$ copies of $Z_j$. Below, we present the proof for the simplified case $M = 1$, and the conclusion holds for general $M$ using the linearity and independence across $\widehat{Z}_i^{(j,k)}$. The proof proceeds by deriving the von Mises expansion (Du et al., 2025b) for the estimator $\widehat{\phi}_{Z_j} = \mathbb{P}_n[\omega(O; \widehat{T})]$. We decompose the estimation error as follows:

$$
\begin{aligned}
\widehat{\phi}_{Z_j} - \phi_{Z_j} &= \mathbb{P}_n[\omega(O; \widehat{T})] - \mathbb{E}[\omega(O; T)] \\
&= \underbrace{(\mathbb{P}_n - \mathbb{P})[\omega(O; T)]}_{\text{Term I}} + \underbrace{\mathbb{P}[\omega(O; \widehat{T}) - \omega(O; T)]}_{\text{Term II}} + \underbrace{(\mathbb{P}_n - \mathbb{P})[\omega(O; \widehat{T}) - \omega(O; T)]}_{\text{Term III}}.
\end{aligned}
$$

**Term I** is the desired empirical average of the influence function's main part, centered to have zero mean. By the central limit theorem, it converges to a normal distribution when $\mathbb{V}(\psi) > 0$.

**Term II** is the first-order bias term due to plugging in the estimated nuisance function $\widehat{T}$. By Assumption A4 (i), the Taylor expansion of $\phi(T) := \mathbb{E}[\omega(O; T)]$ around the true $T$ gives:

$$\text{Term II} = \phi(\widehat{T}) - \phi(T) = \nabla\phi(T)[\widehat{T} - T] + \mathcal{O}_{\mathbb{P}}(\|\widehat{T} - T\|_{L_2}^2),$$

where $\nabla\phi(T)[\delta_T]$ is the Gateaux derivative of $\phi$ in the direction $\delta_T = \widehat{T} - T$, given by:

$$
\begin{aligned}
\nabla\phi(T)[\delta_T] &= \frac{\mathrm{d}}{\mathrm{d}\epsilon}\bigg|_{\epsilon=0} \mathbb{E}[\omega(O; T + \epsilon\delta_T)] \\
&= \mathbb{E}\left[\frac{\mathrm{d}}{\mathrm{d}\epsilon}\bigg|_{\epsilon=0} \frac{1}{2}\left(\ell(f((T + \epsilon\delta_T)^{-1}(Z^{(j)})), Y) - \ell(f((T + \epsilon\delta_T)^{-1}(Z)), Y)\right)\right].
\end{aligned}
$$

Using the chain rule and the identity $\frac{\mathrm{d}}{\mathrm{d}\epsilon}|_{\epsilon=0}(T + \epsilon\delta_T)^{-1}(z) = -\frac{\delta_T(T^{-1}(z))}{T'(T^{-1}(z))}$, we obtain:

$$\nabla\phi(T)[\delta_T] = \frac{1}{2}\mathbb{E}\left[\ell'(f(X), Y)f'(X)\left(-\frac{\delta_T(X)}{T'(X)}\right) - \ell'(f(X^{(j)}), Y)f'(X^{(j)})\left(-\frac{\delta_T(X^{(j)})}{T'(X^{(j)})}\right)\right],$$

where $X = T^{-1}(Z)$ and $X^{(j)} = T^{-1}(Z^{(j)})$. Using the law of iterated expectations and the definition of $D_X(x)$, this simplifies to:

$$\nabla\phi(T)[\delta_T] = \frac{1}{2}\left(\mathbb{E}_{X^{(j)}}[D_X(X^{(j)})\delta_T(X^{(j)})] - \mathbb{E}_X[D_X(X)\delta_T(X)]\right).$$

By Assumption A4 (iv), $\widehat{T} - T$ can be represented via its influence function: $\widehat{T}(x) - T(x) = \mathbb{P}_m^{\text{aux}}[\text{IF}_T(O; x)] + \mathcal{O}_{\mathbb{P}}(\|\widehat{T} - T\|_{L_2}^2)$. Substituting $\delta_T = \widehat{T} - T$ and its IF representation:

$$
\begin{aligned}
\nabla\phi(T)[\widehat{T} - T] &= \nabla\phi(T)\left[\mathbb{P}_m^{\text{aux}}[\text{IF}_T(O; \cdot)]\right] + \mathcal{O}_{\mathbb{P}}(\|\widehat{T} - T\|_{L_2}^2) \\
&= \frac{1}{n}\sum_{i=1}^{m}\nabla\phi(T)[\text{IF}_T(O_i^{\text{aux}}; \cdot)] + \mathcal{O}_{\mathbb{P}}(\mathcal{E}_m^2) \\
&= \mathbb{P}_m[-\varphi_{\text{corr}}(O_i)] + \mathcal{O}_{\mathbb{P}}(\mathcal{E}_m^2),
\end{aligned}
$$

where $\varphi_{\text{corr}}(O) = -\nabla\phi(T)[\text{IF}_T(O;\cdot)]$. Note that $\mathbb{E}[\varphi_{\text{corr}}(O)] = -\nabla\phi(T)[\mathbb{E}[\text{IF}_T(O;\cdot)]] = 0$. Thus, $\mathbb{P}_m[-\varphi_{\text{corr}}] = (\mathbb{P}_m - \mathbb{P})[-\varphi_{\text{corr}}]$ and Term II reduces $(\mathbb{P}_m - \mathbb{P})[-\varphi_{\text{corr}}] + \mathcal{O}_{\mathbb{P}}(\mathcal{E}_m^2)$.

**Term III** is an empirical process term. Under Assumption A4 (ii) and (iv), this term is of a smaller order. Specifically, $(\mathbb{P}_n - \mathbb{P})[g(\widehat{T}) - g(T)]$ is stochastically equicontinuous, leading to Term III being $\mathcal{O}_{\mathbb{P}}(n^{-1/2}\|\widehat{T} - T\|) = \mathcal{O}_{\mathbb{P}}(n^{-1/2}\mathcal{E}_m)$.

Combining terms, we have:

$$\widehat{\phi}_{Z_j} - \phi_{Z_j} = (\mathbb{P}_n - \mathbb{P})\{\omega(O;T) - \phi_{Z_j}(\mathbb{P})\} - (\mathbb{P}_m - \mathbb{P})\{\varphi_{\text{corr}}(O)\} + R_{n,m}$$
$$= (\mathbb{P}_n - \mathbb{P})\{\omega(O;T) - \phi_{Z_j}(\mathbb{P})\} + (\mathbb{P}_m - \mathbb{P})\{\nabla\phi(T)[\text{IF}_T(O;\cdot)]\} + R_{n,m},$$

where the remainder $R_{n,m} = \mathcal{O}_{\mathbb{P}}(n^{-1/2}\mathcal{E}_m + \mathcal{E}_m^2)$. This completes the proof and establishes the form of the efficient influence function $\varphi_{Z_j}(O;\mathbb{P})$. □

### C.2.2 PROOF OF THEOREM 3.1

*Proof of Theorem 3.1.* The derivation of the EIF follows from Theorem C.1 by noting that the correction term in the definition of efficient influence function vanishes when the Neyman orthogonality condition holds ($\nabla\phi(T) = 0$). This shows that $\varphi_{Z_j}(O;\mathbb{P}) := \omega(O;T) - \phi_{Z_j}(\mathbb{P})$ is the EIF:

$$\widehat{\phi}_{Z_j}(\mathbb{P}) - \phi_{Z_j}(\mathbb{P}) = (\mathbb{P}_n - \mathbb{P})\{\varphi_{Z_j}(O;\mathbb{P})\} + \mathcal{O}_{\mathbb{P}}(n^{-1/2}\mathcal{E}_m + \mathcal{E}_m^2),$$

where $\mathcal{E}_m = \|T - \widehat{T}\|_{L_2}$.

Recall that $\widehat{U}_t$ is the flow map obtained by solving the above ODE with $v_t$ replaced with $\widehat{v}_t$, and the transport map can be represented by $\widehat{T} = \widehat{U}_1$.

From Lemma B.2 and the rate condition, we have

$$\mathcal{E}_m^2 = \|T - \widehat{T}\|_{L_2}^2 \leq e^{1+2K} \int_0^1 \|v_t - \widehat{v}_t\|_{L_2}^2 \mathrm{d}t = o_{\mathbb{P}}(n^{-\frac{1}{2}}).$$

Therefore, we conclude that

$$\widehat{\phi}_{Z_j}(\mathbb{P}) - \phi_{Z_j}(\mathbb{P}) = (\mathbb{P}_n - \mathbb{P})\{\varphi_{Z_j}(O;\mathbb{P})\} + o_{\mathbb{P}}(n^{-1/2}),$$

and consequently, the asymptotic normality follows. □

### C.3 PROOF OF PROPOSITION 3.2

*Proof of Proposition 3.2.* The proof proceeds by first deriving the efficient influence function (EIF) for a single component $\phi_{jl}(\mathbb{P})$ of the total importance score $\phi_{X_l}(\mathbb{P})$, and then aggregating the results. By the linearity of the influence function operator, the EIF for $\phi_{X_l}(\mathbb{P}) = \sum_{j=1}^d \phi_{jl}(\mathbb{P})$ is simply the sum of the EIFs for each component, i.e., $\varphi_{X_l}(O;\mathbb{P}) = \sum_{j=1}^d \varphi_{jl}(O;\mathbb{P})$.

**Part 1: Deriving the EIF for a single component $\phi_{jl}(\mathbb{P})$.** Let $\phi_{jl}(T) = \mathbb{E}[\omega_j(O;T)H_{jl}(X;T)]$, where we make the dependence of the sensitivity term $H_{jl}(X) = (\partial X_l/\partial Z_j)^2$ on the transport map $T$ explicit. The cross-fit estimator is $\widehat{\phi}_{jl} = \mathbb{P}_n[\omega_j(O;\widehat{T})H_{jl}(X;\widehat{T})]$. We perform a von Mises expansion of the estimation error $\widehat{\phi}_{jl}(\mathbb{P}) - \phi_{jl}(\mathbb{P})$:

$$\widehat{\phi}_{jl}(\mathbb{P}) - \phi_{jl}(\mathbb{P}) = \mathbb{P}_n[\omega_j(O;\widehat{T})H_{jl}(X;\widehat{T})] - \mathbb{E}[\omega_j(O;T)H_{jl}(X;T)]$$
$$= \underbrace{(\mathbb{P}_n - \mathbb{P})[\omega_j(O;T)H_{jl}(X;T)]}_{\text{Term I}} + \underbrace{\mathbb{E}[\omega_j(O;\widehat{T})H_{jl}(X;\widehat{T}) - \omega_j(O;T)H_{jl}(X;T)]}_{\text{Term II}}$$
$$+ \underbrace{(\mathbb{P}_n - \mathbb{P})[\omega_j(O;\widehat{T})H_{jl}(X;\widehat{T}) - \omega_j(O;T)H_{jl}(X;T)]}_{\text{Term III}}.$$

Next, we analyze each term separately.

**Term I** is the standard empirical process term that converges to a normal distribution after $\sqrt{n}$ scaling by the Central Limit Theorem.

**Term II** is the first-order bias from due to the nuisance estimator $\widehat{T}$. We linearize this term using the functional derivative of $\phi_{jl}(T)$. Let $\delta_T = \widehat{T} - T$. By Fréchet differentiability in Assumption A4 (i) and the functional product rule:

$$
\begin{aligned}
\text{Term II} &= \phi_{jl}(\widehat{T}) - \phi_{jl}(T) \\
&= \nabla_T \phi_{jl}(T)[\delta_T] + \mathcal{O}_p(\|\delta_T\|_{L_2}^2) \\
&= \mathbb{E}\left[(\nabla_T \omega_j(O;T)[\delta_T]) H_{jl}(X;T)\right] + \mathbb{E}\left[\omega_j(O;T)(\nabla_T H_{jl}(X;T)[\delta_T])\right] + \mathcal{O}_p(\|\delta_T\|_{L_2}^2).
\end{aligned}
$$

When the predictor $f$ is Bayes optimal, the parameter $\phi_{Z_j}(T) = \mathbb{E}[\omega_j(O;T)]$ is Neyman-orthogonal with respect to $T$. This implies that its Gateaux derivative $\nabla_T \phi_{Z_j}(T)[\cdot] = \mathbb{E}[\nabla_T \omega_j(O;T)[\cdot]]$ is zero. Under regularity conditions allowing the interchange of derivative and expectation, the weighted expectation also vanishes, i.e., $\mathbb{E}[(\nabla_T \omega_j(O;T)[\delta_T]) H_{jl}(X;T)] = 0$.

Therefore, the bias term simplifies to its second component. Using the influence function representation for $\widehat{T}$ from Assumption A4 (iv), $\delta_T = \widehat{T} - T = \mathbb{P}_m^{\text{aux}}[\text{IF}_T(O^{\text{aux}};\cdot)] + \mathcal{O}_{\mathbb{P}}(\mathcal{E}_m^2)$, we have:

$$
\begin{aligned}
\text{Term II} &= \mathbb{E}\left[\omega_j(O;T)(\nabla_T H_{jl}(X;T)[\mathbb{P}_m^{\text{aux}}[\text{IF}_T]])\right] + \mathcal{O}_{\mathbb{P}}(\mathcal{E}_m^2) \\
&= \frac{1}{n_{\text{aux}}} \sum_{i=1}^{n_{\text{aux}}} \mathbb{E}\left[\omega_j(O;T)(\nabla_T H_{jl}(X;T)[\text{IF}_T(O_i^{\text{aux}};\cdot)])\right] + \mathcal{O}_{\mathbb{P}}(\mathcal{E}_m^2) \\
&= (\mathbb{P}_m^{\text{aux}} - \mathbb{P})\left\{\mathbb{E}_{O'}\left[\omega_j(O';T)(\nabla_T H_{jl}(X';T)[\text{IF}_T(O;\cdot)])\right]\right\} + \mathcal{O}_{\mathbb{P}}(\mathcal{E}_m^2),
\end{aligned}
$$

where the final step uses $\mathbb{E}[\text{IF}_T] = 0$. This term is the empirical average of the correction term evaluated on the auxiliary data.

**Term III** is a higher-order empirical process term. Under Donsker condition (Assumption A4 (ii)), this term is of order $o_{\mathbb{P}}(n^{-1/2})$.

**Part 2: Asymptotic normality.** Combining the terms, and noting that the main sample (for $\mathbb{P}_n$) and the auxiliary sample (for $\mathbb{P}_m^{\text{aux}}$) are independent, the total estimation error can be written as an empirical average over a single sample:

$$
\begin{aligned}
\widehat{\phi}_{jl}(\mathbb{P}) - \phi_{jl}(\mathbb{P}) = (\mathbb{P}_n - \mathbb{P})\{&\omega_j(O;T) H_{jl}(X;T) \\
&+ \mathbb{E}_{O'}\left[\omega_j(O';T)(\nabla_T H_{jl}(X';T)[\text{IF}_T(O;\cdot)])\right]\} + \mathcal{O}_{\mathbb{P}}(\mathcal{E}_m^2).
\end{aligned}
$$

By subtracting the mean $\phi_{jl}(\mathbb{P})$ from the first part, we identify the EIF for $\phi_{jl}$ as:

$$
\varphi_{jl}(O;\mathbb{P}) = \underbrace{(\omega_j(O;T) H_{jl}(X;T) - \phi_{jl}(\mathbb{P}))}_{\text{Naive Term}} + \underbrace{\mathbb{E}_{O'}\left[\omega_j(O';T)(\nabla_T H_{jl}(X';T)[\text{IF}_T(O;\cdot)])\right]}_{\text{Correction Term}}.
$$

By defining $\text{IF}_{H_{jl}} := \nabla_T H_{jl}[\text{IF}_T]$ as the influence function of the estimator for the function $H_{jl}$, and using the fact that influence functions have zero mean, the term becomes a covariance $\text{Cov}(\omega_j(O;T), \text{IF}_{H_{jl}}(O;\cdot))$. Thus, the total EIF for $\phi_{X_l}(\mathbb{P})$ is $\varphi_{X_l}(O;\mathbb{P}) = \sum_{j=1}^{d} \varphi_{jl}(O;\mathbb{P})$. The vector of estimators $\widehat{\phi}_X$ thus has the asymptotic linear expansion:

$$
\sqrt{n}(\widehat{\phi}_X(\mathbb{P}) - \phi_X(\mathbb{P})) = \frac{1}{\sqrt{n}} \sum_{i=1}^{n} \psi(O_i;\mathbb{P}) + o_{\mathbb{P}}(1),
$$

where $\psi(O_i;\mathbb{P}) = (\psi_1(O_i;\mathbb{P}), \ldots, \psi_d(O_i;\mathbb{P}))^\top$. By the multivariate Central Limit Theorem, it follows that $\sqrt{n}(\widehat{\phi}_X(\mathbb{P}) - \phi_X(\mathbb{P})) \xrightarrow{\text{d}} \mathcal{N}(0, \Sigma_\phi)$, where $\Sigma_\phi = \mathbb{E}[\psi(O;\mathbb{P})\psi(O;\mathbb{P})^\top]$, with entries $(\Sigma_\phi)_{lk} = \text{Cov}(\varphi_{X_l}(O;\mathbb{P}), \psi_k(O;\mathbb{P}))$. This completes the proof. □

# D    COMPUTATIONAL DETAILS

## D.1    ALGORITHM AND COMPUTATIONAL DEVICES

**Algorithm.**    The full algorithm of FDFI is given in Algorithm D.1.

---

**Algorithm D.1** Flow-disentangled feature importance

---

**Require:** Labeled data $\mathcal{D}_{\text{est}} = \{O_i = (X_i, Y_i)\}_{i=1}^n$, a black-box model $f$, a loss function $\ell$. Independent auxiliary unlabeled data $\mathcal{D}_X = \{\widetilde{X}_i\}_{i=1}^m$ and a flow model training procedure $\mathcal{M}(\cdot)$ that returns a map $\widehat{T}$. The null adjustment constant $c$ and the Monte Carlo sample size $M$.

**Ensure:** FDFI scores $\{\widehat{\phi}_{Z_j}\}_{j=1}^d$, $\{\widehat{\phi}_{X_l}\}_{l=1}^d$ and p-values $\{p_{Z_j}\}_{j=1}^d$, $\{p_{X_l}\}_{l=1}^d$.

1: Obtain transport map through flow matching $\widehat{T} = \mathcal{M}(\mathcal{D}_X)$.        ▷ Disentangled map estimation
2: Initialize point-wise score storage: $\Psi \leftarrow$ empty $n \times d$ matrix, $\Omega \leftarrow$ empty $n \times d$ matrix.
3: **for** $i = 1$ **to** $n$ **do**                           ▷ Compute scores on the labeled data
4:    Compute latent vector $\widehat{Z}_i \leftarrow \widehat{T}(X_i)$ and generative Jacobian $J_i \leftarrow \nabla_Z \widehat{T}^{-1}(Z)|_{Z=\widehat{Z}_i}$.
5:    Compute squared sensitivities $\widehat{H}_{jl}(\widehat{Z}_i) \leftarrow (J_i)_{lj}^2$ for all $j, l \in \{1, \ldots, d\}$.
6:    **for** $j = 1$ **to** $d$ **do**                        ▷ Point-wise latent scores
7:       Let $\{\widehat{Z}_i^{(j,m)}\}_{m=1}^M$ be $M$ copies of $\widehat{Z}_i$ where the $j$-th coord is resampled from $p_{Z_j}$.
8:       $\Omega_{ij} \leftarrow \frac{1}{2M} \sum_{k=1}^M \left[ \ell(Y_i, f(\widehat{T}^{-1}(\widehat{Z}_i^{(j,k)}))) - \ell(Y_i, f(X_i)) \right]$.
9:    **end for**
10:    **for** $l = 1$ **to** $d$ **do**                       ▷ Attribution to original features
11:       $\Psi_{il} \leftarrow \sum_{j=1}^d \Omega_{ij} \cdot \widehat{H}_{jl}(\widehat{Z}_i)$.
12:    **end for**
13: **end for**
14: **for** $j = 1$ **to** $d$ **do**                          ▷ Compute latent feature importance $\phi_{Z_j}$
15:    $\widehat{\phi}_{Z_j} \leftarrow \frac{1}{n} \sum_{i=1}^n \Omega_{ij}$.                          ▷ Cross-fit estimator (5)
16:    $\widehat{\varphi}_{ij} \leftarrow \Omega_{ij} - \widehat{\phi}_{Z_j}$ for $i = 1 \ldots n$.                    ▷ EIF components
17:    $\text{se}_{Z_j}^2 \leftarrow (\mathbb{V}_n\{\widehat{\varphi}_{ij}\} + c)/n$.                ▷ Adjust standard error for inference
18:    $p_{Z_j} \leftarrow 1 - \Phi(\widehat{\phi}_{Z_j}/\text{se}_{Z_j})$.                          ▷ Compute p-values
19: **end for**
20: **for** $l = 1$ **to** $d$ **do**                          ▷ Compute original feature importance $\phi_{X_l}$
21:    $\widehat{\phi}_{X_l} \leftarrow \frac{1}{n} \sum_{i=1}^n \Psi_{il}$.                           ▷ Estimator (7)
22:    $\widehat{\varphi}_{il}^{\text{a}} \leftarrow \Psi_{il} - \widehat{\phi}_{X_l}$ for $i = 1 \ldots n$.                ▷ Approximate EIF components
23:    $\text{se}_{X_l}^2 \leftarrow (\mathbb{V}_n\{\widehat{\varphi}_{il}^{\text{a}}\} + c)/n$.                ▷ Adjust standard error for inference
24:    $p_{X_l} \leftarrow 1 - \Phi(\widehat{\phi}_{X_l}/\text{se}_{X_l})$.                          ▷ Compute p-values
25: **end for**
26: **return** Estimated importance scores and their uncertainty.

---

**Variants of FDFI estimators.**    There are two different ways to construct an estimator for latent FDFI (2). One can adopt the idea of CPI (Strobl et al., 2008), SCPI (Reyero-Lobo et al., 2026), and LOCO (Lei et al., 2018) to estimate this quantity. More specifically, we can define

$$\widehat{\phi}_{Z_j}^{\text{CPI}}(\mathbb{P}) := \frac{1}{n} \sum_{i=1}^n \left[ \left( \frac{1}{2M} \sum_{k=1}^M \left[ \ell(Y_i, f(\widehat{T}^{-1}(\widehat{Z}_i^{(j,k)}))) - \ell(Y_i, f(\widehat{T}^{-1}(\widehat{Z}_i))) \right] \right) \right], \quad (12)$$

$$\widehat{\phi}_{Z_j}^{\text{SCPI}}(\mathbb{P}) := \frac{1}{n} \sum_{i=1}^n \left[ \ell\left(Y_i, \frac{1}{M} \sum_{k=1}^M \left[ f(\widehat{T}^{-1}(\widehat{Z}_i^{(j,k)})) \right] \right) - \ell(Y_i, f(\widehat{T}^{-1}(\widehat{Z}_i))) \right], \quad (13)$$

where (12) coincides with the latent FDFI estimator (5) presented in the main text.

The final FDFI estimator of the original feature can be constructed as follows:

$$\widehat{\phi}_{X_l}^{\mathrm{CPI}}(\mathbb{P}) := \sum_{j=1}^{d} \frac{1}{n} \sum_{i=1}^{n} \left[ \frac{1}{2M} \sum_{k=1}^{M} \left[ \ell(Y_i, f(\widehat{T}^{-1}(\widehat{Z}_i^{(j,k)}))) - \ell(Y_i, f(\widehat{T}^{-1}(\widehat{Z}_i))) \right] \widehat{H}_{jl}(\widehat{Z}_i) \right] \quad (14)$$

$$\widehat{\phi}_{X_l}^{\mathrm{SCPI}}(\mathbb{P}) := \sum_{j=1}^{d} \frac{1}{n} \sum_{i=1}^{n} \left[ \ell\left( Y_i, \frac{1}{M} \sum_{k=1}^{M} \left[ f(\widehat{T}^{-1}(\widehat{Z}_i^{(j,k)})) \right] \right) - \ell(Y_i, f(\widehat{T}^{-1}(\widehat{Z}_i))) \right] \widehat{H}_{jl}(\widehat{Z}_i), \quad (15)$$

where $\widehat{H}_{jl}(Z) = [\nabla \widehat{T}^{-1}(Z)]_{jl}^2$ is the square of estimated Jacobian of $X_l$ with respect to $Z_j$.

One can, in principle, construct a LOCO-type estimator; however, this requires refitting submodels for $f \circ T^{-1}$, which is computationally expensive in general. Except in the special case of $\ell_2$ loss, the LOCO-type estimator coincides with the SCPI-type estimator if $f$ is the Bayes optimal predictor. For this reason, we didn't explore this variant in the current paper.

**Computational devices.** All experiments were conducted on dedicated computing platforms. All experiments, except for Appendix D.4, were executed on a server equipped with an AMD EPYC 7542 32-Core Processor CPU and NVIDIA RTX 3090 GPUs. Appendix D.4 was carried out on a personal computer with an Intel Core i5-14600KF CPU and an NVIDIA RTX 5070 Ti GPU.

### D.2 FLOW MATCHING MODEL

Flow matching is a concise and powerful framework for generative modeling that has advanced the state of the art across various domains and applications. Following Lipman et al. (2022; 2024), we adopt the Conditional Flow Matching (CFM) model as the backbone of our approach. Similar to the setting in Lipman et al. (2024), we train the model for 5000 steps per 1000 samples. Because performance is sensitive to architectural and optimization choices—including network depth, hidden dimensions, and batch size—we systematically explore multiple hyperparameter combinations to identify a configuration that offers a favorable performance–efficiency trade-off for training.

In statistics, the Maximum Mean Discrepancy (MMD) is a widely used metric for quantifying the difference between two probability distributions. A smaller MMD value indicates that the two distributions are more similar, whereas a larger value reflects greater divergence. For the target data, we split the dataset into training and testing subsets with a 1:1 ratio. Using the training set, we trained the model to generate samples from a standard normal distribution, thereby obtaining the generated target distribution. We then computed the $\mathrm{MMD}^2$ between the real and generated distributions on both the training and testing sets. These results were used as the criterion for selecting appropriate configurations.

To select a suitable hyperparameter configuration, we proceeded as follows. As an example, in Section 4.1 we fixed the training sample size at 1000 and trained the model for 5000 steps using the Adam optimizer with a learning rate of $1 \times 10^{-3}$. In each trial, a distinct random seed was used to generate a new simulated dataset; for that dataset, we evaluated multiple hyperparameter configurations. This procedure was repeated for 50 seeds in total, and for every configuration-seed pair, we recorded the resulting $\mathrm{MMD}^2$ on both the training and test sets.

A suitable set of hyperparameters should yield relatively small $\mathrm{MMD}^2$ values on both the training and test sets. Moreover, the gap between training and test $\mathrm{MMD}^2$ values should remain modest; otherwise, the model risks underfitting or overfitting. As shown in Table D1, the configuration with hidden dimension 128, batch size 256, and a two-layer network satisfies the above criteria: it attains relatively small $\mathrm{MMD}^2$ on both training and test sets with only a modest gap between them. By contrast, smaller hidden dimensions and a single-layer network yield large $\mathrm{MMD}^2$ on both splits, indicating underfitting. Increasing the hidden dimension and depth further reduces the training $\mathrm{MMD}^2$ but enlarges the train–test gap, signaling an overfitting risk. Although the setting with hidden dimension 256, batch size 128, and two layers performs comparably, we ultimately select the hidden-dimension-128, batch-size-256, two-layer configuration as it offers a more favorable balance between model performance and computational efficiency.

After identifying an appropriate hyperparameter configuration based on the preceding selection procedure, we examined the effect of proportionally increasing the training sample size and steps, scal-

**Table D1:** Train and test $\mathrm{MMD}^2$ (mean $\pm$ std) across hidden dimensions, batch sizes, and network depths.

| Hidden dim | Batch size | Train (1 layer) | Test (1 layer) | Train (2 layers) | Test (2 layers) | Train (3 layers) | Test (3 layers) |
|---|---|---|---|---|---|---|---|
| 64 | 32 | 0.0100±0.0069 | 0.0109±0.0056 | 0.0079±0.0039 | 0.0096±0.0052 | 0.0076±0.0044 | 0.0084±0.0042 |
| 64 | 64 | 0.0065±0.0040 | 0.0073±0.0047 | 0.0072±0.0038 | 0.0080±0.0039 | 0.0055±0.0037 | 0.0069±0.0054 |
| 64 | 128 | 0.0071±0.0043 | 0.0076±0.0049 | 0.0053±0.0028 | 0.0058±0.0033 | 0.0038±0.0020 | 0.0046±0.0023 |
| 64 | 256 | 0.0052±0.0028 | 0.0064±0.0031 | 0.0047±0.0025 | 0.0059±0.0027 | 0.0041±0.0029 | 0.0046±0.0025 |
| 64 | 384 | 0.0048±0.0024 | 0.0054±0.0027 | 0.0041±0.0031 | 0.0057±0.0032 | 0.0038±0.0022 | 0.0046±0.0020 |
| 128 | 32 | 0.0062±0.0025 | 0.0068±0.0024 | 0.0057±0.0041 | 0.0067±0.0045 | 0.0061±0.0032 | 0.0070±0.0030 |
| 128 | 64 | 0.0052±0.0032 | 0.0062±0.0034 | 0.0042±0.0029 | 0.0059±0.0041 | 0.0043±0.0025 | 0.0058±0.0025 |
| 128 | 128 | 0.0050±0.0024 | 0.0061±0.0023 | 0.0036±0.0023 | 0.0046±0.0026 | 0.0027±0.0021 | 0.0035±0.0020 |
| 128 | 256 | 0.0044±0.0026 | 0.0052±0.0019 | **0.0024±0.0022** | **0.0030±0.0022** | 0.0021±0.0016 | 0.0035±0.0015 |
| 128 | 384 | 0.0041±0.0020 | 0.0052±0.0020 | 0.0030±0.0022 | 0.0040±0.0026 | 0.0023±0.0014 | 0.0033±0.0011 |
| 256 | 32 | 0.0036±0.0017 | 0.0041±0.0018 | 0.0035±0.0018 | 0.0041±0.0022 | 0.0032±0.0019 | 0.0039±0.0028 |
| 256 | 64 | 0.0026±0.0016 | 0.0034±0.0017 | 0.0024±0.0019 | 0.0032±0.0026 | 0.0025±0.0021 | 0.0035±0.0019 |
| 256 | 128 | 0.0018±0.0012 | 0.0028±0.0016 | **0.0023±0.0016** | **0.0028±0.0015** | 0.0020±0.0012 | 0.0031±0.0015 |
| 256 | 256 | 0.0017±0.0014 | 0.0027±0.0019 | 0.0014±0.0012 | 0.0025±0.0010 | 0.0011±0.0006 | 0.0026±0.0008 |
| 256 | 384 | 0.0024±0.0018 | 0.0031±0.0019 | 0.0013±0.0011 | 0.0024±0.0013 | 0.0012±0.0011 | 0.0029±0.0010 |
| 384 | 32 | 0.0016±0.0010 | 0.0024±0.0019 | 0.0015±0.0009 | 0.0028±0.0012 | 0.0027±0.0021 | 0.0040±0.0024 |
| 384 | 64 | 0.0016±0.0011 | 0.0026±0.0017 | 0.0013±0.0011 | 0.0023±0.0013 | 0.0022±0.0010 | 0.0029±0.0013 |
| 384 | 128 | 0.0015±0.0013 | 0.0030±0.0012 | 0.0010±0.0008 | 0.0019±0.0010 | 0.0016±0.0011 | 0.0023±0.0011 |
| 384 | 256 | 0.0016±0.0013 | 0.0025±0.0014 | 0.0012±0.0009 | 0.0023±0.0012 | 0.0020±0.0009 | 0.0036±0.0015 |
| 384 | 384 | 0.0018±0.0008 | 0.0027±0.0011 | 0.0015±0.0011 | 0.0027±0.0009 | 0.0023±0.0014 | 0.0039±0.0018 |

ing from $(1000, 5000)$ to $(2000, 10000)$ and beyond. For comparability, the procedure was repeated 50 times, each with a fixed random seed; within each run, datasets of different sizes were generated using the same seed to ensure consistency. The results in Figure D1 show that when the sample size reaches 3000 or above, the $\mathrm{MMD}^2$ values on both training and test sets stabilize. While larger sample sizes can still improve generation quality, we adopt 3000 as the sample size, balancing performance and computational cost.

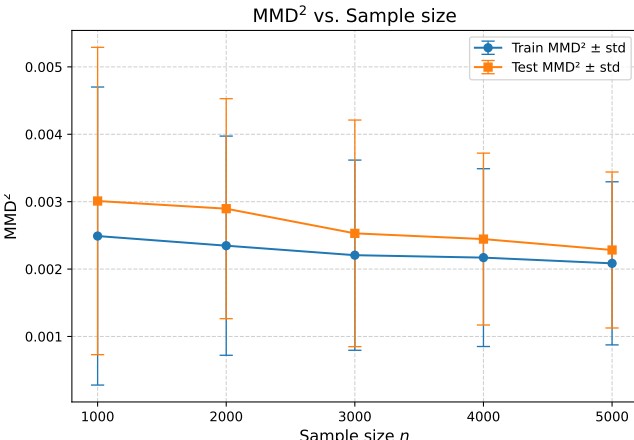

**Figure D1:** Comparison of Train and Test $\mathrm{MMD}^2$ across varying sample sizes.

### D.3    Statitsical inference

#### D.3.1    Approximation for EIF of $\phi_{X_l}$

Recall that the component EIF from Proposition 3.2 is given by

$$\varphi_{jl}(O; \mathbb{P}) := (\omega_j(O; T) H_{jl}(X) - \phi_{jl}(\mathbb{P})) + \mathrm{Cov}(\omega_j(O; T), \mathrm{IF}_{H_{jl}}(O; \cdot))]$$
$$=: \varphi_{jl}^{\mathrm{a}}(O; \mathbb{P}) + C_{jl}.$$

The first-order approximate term $\varphi_{jl}^{\mathrm{a}}(O; \mathbb{P})$ can be estimated using a cross-fit estimator; the correction term $C$, however, requires evaluation of the influence function of $H_{jl}$. Analytically deriving $\mathrm{IF}_T$ (and thus $\mathrm{IF}_{H_{jl}}$) for a flow-based model is generally intractable. In practice, the correction term must be estimated using resampling methods.

---

**Algorithm D.2** Jackknife estimation of the EIF correction term ($C_{jl}$)

---

**Require:** Dataset $\mathcal{D} = \{O_i\}_{i=1}^n$, where $O_i = (X_i, Y_i)$. Flow model training procedure $\mathcal{M}(\cdot)$ that returns a map $\widehat{T}$. Point-wise latent score function $\omega_j(O; T)$.
**Ensure:** Estimated correction term $\widehat{C}_{jl} = \mathbb{C}_n(\omega_j(O; T), \mathrm{IF}_{H_{jl}}(O; \cdot))$.
  1: Initialize score vector $W \leftarrow []$.                          ▷ Step 1: Compute point-wise scores using $\widehat{T}$.
  2: **for** $i = 1$ **to** $n$ **do**
  3:     $W_i \leftarrow \omega_j(O_i; \widehat{T})$
  4:     Append $W_i$ to $W$.
  5: **end for**
  6: Let $\widehat{H}_{jl}(\cdot)$ be the estimator function derived from the full map $\widehat{T}$.
  7: Initialize influence vector $I \leftarrow []$.
  8: **for** $i = 1$ **to** $n$ **do**
  9:     Let $\mathcal{D}_X^{(-i)} \leftarrow \{X_1, \ldots, X_{i-1}, X_{i+1}, \ldots, X_n\}$.
 10:     Retrain model: $\widehat{T}^{(-i)} \leftarrow \mathcal{M}(\mathcal{D}_X^{(-i)})$.              ▷ Step 2: Compute leave-one-out estimate.
 11:     Derive the leave-one-out estimator function $\widehat{H}_{jl}^{(-i)}(\cdot)$ from $\widehat{T}^{(-i)}$.
 12:     $I_i \leftarrow (n-1)\left(\widehat{H}_{jl}(X_i) - \widehat{H}_{jl}^{(-i)}(X_i)\right)$              ▷ Step 3: Estimate point-wise influence.
 13:     Append $I_i$ to $I$.
 14: **end for**
 15: $\widehat{C}_{jl} \leftarrow \mathbb{C}_n(W, I)$.                          ▷ Step 4: Compute the sample covariance.
 16: **return** $\widehat{C}_{jl}$

---

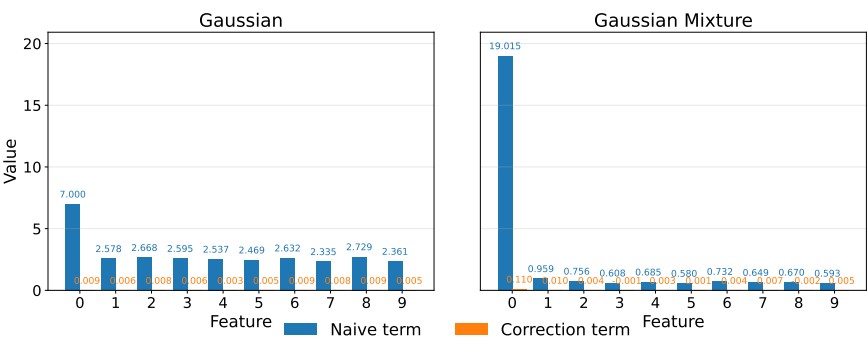

**Figure D2:** Comparison between first-order approximate term $\varphi_{jl}^{\mathrm{a}}(O; \mathbb{P})$ and the correction term $C_{jl}$. The correction term $C_{jl}$ is of smaller order than the first-order approximate term $\varphi_{jl}^{\mathrm{a}}(O; \mathbb{P})$.

The influence function $\mathrm{IF}_{H_{jl}}$ measures the sensitivity of the estimator $\widehat{H}_{jl}$ to the data. To measure this, one must re-estimate $\widehat{H}_{jl}$ on perturbed versions of the data. A direct, though computationally intensive, method to estimate the covariance term is the jackknife, as outlined in Algorithm D.2.

We fix the dimension $d = 10$ and study the response model $y = 5X_0 + \epsilon$, where $\epsilon \sim \mathcal{N}(0, 1)$. The covariates $X \in \mathbb{R}^{10}$ are sampled under two regimes: (i) **Gaussian.** $X \sim \mathcal{N}(0, \Sigma_{0.8})$, and (ii) **Gaussian mixture.** $X \sim 0.2\mathcal{N}(0, \Sigma_{0.8}) + 0.8\mathcal{N}(0, \Sigma_{0.2})$, where $\Sigma_\rho \in \mathbb{R}^{10 \times 10}$, with $[\Sigma_\rho]_{ij} = 1$ if $i = j$ and $\rho$ otherwise, is the AR1 covariance matrix with parameter $\rho$.

In this part, we choose the sample size as $n = 1000$ and compute the approximate term $\varphi_{jl}^{\mathrm{a}}(O; \mathbb{P})$ and the correction term $C_{jl}$, respectively. Crucially, this experiment is conducted without sample splitting; the same data is used to estimate the transport map $\widehat{T}$ and to subsequently compute the importance scores. As shown in Figure D2, the correction term is of smaller order than the approximate term. For this reason, we can approximate the EIF of $\varphi_{X_l}$ with the individual first-order approximate EIFs $\varphi_{jl}^{\mathrm{a}}$. This provides strong empirical evidence that while sample splitting is the most theoretically rigorous approach, practitioners may often be able to use the simpler approximate EIF for inference without splitting the data, which offers a significant computational advantage.

### D.3.2 INFERENCE NEAR THE NULL

As stated in Verdinelli & Wasserman (2024a), when dealing with the quadratic functional, under the null hypothesis, the influence function for the parameter vanishes, the coverage rate becomes $n^{-1}$, and the limiting distribution is no longer a Gaussian distribution. This hinders variable selection with direct statistical guarantees, which is an essential component of reliable scientific discovery (Reyero-Lobo et al. (2026)). In Verdinelli & Wasserman (2024a), they propose to replace the standard error $se_n$ with $\sqrt{se_n^2 + c/n}$ to ensure control of the type-I error, where $c$ is a constant, and many constants could be used. In this paper, we choose $c = \frac{1}{d^2}\min\{\sqrt{\mathbb{V}(Y)}, \mathbb{V}(Y), \mathbb{V}(Y)^2, \mathbb{V}(Y)^4\}$, where $\mathbb{V}(Y)$ is the variance of the output and $d$ is the dimension of all features.

Due to the quadratic nature of the statistic $\widehat{\phi}$, under the null hypothesis, the variance vanishes as $\mathbb{V}(\widehat{\phi}) = O(n^{-\gamma})$ with $\gamma > 1$. It can be found that under the null hypothesis, using Cantelli's inequality, we have $\mathbb{P}_{\mathcal{H}_0}\left(\widehat{\phi} \geq z_\alpha\sqrt{se_n^2 + c/n}\right) \leq \mathbb{V}(\widehat{\phi})/\left(z_\alpha\sqrt{se_n^2 + c/n}\right)^2 \to 0$. This guarantees that the expanded confidence interval controls the Type I error.

### D.4 COMPUTATION TIME COMPARISON

For the computational efficiency study, we followed the experimental protocol of Experiment 2. However, computing Shapley value for all 50 features is prohibitively time-consuming; therefore, for Shapley value (Shapley, 1953), we restrict the feature dimension to $d = 10$ while keeping all other settings identical. Under this setting, we measured the runtime of CPI, LOCO, nLOCO, dLOCO, DFI, FDFI-Z, FDFI, and Shapley value across a range of sample sizes to characterize scalability and comparative efficiency as data volume grows. Each method was run 10 times for every sample size configuration.

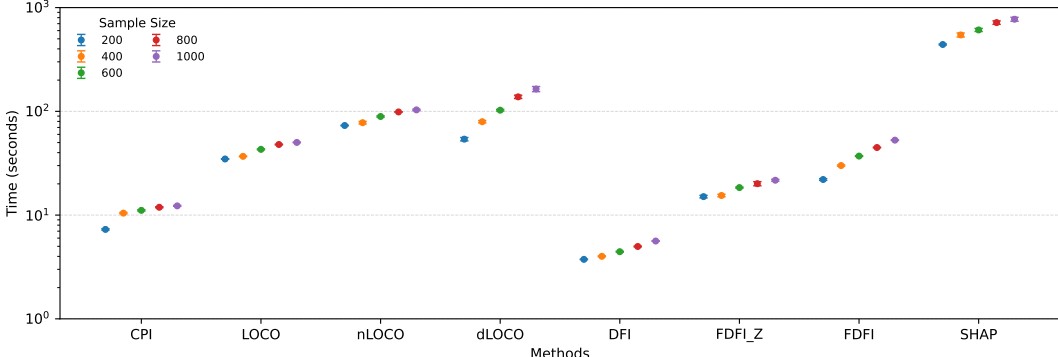

**Figure D3:** Computation time in logarithmic scale of different methods. The error bars indicate standard deviations across 10 random seeds.

The computation-time results show that CPI, FDFI, and DFI are substantially more efficient than LOCO-based and Shapley value methods. Across all sample sizes, both approaches consistently require only a fraction of the runtime compared with LOCO-based and Shapley value estimators, while maintaining stable variance as reflected by the narrow error bars. This highlights their scalability and practical advantage for large-sample applications. In particular, FDFI achieves markedly lower runtime than correlation-aware methods such as dLOCO, further underscoring its efficiency. Although DFI allows for an arbitrary optimal transport map in theory, its implementation in the original paper only uses a Gaussian transport map, which admits a closed-form solution and is indeed efficient, as shown in the above figure. However, if one needs to transform features to another latent distribution, a general OT solver must be used, and it is usually computationally expensive when the feature dimension is large.

# E ADDITIONAL EMPIRICAL RESULTS

## E.1 EXTRA SIMULATION STUDIES

### E.1.1 BENCHMARKING WITH DIFFERENT PREDICTORS

We conduct a comparative evaluation of LOCO, CPI, DFI, FDFI (SCPI), and FDFI (CPI) employing three prediction models: random forests, neural networks, and Lasso in the same setting of Section 4.1, fixing the correlation at $\rho = 0.4$ and the sample size at $n = 1000$. Each configuration is repeated 100 times with different random seeds. We choose $\rho = 0.4$ because the performance of methods such as LOCO and CPI tends to deteriorate as $\rho$ increases; therefore, among $\rho \in \{0.4, 0.6, 0.8\}$, the smallest correlation coefficient was selected to allow a more objective comparison of the performance differences across different predictors.

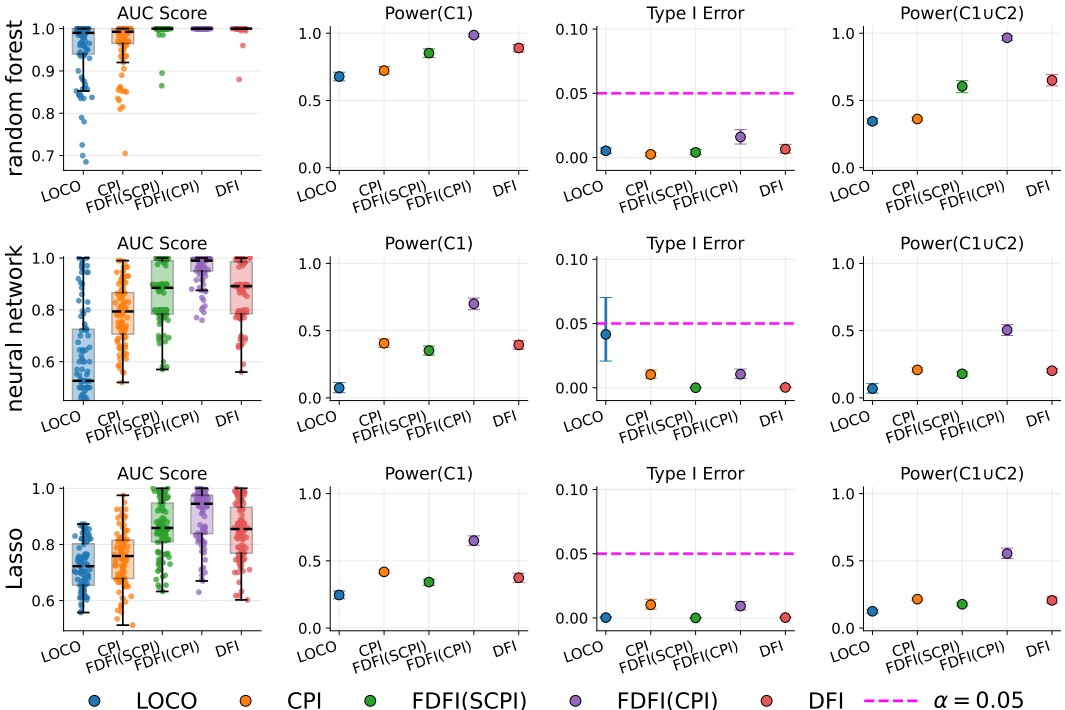

**Figure E4:** Performance comparison across three different predictors under the settings of Section 4.1. Top: results obtained with the random forest predictor. Middle: results obtained with the neural network predictor. Bottom: results obtained with the Lasso predictor. Points indicate mean values, with error bars representing 95% bootstrap confidence intervals over 100 runs.

According to Figure E4, when using the random forest predictor, different methods demonstrate consistently better performance in terms of AUC, Power, and Type I error control. Therefore, we adopt random forests as the predictor in our experiments.

### E.1.2 DISENTANGLED TRANSFORM MAP

The term $(\partial X_l / \partial Z_j)^2$ in Equation (4) quantifies how strongly fluctuations in $Z_j$ are transmitted through $X_l$. Collecting these sensitivities across all pairs $(l, j)$ yields a matrix that characterizes how latent variations propagate into the observed space. This sensitivity matrix reveals the block structure of $X$, where coherent feature groups manifest as localized blocks. To illustrate this phenomenon, we visualize the matrix as a heatmap. Specifically, we consider the nonlinear response model

$$y = \arctan(X_0) + \sin(X_2) + \epsilon,$$

with $X \sim \mathcal{N}_{10}(0, \Sigma)$, and $\epsilon \sim \mathcal{N}(0, 1)$. The covariance matrix $\Sigma$ is block-diagonal with two blocks of equal size ($5 \times 5$). The first block (features $X_0$–$X_4$) is subdivided into two uncorrelated groups,

$X_0, X_1$ and $X_2, X_3, X_4$, while the second block (features $X_5$–$X_9$) forms a single equicorrelated cluster, as in Section 4.1. The within-block correlation is fixed at $\rho = 0.8$.

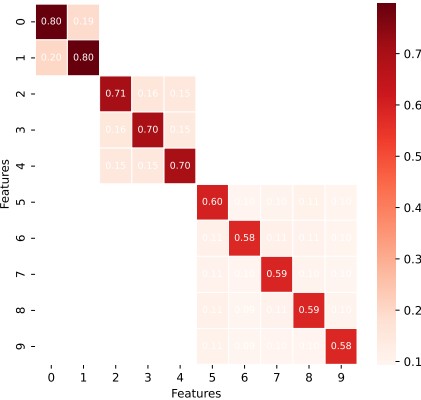

**Figure E5:** Sensitivity matrix under block structure ($\rho = 0.8$). The color bar denotes pairwise correlation strength, with darker red indicating higher correlation.

As shown in Figure E5, the heatmap exhibits clear block patterns consistent with this design. These structures provide direct empirical evidence of grouped dependencies, supporting the claim that the disentangled representation successfully recovers the latent dependency structure of $X$.

### E.1.3 BENCHMARKING ON COMPLEX FEATURE DISTRIBUTIONS

To evaluate in more challenging settings that approximate real-world data, we move from simple Gaussians to Gaussian mixtures, which possess universal approximation capability to capture complex patterns commonly observed in real data (Goodfellow et al., 2016). We retain the same response model from Section 4.1 but generate covariates from a two-component Gaussian mixture: $X \sim 0.2\,\mathcal{N}(0, \Sigma(0.8)) + 0.8\,\mathcal{N}(0, \Sigma(0.2))$. Both covariance matrices, $\Sigma(0.8)$ and $\Sigma(0.2)$, follow the block structure introduced in Section 4.1, with correlation parameters $\rho = 0.8$ and $\rho = 0.2$, respectively. In this non-Gaussian setting, we benchmark 5 feature important measures, including LOCO, nLOCO (normalized LOCO; Verdinelli & Wasserman, 2024b), dLOCO (decorrelated LOCO; Verdinelli & Wasserman, 2024a), DFI, and FDFI, which provide both point estimates and uncertainty quantification. We exclude Shapley-value-based methods from this comparison due to their prohibitive computational cost, as detailed in our runtime analysis (Appendix D.4).

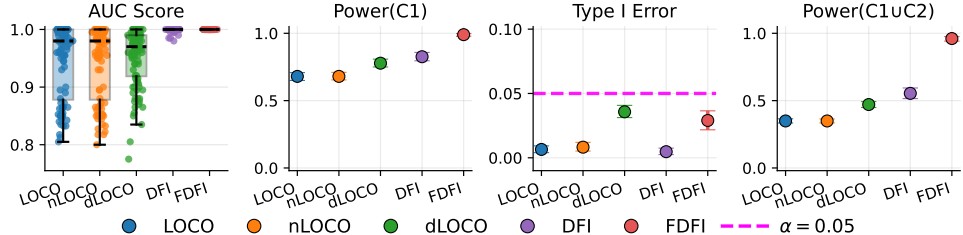

**Figure E6:** Benchmarking results of Gaussian mixture setting with a sample size of $n = 1000$. Points indicate mean values, with error bars representing 95% bootstrap confidence intervals over 100 runs.

The results in Figure E6 demonstrate that FDFI consistently achieves the strongest performance, attaining the highest AUC scores and statistical power while robustly controlling the Type I error. DFI also improves over the LOCO-based methods and maintains well-controlled Type I error. However, DFI's reliance on the Gaussian optimal transport map, which is less flexible than flow matching, limits its discriminative ability on this complex non-Gaussian data, resulting in inferior AUC and power compared to FDFI. By contrast, the LOCO, nLOCO, and dLOCO baselines remain substantially weaker in both AUC and power. Overall, these results highlight that FDFI's flow-based

disentanglement provides superior flexibility and statistical power on complex distributions.

Moreover, we also adapt a more sophisticated design for the covariance matrices and construct $X$ to exhibit heavy-tailed distribution characteristics. We first generate latent Gaussian covariates $Z = (Z_1, \ldots, Z_{1000})^\top \in \mathbb{R}^{1000 \times 50}$ from a two-component Gaussian mixture

$$Z_i \sim 0.2\,\mathcal{N}\big(0, \Sigma(0.8)\big) + 0.8\,\mathcal{N}\big(0, \Sigma(0.2)\big), \qquad i = 1, \ldots, 1000,$$

where $\Sigma(\rho) = I_5 \otimes \Sigma_\rho$ and $(\Sigma_\rho)_{ij} = \rho^{|i-j|/5}$. To induce heavy tails, we then transform $Z$ into $X$ via a Gaussian–$t$ scale mixture: for each $i$, we draw $W_i \sim \chi^2_{10}$ independently and set

$$X_i = \sqrt{\frac{10}{W_i}}\, Z_i,$$

so that the rows of $X$ follow a heavy-tailed distribution of multivariate $t$-type with 10 degrees of freedom and covariance proportional to $\Sigma(\rho)$. We retain the same response model as in Section 4.1 and perform 100 independent experiments, each with a randomly chosen seed. We then compare the performance of LOCO, nLOCO, dLOCO, DFI, and FDFI in terms of AUC score, Type I error, and Power, as reported in Table E2. It can be observed that even under a sophisticated covariance design and heavy-tailed distributions, FDFI still achieves the best performance among all methods by leveraging a flow-based model. This further highlights the advantage of FDFI over DFI in handling complex dependence structures and non-Gaussian data.

**Table E2:** Comparison of LOCO, nLOCO, dLOCO, DFI, and FDFI in terms of AUC score, Type I error, and Power.

| Method | AUC score | Type I error | Power (C1) | Power (C1 ∪ C2) |
|--------|-----------|--------------|------------|------------------|
| LOCO   | 0.9536    | 0.0063       | 0.5180     | 0.2640           |
| nLOCO  | 0.9549    | 0.0095       | 0.5300     | 0.2700           |
| dLOCO  | 0.9893    | 0.0357       | 0.9200     | 0.5080           |
| DFI    | 1.0000    | 0.0037       | 0.9280     | 0.5730           |
| FDFI   | 1.0000    | 0.0307       | 1.0000     | 0.8660           |

### E.1.4 LATENTE INDEPENDENCE DIAGNOSTICS

To inspect the effect of the latent representation $Z$ on FDFI, we compute the pairwise normalized HSIC (nHSIC) with Gaussian kernel:

$$\mathrm{nHSIC}(Z_j, Z_k) = \frac{\mathrm{HSIC}(Z_j, Z_k)}{\sqrt{\mathrm{HSIC}(Z_j, Z_j)\,\mathrm{HSIC}(Z_k, Z_k)}},$$

and the distance correlation (dCor):

$$\mathrm{dCor}^2(Z_j, Z_k) = \frac{\mathrm{dCov}^2(Z_j, Z_k)}{\sqrt{\mathrm{dVar}^2(Z_j)\,\mathrm{dVar}^2(Z_k)}}.$$

between two coordinates $Z_j$ and $Z_k$ for the learned latent representation under different settings. Here, $\mathrm{dCov}$ and $\mathrm{dVar}$ are the covariance and variance of the Euclidean distance matrix.

To illustrate how one can diagnose the latent independence and its effects on statistical inference, we retain the experimental setup of Section 4.1 and keep all other settings fixed. By reducing the width of the neural network and the number of training steps (hidden dimension $128 \to 16$, training steps $5000 \to 5$), , we deliberately deteriorate the training performance so that the correlations among the components of $Z$ increase, and then investigate the resulting impact. For both the "good" and "bad" training regimes, we perform 50 independent runs with different random seeds. In each run, we compute the two metrics between the coordinates of $Z$ and summarize the dependence structure by averaging all off-diagonal entries (i.e., excluding the self-dependence terms with $j = k$) of the resulting pairwise nHSIC and distance-correlation matrices. Finally, we average these run-wise summaries across the 50 runs and relate them to the average empirical power and the standard deviation of the feature importance of the first covariate $X_0$. The results are reported in Table E3. As expected, bad training fails to decorrelate $Z$'s coordinates (large nHSIC and dCor), and hence deteriorates the statistical inference.

**Table E3:** Effect of average dependence in $Z$ (pairwise nHSIC and distance correlation) on test performance and feature importance stability.

| Regime | nHSIC($Z$) | dCor($Z$) | Power($C_1 \cup C_2$) | sd($\widehat{\phi}(X_0)$) |
|---|---|---|---|---|
| Good training | 0.0032 | 0.0543 | 1.0000 | 0.0147 |
| Bad training | 0.0711 | 0.1872 | 0.4480 | 0.0229 |

### E.1.5 SENSITIVITY TO NUISANCE QUALITY

To analyze the sensitivity of FDFI to different flow training, we retain the model construction in Section 4.1 and inspect the performance with different parameter values. According to Figure 2, FDFI becomes highly stable when the sample size is sufficiently large. Therefore, to better investigate how these hyperparameters affect performance, we fix the sample size at 300 in the following experiments.

In Section 4.1, the response is generated as

$$y = \arctan(X_0 + X_1)\, \mathbb{1}_{\{X_2>0\}} + \sin(X_3 X_4)\, \mathbb{1}_{\{X_2<0\}} + \epsilon.$$

Therefore, for $X_0$ and $X_1$, their feature importances should be identical in theory. Motivated by this, we introduce an additional metric to assess the effect of hyperparameters on the training outcome, namely the *geometric-relative discrepancy* between $\phi_0$ and $\phi_1$, defined by

$$\Delta_{\text{geo}}(\phi_0, \phi_1) = \frac{|\phi_0 - \phi_1|}{\sqrt{\phi_0 \phi_1}}.$$

Ideally, this quantity should be as small as possible.

We consider different configurations of each hyperparameter (and keep all other settings identical to Section 4.1): *(i) The size of auxiliary set.* We vary the auxiliary set size $m$, while keeping the training steps proportional to $m$, to ensure sufficient training. *(ii) Latent resample size.* We vary the number of latent resamples $M$ from 1 to 50. *(iii) Model depth/width.* We evaluate 5 configurations of the hidden dimension (width) and the number of layers (depth) of the neural network.

The statistical power, geometric-relative discrepancy, and Type I error in all configurations are presented in Table E4. We observe that increasing the auxiliary set size $m$, the latent resample size $M$, or the network width and depth consistently improves power and reduces geometric discrepancy while maintaining well-controlled Type I error. Notably, as long as the hyperparameters lie within a reasonable range, the overall performance remains highly stable, showing that FDFI maintains stable performance without being overly sensitive to the specific choice of hyperparameters.

**Table E4:** Sensitivity analysis of FDFI to flow training hyperparameters on Power, geometric-relative discrepancy, and Type I error.

| Hyperparameter | | Power($C_1 \cup C_2$) | $\Delta_{\text{geo}}(\phi_0, \phi_1)$ | Type I error |
|---|---|---|---|---|
| Size of auxiliary set $m$ | 300 | 0.8560 | 0.2275 | 0.009 |
| | 500 | 0.8880 | 0.2161 | 0.0180 |
| | 1000 | 0.9080 | 0.1821 | 0.0165 |
| Latent resample size $M$ | 1 | 0.7700 | 0.2522 | 0.0115 |
| | 10 | 0.8820 | 0.2143 | 0.0185 |
| | 50 | 0.9080 | 0.1821 | 0.0165 |
| (width, depth) | (16, 1) | 0.4340 | 0.5637 | 0.0215 |
| | (64, 1) | 0.8860 | 0.2402 | 0.0125 |
| | (128, 1) | 0.8980 | 0.2251 | 0.0205 |
| | (128, 2) | 0.9080 | 0.1821 | 0.0165 |
| | (128, 3) | 0.9040 | 0.2023 | 0.0135 |

### E.2 EXTRA REAL DATA STUDIES

We conducted data analysis on nine different real-world datasets, which vary in sample sizes, feature dimensions, task types, variable types, and domains. We summarize their information in Table E5.

**Table E5:** Summary of all 9 real datasets studied in the paper. In the 'Task' column, 'Cls' denotes classification and 'Reg' denotes regression.

| Dataset | $(n, d)$ | Task | Variable type | Domain |
|---|---|---|---|---|
| Cardiotocography | (2126, 21) | Cls | Continuous | Medical |
| Pima Indians Diabetes | (768, 8) | Cls | Continuous | Medical |
| MicroMass | (571, 1300) | Cls | Discrete | Biological |
| Codon usage | (13028, 69) | Cls | Continuous & Discrete | Biological |
| Default of Credit Card Clients | (30000, 23) | Cls | Continuous & Discrete | Commercial |
| Superconductivity Data | (21263, 81) | Reg | Continuous | Industrial |
| Video Transcoding | (68784, 19) | Reg | Continuous & Discrete | Industrial |
| TCGA-PANCAN-HiSeq | (801, 20531) | Cls | Discrete | Biological |
| human single-cell RNA-seq | (632, 23257) | Cls | Discrete | Biological |

### E.2.1 CARDIOTOCOGRAPHY DATASET

The Cardiotocography (CTG) dataset ($n = 2126, d = 21$) (Campos & Bernardes, 2000) consists of 2,126 computer-processed cardiotocograms with 21 diagnostic features derived from fetal heart rate (FHR) and uterine contraction (UC) signals. These features capture baseline FHR, counts of accelerations and decelerations, short- and long-term variability, and FHR-histogram statistics. We use this dataset to classify fetuses as normal or abnormal and to compare feature importance estimates produced by different methods. The Shapley value method was not applied here because its computation time exceeded one day, even with only 21 features.

As shown in Figure E7, LOCO and CPI identify only a small subset of features as significant, in sharp contrast to FDFI and DFI. This divergence stems from the block-diagonal structure of the feature correlation matrix (Section 4.3), which is a manifestation of correlation distortion under multicollinearity. In such settings, LOCO and CPI attenuate or mask true effects, whereas FDFI and DFI are less affected (Verdinelli & Wasserman, 2024b). In the CTG dataset, *LB* anchors the tracing around the expected 110–160 bpm baseline for reassuring status, while central-tendency descriptors (*Mean*, *Median*, *Mode*) provide concordant evidence of the baseline neighborhood (Ayres-de Campos et al., 2015; Jia et al., 2023). Reactivity is reflected by *AC* (accelerations), often accompanied by *FM* (fetal movements), both indicating intact autonomic control and low short-term risk. Excessive *UC* (uterine contractions), i.e., tachysystole, can compromise uteroplacental perfusion and precipitate decelerations (Ayres-de Campos et al., 2015). Beat-to-beat and longer-scale variability are captured by *MSTV/ASTV* (short-term variability mean and abnormal-time proportion) and *MLTV/ALTV* (long-term variability mean and abnormal-time proportion). Adequate variability (higher *MSTV*, *MLTV* and lower *ASTV*, *ALTV*) is reassuring, whereas depressed variability is non-reassuring (Ayres-de Campos et al., 2015; Jia et al., 2023; Stampalija et al., 2023). Deceleration phenotypes—*DL* (mild/early), *DS* (severe/variable), and *DP* (prolonged)—span a spectrum from benign positional patterns to forms associated with cord compression and hypoxia/acidemia, with *DP* ($> 2$ min) and repetitive deep *DS* most strongly linked to metabolic risk (Parer & Ikeda, 2007). Histogram-based morphology provides supportive, context-dependent information: overall spread via *Width* and *Variance* (very low values echo variability loss; excessively wide or erratic distributions indicate nonreassurance), extremes via *Min/Max* (sustained brady- or tachycardic periods), modal structure via *Nmax/Nzeros* (multimodality and sparsity cues), and slow drift via *Tendency*. These descriptors are standard in computer-aided CTG (e.g., SisPorto) and are highlighted in contemporary reviews (Ayres-de Campos et al., 2000; Romano et al., 2016; Zhao et al., 2018).

FDFI consistently isolates the physiologically meaningful signal and preserves it under strong correlation and Bonferroni control, yielding a stable, guideline-concordant importance profile. This pattern demonstrates clear advantages in interpretability and robustness, retaining clinically relevant effects despite multicollinearity and multiple testing, thereby providing a more reliable attribution map than methods that are prone to correlation distortion.

**Case-wise attributions:** Recall that the importance of feature $X_l$ is estimated by Algorithm D.1 as:

$$\Psi_{il} = \sum_{j=1}^{d} \Omega_{ij} \cdot \widehat{H}_{jl}(\widehat{Z}_i), \qquad \widehat{\phi}_{X_l} = \frac{1}{n} \sum_{i=1}^{n} \Psi_{il}.$$

**Table E6:** Description of 21 features in the Cardiotocography dataset $(n = 2126, d = 21)$. Where $n$ is the sample size and $d$ is the feature dimension. 'FHR' means 'Fetal Heart Rate'.

| Broad category | Feature name | Meaning |
|---|---|---|
| Baseline & counts | LB | FHR baseline (beats per minute) |
| | AC | Number of accelerations per second |
| | FM | Number of fetal movements per second |
| | UC | Number of uterine contractions per second |
| Decelerations | DL | Number of light decelerations per second |
| | DS | Number of severe decelerations per second |
| | DP | Number of prolonged decelerations per second |
| Short-term variability | ASTV | Percentage of time with abnormal short-term variability |
| | MSTV | Mean value of short-term variability |
| Long-term variability | ALTV | Percentage of time with abnormal long-term variability |
| | MLTV | Mean value of long-term variability |
| Histogram (FHR) | Width | Width of the FHR histogram |
| | Min | Minimum value of the FHR histogram |
| | Max | Maximum value of the FHR histogram |
| | Nmax | Number of histogram peaks |
| | Nzeros | Number of histogram zeros |
| | Mode | Mode of the histogram |
| | Mean | Mean of the histogram |
| | Median | Median of the histogram |
| | Variance | Variance of the histogram |
| | Tendency | Tendency of the histogram |

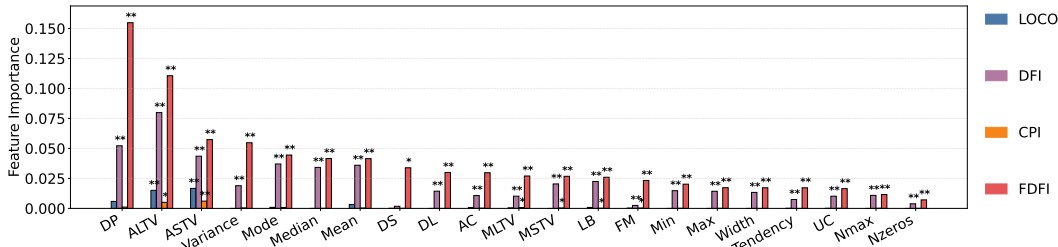

**Figure E7:** Bar plot of feature importances of the random forest classifier on the Cardiotocography (CTG) dataset. The symbols **\*** and **\*\*** on the bars denote statistical significance at a level of $\alpha = 0.05$ and $\alpha = 0.05/21$, respectively. The meaning of all features is provided in Table E6.

In Figure 3(a), the heatmap exhibits a strong block-diagonal structure among *LB*, *Mean*, *Mode*, and *Median*. For feature *LB*, we define the block-restricted contribution for *LB* as

$$\Psi_{i,\mathrm{LB}}^{\mathrm{block}} = \sum_{j \in \mathcal{B}_{\mathrm{LB}}} \Omega_{ij} \, \widehat{H}_{j,\mathrm{LB}}(\widehat{Z}_i), \qquad \mathcal{B}_{\mathrm{LB}} := \{\mathrm{LB}, \mathrm{Mean}, \mathrm{Mode}, \mathrm{Median}\}.$$

Averaging these restricted sample-wise contributions over all $i$ yields a block-based feature importance $\widehat{\phi}_{X_{\mathrm{LB}}}^{\mathrm{block}} = \frac{1}{n} \sum_i \Psi_{i,\mathrm{LB}}^{\mathrm{block}}$. For comparison, $\widehat{\phi}_{X_{\mathrm{LB}}} = \frac{1}{n} \sum_i \Psi_{i,\mathrm{LB}}$ denotes the overall feature importance of *LB* computed using all features. In our real data analysis, the overall importance $\widehat{\phi}_{X_{\mathrm{LB}}} = 0.0286$, the block-based importance $\widehat{\phi}_{X_{\mathrm{LB}}}^{\mathrm{block}} = 0.0228$, and their ratio is $0.7972$, meaning that the four-feature block accounts for nearly $80\%$ of the importance of *LB*.

### E.2.2 PIAM DIABETES DATASET

The Pima Indians Diabetes dataset $(n = 768, d = 8)$ (Smith et al., 1988) contains medical records from female patients of Pima Indian heritage. Each record includes eight clinical attributes such as the number of pregnancies, plasma glucose concentration, blood pressure, skinfold thickness, insulin level, body mass index (BMI), diabetes pedigree function, and age. The task is to predict the onset of Type 2 diabetes, providing a widely used benchmark for classification in clinical settings.

**Table E7:** Description of 8 features in the Pima Indians Diabetes dataset $(n = 768, d = 8)$. Where $n$ is the sample size and $d$ is the feature dimension.

| Broad category | Feature name | Meaning |
|---|---|---|
| Obstetric history | Pregnancies | Number of times pregnant |
| Glycemia | Glucose | Plasma glucose concentration at 2 hours in an OGTT (mg/dL) |
| Blood pressure | BloodPressure | Diastolic blood pressure (mm Hg) |
| Adiposity | SkinThickness
BMI | Triceps skinfold thickness (mm)
Body mass index (kg/m$^2$) |
| Insulinemia | Insulin | 2-hour serum insulin (mU/L) |
| Family history | DPF | Diabetes pedigree function (family history–based risk score) |
| Demographics | Age | Age (years) |

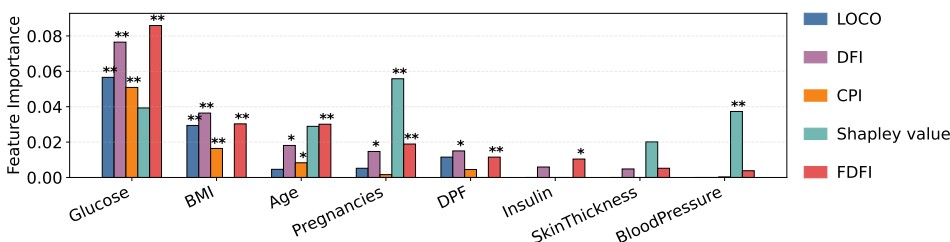

**Figure E8:** Bar plot of feature importances of the random forest classifier on the Piam diabetes dataset. The symbols * and ** on the bars denote statistical significance at a level of $\alpha = 0.05$ and $\alpha = 0.05/8$, respectively. The meaning of all features is provided in Table E7.

On the Piam diabetes dataset, the results in Figure E8 align with clinical knowledge while underscoring FDFI's robustness under correlation and multiple testing (Bonferroni $\alpha = 0.00625$). (i) Diagnostic marker: *Glucose* is consistently identified as significant by all methods except Shapley value and remains so after Bonferroni adjustment, reflecting its established diagnostic role in diabetes (ElSayed et al., 2023). (ii) Risk factors: *BMI* shows a nearly identical pattern, being retained after correction for all methods except Shapley value, consistent with evidence that adiposity strongly elevates diabetes risk, though its impact diminishes once baseline glycemia is considered (Jayedi et al., 2022). FDFI additionally detects *Age*, *Pregnancies*, and *DPF* after correction and assigns non-negligible weight to *BloodPressure*, aligning with their interpretation as background screening factors rather than diagnostic markers (Davidson et al., 2021; Valdez et al., 2007; Emdin et al., 2015). (iii) Physiology-aligned signals: Beyond these established predictors, FDFI also highlights *Insulin* at the nominal level and assigns non-negligible weight to *SkinThickness*, consistent with prior evidence linking parity, OGTT-based insulin, and skinfold adiposity with type 2 diabetes risk (Guo et al., 2017; Nicholson et al., 2006; Hanley et al., 2003; Ruiz-Alejos et al., 2020).

### E.2.3 MicroMass dataset

The MicroMass dataset $(n = 571, d = 1300)$ (Mah & Veyrieras, 2014) contains mass spectrometry measurements from 571 bacterial samples. Each sample is encoded as a 1300-dimensional spectrum of mass-to-charge intensity values, which serve as high-dimensional fingerprints of bacterial composition. While originally designed for species-level identification, the dataset also provides a natural binary partition between Gram-positive and Gram-negative bacteria. In our work, we leverage this property to frame the problem as a binary classification task, aiming to discriminate Gram-positive from Gram-negative organisms. This setting provides a biologically meaningful benchmark for evaluating the effectiveness of the proposed method in high-dimensional, structurally complex data.

For the MicroMass dataset, due to the large number of features, we report the counts of features identified as significant by the four methods (LOCO, CPI, DFI, and FDFI) under two significance thresholds: the nominal 0.05 level and the Bonferroni-adjusted 0.05/1300 level, as summarized in Figure E9(a). Results show that DFI and FDFI identify a considerably larger number of significant

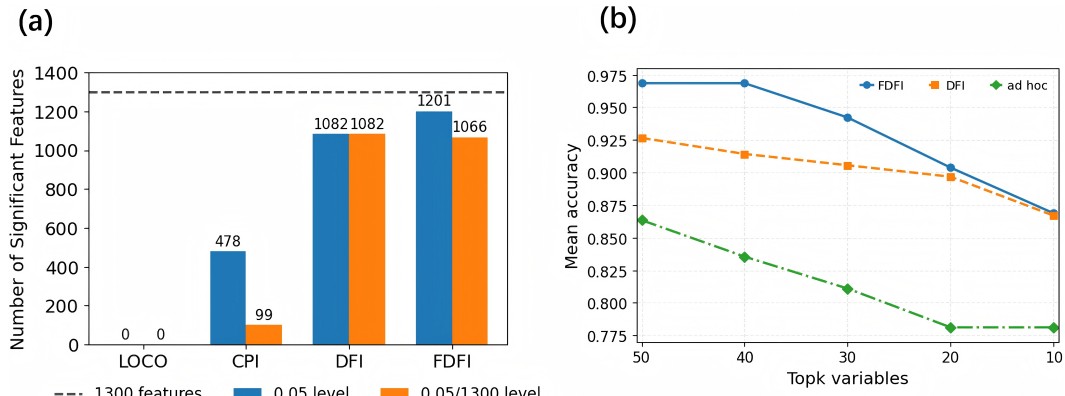

**Figure E9:** Data analysis of the MicroMass dataset. (a) Number of significant features identified by LOCO, CPI, DFI, and FDFI under different significance levels on the MicroMass dataset. (b) Prediction accuracy with selected important features for FDFI, DFI, and an ad hoc method that applies CPI on the cluster-representative features.

features, whereas CPI yields only a few significant findings and LOCO fails to detect any, reflecting their susceptibility to correlation distortion under multicollinearity.

Consistent with the preceding analysis in Appendix E.2, we also compare FDFI, DFI, and an ad-hoc procedure that first clusters features on the Spearman correlation matrix, selects one medoid per cluster as a representative, and then applies CPI on this reduced feature set. As illustrated in Figure E9(b), on the MicroMass dataset, FDFI exhibits substantial superiority over both DFI and the ad hoc method, empirically confirming the effectiveness of the proposed approach under high-dimensional, structured-feature settings. Notably, by operating directly on the full feature space, FDFI avoids the information loss inherent to cluster selection and remains robust to correlation distortion, yielding consistently higher accuracy across the range of $k$.

### E.2.4 Results on Four Additional Large-Scale Datasets

In this section, we conduct experiments on four real-world datasets of moderately large sample sizes. These datasets cover both classification and regression tasks. The larger sample sizes allow for more reliable results, making it possible to assess the robustness and generalizability of the models across different task types. We compare average prediction performance across datasets for features selected by FDFI, DFI, and an ad hoc CPI-based clustering approach, using prediction accuracy for classification and root mean squared error (RMSE) for regression, with results averaged over two-fold splits for each dataset. For the classification task, we conduct experiments on datasets Codon usage ($n = 13028, d = 69$) and Default of Credit Card Clients (DCCC) ($n = 30000, d = 23$), and for the regression task, we conduct experiments on datasets Superconductivity Data ($n = 21263, d = 81$) and Video Transcoding ($n = 68784, d = 19$). The results reported in Table E8 clearly demonstrate the superiority of our method. FDFI consistently outperforms both DFI and the ad hoc CPI-based clustering approach across these datasets for both classification and regression tasks. This superiority is evident in its ability to more effectively select the top-k features, which leads to better model performance.

### E.2.5 Extra Results on Two RNA-seq Datasets

To further demonstrate that our method can achieve superior performance over existing approaches in genuinely high-dimensional regimes with intricate inter-feature correlations, we additionally conduct experiments on two challenging RNA-seq datasets. (i) The TCGA-PANCAN-HiSeq bulk RNA-seq dataset ($n = 801, d = 20531$) (Weinstein et al., 2013), where the goal is to classify samples into five tumor types: breast invasive carcinoma (BRCA), kidney renal clear cell carcinoma (KIRC), colon adenocarcinoma (COAD), lung adenocarcinoma (LUAD), and prostate adenocarcinoma (PRAD). This dataset is representative of large-scale transcriptomic studies with strong co-expression structures and severe $p \gg n$ imbalance. (ii) The human single-cell RNA-seq dataset ($n = 632, d = 23257$) (Darmanis et al., 2017), which is used to distinguish neoplastic cells originating from the tumor core versus those from the periphery. This dataset exemplifies ultra–high-

**Table E8:** Classification performance (accuracy) of three methods with different top-$k$ feature sets on 4 datasets with large sample sizes.

| Type | Dataset | Codon usage | | | Default of Credit Card Clients | | |
|---|---|---|---|---|---|---|---|
| | Method | Top-8 | Top-4 | Top-2 | Top-8 | Top-4 | Top-2 |
| Classification | FDFI | 0.7642 | 0.6448 | 0.5401 | 0.8182 | 0.8197 | 0.8197 |
| | DFI | 0.7588 | 0.6391 | 0.5142 | 0.8150 | 0.8137 | 0.8173 |
| | ad-hoc | 0.7132 | 0.5839 | 0.4479 | 0.8173 | 0.8118 | 0.8173 |
| Type | Dataset | Superconductivty Data | | | Video Transcoding | | |
| | Method | Top-8 | Top-4 | Top-2 | Top-8 | Top-4 | Top-2 |
| Regression | FDFI | 11.0347 | 12.7988 | 15.0577 | 6.8195 | 7.9727 | 9.0607 |
| | DFI | 11.4675 | 13.4683 | 16.9657 | 6.8588 | 8.2657 | 9.3723 |
| | ad-hoc | 12.0083 | 13.1241 | 17.8346 | 6.8991 | 8.2052 | 9.1645 |

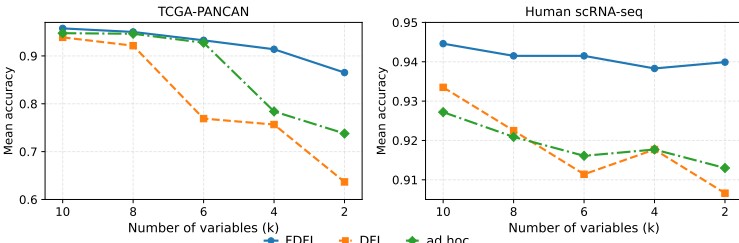

**Figure E10:** Prediction accuracy vs. top-k selected variables across two RNA-seq datasets.

dimensional single-cell measurements with heterogeneous cell populations and complex gene–gene dependencies, providing a stringent testbed for evaluating feature-importance methods. Following the preprocessing procedures in (Yan et al., 2025), we selected 1,500 highly variable genes (HVGs) for the TCGA-PANCAN-HiSeq bulk RNA-seq dataset and 2,000 HVGs for the human single-cell RNA-seq dataset, and performed all downstream analyses on these HVG subsets. We then compared the average prediction accuracy across datasets for the important features selected by FDFI, DFI, and an ad hoc CPI-based clustering approach, using two-fold splits and reporting the mean accuracy for each dataset.

In particular, for the human single-cell RNA-seq dataset, we found that the Top-20 genes selected by our method include *ALDOC*, *HES6*, and *CPE*, all of which have been previously implicated in glioma biology. *ALDOC*, a glycolytic enzyme highly enriched in neural tissue, has been implicated in glioma metabolic reprogramming, where altered *ALDOC* expression is associated with enhanced glycolytic activity, tumor progression, and increased migratory potential of glioma cells (Chang et al., 2024). *HES6*, a basic helix–loop–helix transcription factor, is selectively overexpressed in glioma and functions as an important regulator driving tumor-cell proliferation, migration, and lineage plasticity (Haapa-Paananen et al., 2012). *CPE* which encodes a neuroendocrine peptide-processing enzyme, has more recently been recognized as an oncogenic factor in high-grade gliomas, where its elevated expression promotes tumor-cell survival, invasion, and stress adaptation through enhanced metabolic resilience and extracellular-matrix remodeling (Hareendran et al., 2022).

These concordances between known glioma biology and the features automatically selected by our method provide additional support for the clinical relevance of our approach. In contrast, for the TCGA-PANCAN-HiSeq dataset, only gene indices (rather than gene symbols) are provided, so we report prediction accuracy only and do not attempt to interpret the selected genes in terms of their clinical relevance.

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
