# OpenReview forum: "Flow-Disentangled Feature Importance"
_ICLR.cc/2026/Conference — ICLR 2026 Poster_

### Official Review · Reviewer_akD3 · 2025-10-28

**Soundness:** 3
**Presentation:** 3
**Contribution:** 3
**Rating:** 8
**Confidence:** 3

**Summary:**

This paper introduces Flow-Disentangled Feature Importance (FDFI), a model-agnostic interpretability framework designed to provide statistically valid feature importance estimation even in the presence of strong feature correlations. The key idea is to use flow matching—a flexible generative modeling technique—to learn a disentanglement map that transforms correlated features into an approximately independent latent space. Once features are decorrelated in this latent space, feature importance is computed and then projected back to the original space, along with uncertainty quantification based on semiparametric efficiency theory.
FDFI generalizes earlier Disentangled Feature Importance (DFI) work, extending it from the restrictive $\ell_2\text{-loss}$ setting to arbitrary differentiable loss functions, thereby supporting both regression and classification tasks. The paper provides theoretical results proving the asymptotic normality and semiparametric efficiency of the proposed estimators, along with empirical results on synthetic and biomedical datasets (e.g., Cardiotocography, Diabetes, and MicroMass). Empirically, FDFI achieves higher AUC, power, and more robust inference than competing model-agnostic methods such as LOCO, CPI, and DFI, especially under multicollinearity.

**Strengths:**

1. Sound theoretical framing: The paper provides a clear unification of LOCO, CPI, and SCPI under general differentiable loss functions, and introduces a principled FDFI definition that subsumes DFI as the ℓ₂ special case. The semiparametric efficiency and asymptotic normality proofs are technically rigorous and well-documented.
2. Flexible, scalable transport: The flow-matching approach yields expressive and stable mappings with a well-defined training objective.
3. Compelling empirical results: FDFI consistently achieves higher AUC and statistical power while maintaining Type-I error control across correlated and non-Gaussian settings (Figures 2–3). The CTG, Pima, and MicroMass studies further confirm interpretability and clinical relevance, including recovery of known physiological patterns and robustness in high-dimensional regimes (d ≈ 1300).

**Weaknesses:**

1. Because the disentanglement map $T$ is learned via flow matching rather than optimal transport, different flow paths could, in principle, lead to distinct ‘independent’ latent representations $Z$. While Appendix B establishes uniqueness of the ODE solution for a chosen interpolation path, semantic identifiability of the latent axes remains uncertain. This limits how literally one can interpret the Jacobian heatmaps as explanatory structures. Incorporating quantitative independence diagnostics (e.g., distance correlation or energy tests on $Z$) would strengthen interpretability claims.
2. Although the asymptotic theory accommodates $n^{-\frac{1}{4}}$ convergence rates, finite-sample behavior may be sensitive to inaccuracies in the estimated flow $T$. The paper could be strengthened by an ablation over auxiliary-sample size $m$ and architectural choices (depth, width) to examine effects on MMD independence, feature scores, and variance estimates. Appendix D.2 provides training details, but robustness to nuisance quality remains empirically under-explored.

**Questions:**

1. Latent independence diagnostics: Can you report quantitative independence scores (e.g., pairwise HSIC or distance correlation) for the learned $Z$ in the main experiments, and relate them to attribution stability and power?
2. Sensitivity to flow training: In Appendix D.2 you select hyperparameters with MMD; could you add (a) an ablation on auxiliary set size $m$, (b) $M$ (latent resamples) vs. variance/power, and (c) model depth/width vs. AUC/power?
3. Interpretability of Jacobians: The CTG heatmap (Fig. 4a) is compelling. Can you provide case‑wise attributions (e.g., conditional on ranges of clinical variables) to illustrate how Jacobian structure plus latent scores produce the final importance?
4.When FDFI may fail: Are there data‑generating processes (e.g., strong higher‑order interactions that can’t be linearized locally) where the squared‑Jacobian weighting is systematically biased, even if the latent scores are well estimated?

---

> ### Author Response · Authors · 2025-11-20
> **Response to Reviewer akD3**
>
> Thank you for recognizing the unified framing, stable transport, and strong power/Type-I performance across correlated and non-Gaussian settings. Please find our response to your comments below:
>
> **[W1, Q1] (Independence)** Thank you for the great suggestion! To inspect the effect of the latent representation $Z$ on FDFI, we compute the pairwise normalized HSIC (nHSIC) with Gaussian kernels and the distance correlation (dCor) (see Appendix E.1.4 for precise definitions) between two coordinates $Z_j$ and $Z_k$ for the learned latent representation under different settings.
>
> To illustrate how one can diagnose the latent independence and its effects on statistical inference, we retain the experimental setup of Section 4.1 and keep all other settings fixed. By reducing the width of the neural network and the number of training steps (hidden dimension $128 \to 16$, training steps $5000 \to 5$), we deliberately deteriorate the training performance so that the correlations among the components of $Z$ increase, and then investigate the resulting impact. For both the ''good'' and ''bad'' training regimes, we perform 50 independent runs with different random seeds. In each run, we compute the two metrics between the coordinates of $Z$ and summarize the dependence structure by averaging all off-diagonal entries (i.e., excluding the self-dependence terms with $j = k$) of the resulting pairwise nHSIC and distance-correlation matrices. Finally, we average these run-wise summaries across the 50 runs and relate them to the average empirical power and the standard deviation of the feature importance of the first covariate $X_0$. The results are reported in the table below. As expected, bad training fails to decorrelate $Z$'s coordinates (large nHSIC and dCor), and hence deteriorates the statistical inference.
>
> | Regime        | nHSIC(Z) | dCor(Z) | Power (C1 ∪ C2) | sd(φ̂(X₀)) |
> |---------------|----------|---------|------------------|------------|
> | Good training | 0.0032 | 0.0543  | 1.0000 | 0.0147 |
> | Bad training  | 0.0711 | 0.1872  | 0.4480 | 0.0229 |
>
> In addition, we conducted more sensitivity analyses and reported these metrics per your suggestions (see **[W2,Q2]** below).
>
> **[W2,Q2] (Sensitivity to nuisance quality)** To analyze the sensitivity of FDFI to different flow training, we retain the model construction in Section 4.1 and inspect the performance with different parameter values according to your suggestions.
>
> Following the same data model in Section 4.1, where the feature importances for $X_0$ and $X_1$ are identical, we introduce an additional metric to assess the effect of hyperparameters on the training outcome, namely the *geometric-relative discrepancy* between $\phi_0$ and $\phi_1$:
> $$\Delta_{\mathrm{geo}}(\phi_0,\phi_1)= \frac{\lvert \phi_0 - \phi_1\rvert}{\sqrt{\phi_0 \phi_1}}.$$
> Ideally, this quantity should be as small as possible.
>
> We consider different configurations of each hyperparameter (and keep all other settings identical to Section~4.1):
> - The size of auxiliary set. We vary the auxiliary set size $m$, while keeping the training steps proportional to $m$, to ensure sufficient training.
> - Latent resample size. We vary the number of latent resamples $M$ from 1 to 50.
> - Model depth/width. We evaluate 5 configurations of the hidden dimension (width) and the number of layers (depth) of the neural network.
>
> The statistical power, geometric-relative discrepancy, and Type~I error in all configurations are presented in the table below. We observe that increasing the auxiliary set size $m$, the latent resample size $M$, or the network width and depth consistently improves power and reduces geometric discrepancy while maintaining well-controlled Type I error. Notably, as long as the hyperparameters lie within a reasonable range, the overall performance remains highly stable, showing that FDFI maintains stable performance without being overly sensitive to the specific choice of hyperparameters.
>
> | Hyperparameter              | Value      | Power (C1 ∪ C2) | Δ_geo(φ₀, φ₁) | Type I error |
> |----------------------------|------------|------------------|---------------|--------------|
> | Size of auxiliary set m    | 300 | 0.8560 | 0.2275  | 0.0090       |
> | Size of auxiliary set m    | 500  | 0.8880 | 0.2161        | 0.0180       |
> | Size of auxiliary set m    | 1000   | 0.9080 | 0.1821        | 0.0165       |
> | Latent resample size M | 1 | 0.7700  | 0.2522        | 0.0115       |
> | Latent resample size M  | 10         | 0.8820  | 0.2143        | 0.0185       |
> | Latent resample size M   | 50 | 0.9080 | 0.1821        | 0.0165       |
> | (width, depth) | (16, 1)    | 0.4340 | 0.5637        | 0.0215       |
> | (width, depth) | (64, 1)    | 0.8860      | 0.2402 | 0.0125 |
> | (width, depth) | (128, 1)   | 0.8980   | 0.2251  | 0.0205       |
> | (width, depth) | (128, 2)   | 0.9080   | 0.1821  | 0.0165       |
> | (width, depth) | (128, 3)   | 0.9040           | 0.2023        | 0.0135       |

---

> ### Author Response · Authors · 2025-11-20
> **Response to Reviewer akD3 (continued)**
>
> **[Q3] (Case-wise attributions)** Recall that for each observation $i$ and feature $X_l$, Algorithm D.1 computes a sample-wise contribution
> $$\Psi_{il}=\sum_{j=1}^{d} \Omega_{ij} \hat{H}\_{jl}(\hat{Z}\_i),\qquad\hat{\phi}\_{X_l} = \frac{1}{n} \sum_{i=1}^{n} \Psi_{il},$$
> where $\Omega_{ij}$ captures the task-relevant effect along latent direction $j$ for case $i$ (via loss contrasts in Algorithm D.1), and $\hat{H}\_{jl}(\hat{Z}\_i) = [\nabla \hat{T}^{-1}(\hat{Z}\_i)]\_{jl}^2$ encodes how strongly feature $X_l$ locally loads on latent coordinate $j$. Thus, the global importance $\hat{\phi}\_{X_l}$ is obtained by aggregating case-wise contributions that combine latent scores $\Omega_{ij}$ with the Jacobian structure $\hat{H}_{jl}$.
>
> In Figure 4(a), the heatmap of $(\hat{H}\_{jl})$ exhibits a strong block-diagonal structure among *LB*, *Mean*, *Mode*, and *Median*. To quantify how this block drives the importance of *LB*, we define the block-restricted contribution
> $$\Psi^{\text{block}}\_{i,\mathrm{LB}}:=\sum_{j \in \mathcal{B}\_{\mathrm{LB}}} \Omega_{ij}\,\hat{H}\_{j,\mathrm{LB}}(\hat{Z}\_i),\qquad\mathcal{B}\_{\mathrm{LB}} := \{\mathrm{LB},\mathrm{Mean},\mathrm{Mode},\mathrm{Median}\},$$
> and the corresponding block-based importance
> $$\hat{\phi}^{\text{block}}\_{X_{\mathrm{LB}}}\;=\;\frac{1}{n} \sum_{i=1}^n \Psi^{\text{block}}\_{i,\mathrm{LB}}.$$
> For comparison, $\hat{\phi}\_{X_{\mathrm{LB}}} = \frac{1}{n} \sum_{i=1}^n \Psi_{i,\mathrm{LB}}$ denotes the overall importance of *LB* using all latent directions. In our CTG analysis, the overall importance is $\hat{\phi}\_{X\_{\mathrm{LB}}} = 0.0286$, whereas the block-based importance is $\hat{\phi}^{\text{block}}\_{X\_{\mathrm{LB}}} = 0.0228$, so that the four-variable block accounts for approximately $80\%$ of the importance of *LB*. In the revised manuscript, we report such conditional attributions for clinically meaningful strata of the CTG variables in Appendix E.2.1, illustrating how the Jacobian structure (via $\hat{H}\_{jl}$) and the latent scores (via $\Omega_{ij}$) combine to yield stable, interpretable case-wise importance patterns.
>
> **[Q4] (When FDFI may fail)** We appreciate the reviewer’s question about potential failure modes of FDFI in the presence of strong higher-order interactions.
>
> Conceptually, FDFI relies on two key ingredients: (i) a representation in which the prediction function $f \circ \hat{T}^{-1}$ is locally well-approximated by a first-order expansion in the latent coordinates, and (ii) the squared-Jacobian weights $\hat{H}_{jl}(Z)$, which map importance in the latent space back to the original features by attributing local variation in $Z_j$ to $X_l$. This implicitly assumes a form of \emph{local linearizability} of the prediction function along the flow and a localized, smoothly varying sensitivity of $X$ to $Z$.
>
> There are indeed data-generating processes where these assumptions can be violated. In particular, if the response surface exhibits extremely strong, non-local higher-order interactions that cannot be captured by local expansions around the observed data (for example, functions that are nearly flat on most of the support but change only on very thin manifolds, or highly oscillatory, parity/XOR-type rules), then small perturbations of $Z$ along the flow may fail to probe the truly influential directions for $f$. In such regimes, even if the latent scores are well estimated, the squared-Jacobian weighting can become systematically biased because it only reflects local sensitivity, while the global risk is driven by rare or combinatorial interactions far from the observed neighborhoods.
>
> Two points mitigate this issue in practice and clarify the scope of our contribution:
> 1. *Scope of our guarantees.* Our theoretical results explicitly assume regularity and smoothness conditions on both the flow and the prediction function (e.g., differentiability and bounded derivatives). Within this regime, the squared-Jacobian reweighting is not ad hoc, but is the transformation induced by the flow map that preserves the semantics of latent perturbations at the feature level.
> 2. *Empirical regime and future directions.* Empirically, FDFI yields stable rankings across resamples and aligns with domain knowledge (e.g., in the LC–MS microbiome data), suggesting that the local-sensitivity assumption is reasonable in these settings. At the same time, extending FDFI to handle highly non-smooth or purely combinatorial interaction structures (e.g., via higher-order or non-local perturbations in the latent space) is an important and interesting direction for future work.
>
> We have added a brief discussion of these potential failure modes and assumptions to the discussion section, clarifying that FDFI is designed for settings where the learned representation and predictor are locally smooth, and highlighting non-smooth, strongly combinatorial regimes as a promising area for further methodological development.

---

> > ### Comment · Reviewer_akD3 · 2025-11-26
> >
> > I want to thank the authors for the reply and the efforts they have made to accomplish it. I will maintain the score.

---

### Official Review · Reviewer_8E6r · 2025-10-30

**Soundness:** 3
**Presentation:** 3
**Contribution:** 3
**Rating:** 4
**Confidence:** 3

**Summary:**

The paper introduces Flow-Disentangled Feature Importance (FDFI), a new model-agnostic framework for estimating feature importance and its statistical uncertainty in ML models. Unlike standard model-agnostic attribution methods (e.g., permutation or removal-based approaches) that become unreliable when features are correlated (correlation distortion), FDFI provides statistically valid and interpretable attributions even under severe multicollinearity.


Previous decorrelation-based methods (e.g., Disentangled Feature Importance, DFI) were limited to l2 loss, but FDFI generalizes this to any differentiable loss function, allowing its use in both regression and classification settings. FDFI employs flow matching, a recent generative modeling technique (Lipman et al.), to learn a disentanglement map that transforms correlated features X into statistically independent latent variables Z. This approach replaces the restrictive Gaussian otpimal transport assumption used in prior work. It enables flexible modeling of arbitrary feature distributions while preserving interpretability of the transformation.

The authors derive a semiparametric inference theory showing that FDFI estimators are asymptotically efficient and allow valid confidence intervals and hypothesis testing with controlled Type I error. The FDFI estimator remains sqrt(n)-consistent even when the flow map T is estimated nonparametrically at slower rates. Furthermore, FDFI quantifies how each feature’s contribution to model loss propagates through the correlation structure of the data. It provides both latent-level and original-feature-level importance scores, linking disentangled representations back to raw variables.

Evaluations on synthetic and biomedical datasets compare FDFI with LOCO (Leave-One-Covariate-Out), CPI (Conditional Permutation Importance), and DFI. FDFI achieves higher statistical power in detecting relevant features, controlled type I error, and robust performance under high feature correlation. FDFI also recovers correlated yet predictive features that other methods fail to identify.

**Strengths:**

The main contribution of the paper is well-motivated, clearly situating its novel approach within an interesting problem setting. The authors provide a compelling analysis of existing methods, outlining its strengths and weaknesses, clearly identifying specific gaps and limitations that effectively frame the need for their proposed framework. This strong introduction is complemented by a solid theoretical foundation. The authors build their first theoretical results on reasonable assumption (A1) to establish strong initial guarantees, such as the tight bounds presented in Theorem 2.1, ensuring the soundness of their method.

The primary theoretical contribution is Theorem 3.1, establishing the semiparametric efficiency of the latent FDFI estimator, $\hat{\phi}^{\text{FDFI}}_{Z_j}$. This is a solid result. It proves that the estimator achieves the optimal $\sqrt{n}$-convergence rate. A significant practical implication of this efficiency is that this $\sqrt{n}$-consistency holds even when using a flexible, nonparametrically-estimated nuisance map, $\hat{T}$, which itself may converge at a slower rate.

This property is important indeed as it allows for the use of complex machine learning models to estimate the nuisance components without sacrificing the statistical validity of the final importance score. Finally, the authors complete their framework by providing the necessary inferential tools in Proposition 3.2. This proposition grounds the theory in practice, allowing for the construction of valid confidence intervals and hypothesis tests for the FDFI scores, making the entire method statistically sound and practically applicable.

Before proceeding to the weaknesses I would like to point out that, although the theoretical results seem sound, I have not checked their validity (especially regarding the proofs).

**Weaknesses:**

The paper extends the DFI framework and identifies two of its key limitations: its reliance on the optimal transport map for disentanglement, which can be computationally expensive and less flexible for complex, high-dimensional distributions. However, this claim seems somewhat not well tackled by FDFI. From the appendix, it becomes evident that the proposed FDFI methods are actually slower in terms of computation time, and the experiments are applied only to relatively simple settings—Gaussian and mixture-of-Gaussian data, along with a real-world dataset of max 21 dimensions. When reading this section, I was hoping that FDFI would be faster than DFI or at laset that the authors would provide stronger evidence demonstrating its advantages on complex, high-dimensional data. Neither of these expectations is convincingly addressed.

Figure 1 focuses primarily on highlighting the benefits of FDFI compared to LOCO, CPI, and SHAP, yet it would be more informative if it also emphasized improvements over DFI. This addition would help clarify and disentangle which of the observed gains are due to the new FDFI framework itself and which are inherited from DFI.

In line 233, the authors claim that DFI is limited because it cannot be applied to classification tasks or models using general differentiable loss functions. However, in Assumption 1 they assume that the loss is M-smooth, which already implies differentiability. It is unclear whether this assumption applies only to the first two theoretical results or to the broader framework. If the former case is the true one, it would be useful for the authors to clarify this connection and more explicitly relate Section 2.1 to the FDFI formulation.

In line 260, the authors mention that Equation (4) recovers the DFI formulation under the l2 loss. It would strengthen the paper if they explicitly demonstrated this equivalence or at least referred readers to an appendix section where the derivation is shown.

Another issue is that Assumptions A2–A4, which seem to be key to the theoretical results in Section 3, are introduced in Theorem 3.1 without any prior mention/discussion or intuitive explanation. Unless I have missed this (in which case I would appreciate if the authors can point me to this), the absence of even a brief description of what these assumptions entail and how restrictive they are leaves the reader with little understanding/comprehension of the conditions under which the results hold. The same issue appears in Proposition 3.2, which refers to Assumption A4(iv) without elaboration. A more detailed or intuitive analysis/explanations of these assumptions in the main text would greatly improve the clarity of the manuscript.

When reviewing Algorithm D.1 in the appendix, it appears quite complex and, when compared to DFI, even slower in execution. Given its importance, a concise yet clear overview of how the inference procedure operates under FDFI should be included in the main text rather than entirely postponed to the appendix. This would make the paper more accessible and self-contained.

Regarding the experiments, the authors conduct three sets—on Gaussian data, mixtures of Gaussians, and a real-world dataset. While these are informative, they do not convincingly support the paper’s claims about handling complex, high-dimensional data distributions. The authors should either perform experiments on higher-dimensional datasets or tone down their claims about performance in such settings. Alternatively, they could provide a justification explaining why their chosen real-world dataset adequately reflects the complexity they aim to address.

In addition, the time analysis presented in Figure D.3 raises further questions. Out of curiosity, could the authors explain the sudden increase in DFI’s runtime between sample sizes 400 and 600? Also, the slope of FDFI’s runtime growth with respect to sample size appears steeper than that of DFI.. These observations suggest scalability issues, and it would be appropriate for the authors to acknowledge this limitation explicitly in the main text or conduct further experiments.

Overall, while the paper makes a strong theoretical contribution and proposes a well-founded framework, the implementation appears surprisingly complex and may benefit from simplification or at least justification of its current form. Some crucial content that supports understanding—such as explanations of key assumptions and the inference algorithm—should be moved from the appendix into the main text. On the other hand, less important material, like parts of the Gaussian mixture experiments (which I do not believe add much), could be moved to the appendix. Doing so would improve  the paper’s structure, its readability and transparency.

**Questions:**

Please see weaknesses section.

---

> ### Author Response · Authors · 2025-11-20
> **Response to Reviewer 8E6r (1)**
>
> We thank the reviewer for the careful reading and for underscoring the importance of semiparametric efficiency with nonparametric nuisance estimation. For ease of navigation, we index the nine paragraphs in weaknesses as W1–W9 and address them thematically, responding point-by-point with specific clarifications, added derivations, and new experiments.
>
> **[W1,W8] (Computational complexity)**
>
> - *Conceptual motivation beyond runtime.* Our primary goal is to address structural and statistical limitations of both DFI and classical LOCO/CPI-type measures in the presence of strong feature dependence, rather than to merely outperform the most optimized Gaussian DFI implementation in wall-clock time. FDFI provides (i) a flexible flow-matching latent representation that can capture complex, non-Gaussian feature distributions; (ii) a unified importance functional that recovers LOCO- and CPI-type quantities in the latent space while yielding scale-comparable scores across features; and (iii) a principled influence-function-based uncertainty quantification scheme. These properties are not available in standard LOCO, CPI, or DFI, and they remain valid regardless of the particular latent distribution used.
>
> - *Runtime.* The apparent gap in DFI’s runtime between $n = 400$ and $n = 600$ in the original Figure D.3 is caused by CPU underclocking due to resource contention when the experiments were run in parallel. This produced an artificial slowdown for DFI at $n = 600$ that is not representative of its algorithmic complexity. To provide a more reliable assessment, we reran the computational time comparison experiment sequentially on an otherwise idle machine and updated Figure~D.3 accordingly; the resulting curves are smoother, and the overall conclusions remain unchanged.\
> In terms of scaling with $n$, the updated figure shows that both DFI (with its closed-form Gaussian transport) and FDFI exhibit approximately linear runtime growth, with FDFI having a slightly steeper slope due to the cost of fitting a flexible flow model. This is expected: FDFI trades a modest constant-factor overhead for substantially greater modeling flexibility. Importantly, FDFI remains comparable to Gaussian DFI in runtime and is substantially more efficient than other baseline attribution methods, such as nLOCO, dLOCO, and Shapley value, which require repeated model retraining or exponentially many perturbations.
>
>
> **[W1,W7] (Experiment breadth)** Regarding the concern that our experiments focus on relatively simple distributions, we clarify that our original real-data analysis already includes the MicroMass dataset $(n = 571, d = 1300)$ (Appendix E.2), while the CTG dataset $(d = 21)$ is shown in the main text because its features are easier to interpret. To further demonstrate that our method can indeed achieve superior performance over other methods in high-dimensional settings with complex inter-feature correlations, we also perform experiments on two additional datasets. (i) The TCGA-PANCAN-HiSeq bulk RNA-seq dataset ($n = 801$, $d = 20531$) where the task is to classify samples into five tumor types: breast invasive carcinoma (BRCA), kidney renal clear cell carcinoma (KIRC), colon adenocarcinoma (COAD), lung adenocarcinoma (LUAD), and prostate adenocarcinoma (PRAD). (ii) The human single-cell RNA-seq dataset ($n = 632$, $d = 23257$) , which is employed to distinguish neoplastic cells from the tumor core versus neoplastic cells from the periphery. We compared the average prediction accuracy across datasets for the important features selected by FDFI, DFI, and an ad hoc approach that applies CPI to cluster-representative features, using two-fold splits and reporting the mean accuracy for each dataset. The results are summarized below:
> | Top-k | TCGA FDFI | TCGA DFI | TCGA ad-hoc | Human FDFI | Human DFI | Human ad-hoc |
> |-------|-----------|----------|-------------|------------|-----------|--------------|
> | 10    | 0.9575    | 0.9388   | 0.9475      | 0.9446     | 0.9335    | 0.9272       |
> | 8     | 0.9501    | 0.9214   | 0.9463      | 0.9415     | 0.9225    | 0.9209       |
> | 6     | 0.9326    | 0.7690   | 0.9276      |0.9415    | 0.9114     | 0.9161       |
> | 4     | 0.9139    | 0.7566   | 0.7840      | 0.9383     | 0.9177    | 0.9177       |
> | 2     | 0.8652    | 0.6367   | 0.7378      | 0.9399     | 0.9066   | 0.9130       |
>
> FDFI consistently outperforms both DFI and the ad hoc method, highlighting its superiority in identifying more predictive and biologically representative gene sets; see Appendix E.2.5 for specific biomarkers identified by FDFI and their clinical relevance.

---

> ### Author Response · Authors · 2025-11-20
> **Response to Reviewer 8E6r (continued)**
>
> **[W2] (Figure illustration)** Thank you for the helpful suggestion. We agree that clearly distinguishing the improvements introduced by FDFI from those inherited from DFI is important for interpretation. In our experience, we did not find a clean and informative way to visually encode these distinctions (in the transport maps and losses) directly in Fig. 1 (e.g., via additional panels or heavy annotation) without overloading what is meant to be a high-level overview figure. Instead, in the revision, we refined the caption of Fig. 1 to (i) explicitly describe which parts of the pipeline are already present in DFI and (ii) state that DFI is recovered as a special case of FDFI when the transport map is linear and the loss is squared error.
>
> **[W3,W5] (Technical assumptions)** We thank the reviewer for pointing out the potential confusion regarding the role of smoothness and differentiability assumptions.
> - *Scope of Assumption A1.* Assumption A1 ($M$-smoothness of $\ell$ in its second argument) is used only in Section 2.1, specifically for the equivalence results in Theorem 2.1 and Lemma 2.2. There, a uniform second-order control is convenient to bound Taylor remainders and relate the original DFI-type perturbation functional to our FDFI formulation. Assumption A1 is not needed for the definition of the FDFI estimand in general, nor for any of the efficiency and inference results in Sections 2.2–3. We have revised the text after Assumption A1 to explicitly clarify the above.\
> Regarding the remark in line 233 about DFI being limited to squared-error loss, our claim was intended to refer to the original DFI functional and associated theory, which are formulated for regression with squared loss and do not cover general supervised learning losses. In contrast, our FDFI formulation and subsequent inference results apply to a broad class of differentiable losses. As already noted before Assumption A1, the binary logistic loss $\ell(y,s) = \log(1 + \exp(-ys))$ with $y \in \{\pm 1\}$ has a $1/4$-Lipschitz gradient in $s$, so certain standard classification surrogates do satisfy A1; we now explicitly state that the $M$-smoothness requirement is only used for the equivalence arguments in Section~2.1.
>
> - *Assumptions for EIF and inference.* The efficiency theory and inference results rely instead on Assumption A4(i), which only requires that $s \mapsto \ell(y,s)$ be differentiable, together with the standard empirical-process/complexity condition in Assumption A4(ii) (or its cross-fitting relaxations). For differentiable but non–$M$-smooth surrogates such as the exponential/AdaBoost loss $\ell(y,s) = \exp(-ys)$ or the probit negative log-likelihood $\ell(y,s) = -\log \Phi(ys)$ with $y \in \{\pm 1\}$, Assumption A4(ii) follows from standard Donsker-closure arguments: under bounded parameters (hence bounded scores), the induced loss classes are locally Lipschitz transforms of finite-dimensional parametric or VC-subgraph predictor classes and are therefore P-Donsker. These settings are thus covered by our inference framework without imposing $M$-smoothness.
>
> - *Intuition for Assumptions A2–A4.* In the revision, we have improved the presentation of assumptions. In particular, we have
>   - added a short paragraph before Theorem 3.1 that informally summarizes A2–A4 and explains their roles: (i) A2 collects basic regularity and integrability conditions ensuring pathwise differentiability of the FDFI functional; (ii) A3 encodes high-level rate conditions on the nuisance estimators that guarantee that their estimation error is second-order; and (iii) A4 provides the complexity conditions needed to control the stochastic fluctuations of the estimated influence function.
>     - included an explanation before Proposition 3.2 of Assumption A4(iv), clarifying that it is a mild strengthening of the complexity requirement that ensures uniform convergence of $\hat{T}$ and can be verified under standard Donsker conditions.
>   - We also direct readers at these locations to the appendix, where the assumptions are stated in full and discussed in more technical detail.
>
> **[W4] (Reduction of FDFI to DFI under  $\ell_2$ loss)** We have added derivations in Appendix A.3, to show how FDFI coincides with DFI under $\ell_2$ loss.
>
> **[W6] (Algorithm overview)** We now include a concise main-text summary (Algorithm 1) and refer readers to Appendix D.1 (Algorithm D.1) for full pseudocode and implementation details.
>
> **[W9] (Organization)** We sincerely thank you for the structural guidance. In the revision, we: (i) move concise explanations of the key assumptions and their roles from the appendix to the main text, (ii) add a brief overview of the inference procedure as Algorithm 1 in the main text while keeping full pseudocode in Appendix D.1, and (iii) replace nonessential Gaussian-mixture simulation with large-scale real data examples in Section 4.2. We hope this reorganization improves readability and transparency.

---

> > ### Comment · Reviewer_8E6r · 2025-11-25
> >
> > I thank the authors for their detailed responses. I was happy to see the additional two experiments, clarifications on the assumptions, as well as changes that the authors made in the manuscript. Therefore I have raised my score to 6.

---

### Official Review · Reviewer_rfpb · 2025-11-01

**Soundness:** 3
**Presentation:** 3
**Contribution:** 3
**Rating:** 6
**Confidence:** 2

**Summary:**

The paper propose a new method to reliably quantify feature importance using flow-disentangled. The method follows Disentangled Feature Importance (DFI) and resolve two critical problems of the previously proposed principled framework for attribution under dependence: 1) relying on optimal transport (OT) and 2) restriction to important score with L2 loss. The paper develop statistical inference theory and show empirical results.

**Strengths:**

- The paper has a strong methodological innovation
  - The previous method Disentangled Feature Importance (DFI) is limited by the optimal transport (OT)  and L-2 loss with restricted applications.
- The evaluation is comprehensive. The experiment settings include both synthetic data and real data settings. The method outperformed other baselines significantly.
- The paper contains solid theory analysis.

**Weaknesses:**

- The experiment setting is over-simplified with some simple correlation and the real experiment setting is too small with less than 3000 data points.

**Questions:**

Is there any downstream task that could be used or improved by the novel feature attribution method?

---

> ### Author Response · Authors · 2025-11-20
> **Response to Reviewer rfpb**
>
> Thank you for highlighting the methodological novelty, comprehensive evaluation, and theoretical grounding.
> Please find our response to your concerns about experiments and questions about the downstream task.
>
>
> **[W1] (Experiment settings)** Thank you for your comments on the experiment settings!
> To improve our numerical evaluation, we have added new simulation experiments with more complex correlation structures in Appendix E.1.3 and new real data analyses with larger sample sizes in Appendix E.2.4.
>
> Regarding ``Oversimplified correlations'', we adopt a more sophisticated design for the covariance matrices and construct $X$ to exhibit heavy-tailed distribution characteristics; see Appendix E.1.3 for more details. We then compare the performance of LOCO, nLOCO, dLOCO, DFI, and FDFI in terms of AUC score, Type I error, and Power, as reported in the table below.
> We observe that FDFI consistently achieves the best performance among all methods.
>
>
> | Method | AUC score | Type I error | Power (C1) | Power (C1 ∪ C2) |
> |--------|-----------|--------------|------------|-----------------|
> | LOCO   | 0.9536    | 0.0063       | 0.5180     | 0.2640          |
> | nLOCO  | 0.9549    | 0.0095       | 0.5300     | 0.2700          |
> | dLOCO  | 0.9893    | 0.0357       | 0.9200     | 0.5080          |
> | DFI    | 1.0000    | 0.0037       | 0.9280     | 0.5730          |
> | FDFI   | 1.0000    | 0.0307       | 1.0000     | 0.8660          |
>
>
> As for the ``Small real data'', we conduct experiments on four real datasets of moderately large scale, covering both classification and regression tasks, with two datasets for each type of task. We then compare average prediction performance across datasets for features selected by FDFI, DFI, and an ad hoc CPI-based clustering approach, using prediction accuracy for classification and RMSE for regression, with results averaged over two-fold splits for each dataset. For the classification task, we conduct experiments on datasets Codon usage ($n=13028, d= 69$) and Default of Credit Card Clients  ($n=30000, d=23$), and for the regression task, we conduct experiments on datasets Superconductivity Data ($n=21263, d=81$) and Video Transcoding ($n=68784, d=19$).
> FDFI outperforms DFI and ac-hoc method, as shown in the tables below
>
> | Method | Codon usage Top-8 | Codon usage Top-4 | Codon usage Top-2 | Default of Credit Card Clients Top-8 | Default of Credit Card Clients Top-4 | Default of Credit Card Clients Top-2 |
> |--------|-------------------|-------------------|-------------------|--------------------------------------|--------------------------------------|--------------------------------------|
> | FDFI   | 0.7642            | 0.6448            | 0.5401            | 0.8182                               | 0.8197                               | 0.8197                               |
> | DFI    | 0.7588            | 0.6391            | 0.5142            | 0.8150                               | 0.8137                               | 0.8173                               |
> | ad-hoc | 0.7132            | 0.5839            | 0.4479            | 0.8173                               | 0.8118                               | 0.8173                               |
>
>
> | Method | Superconductivity Data Top-8 | Superconductivity Data Top-4 | Superconductivity Data Top-2 | Video Transcoding Top-8 | Video Transcoding Top-4 | Video Transcoding Top-2 |
> |--------|------------------------------|------------------------------|------------------------------|-------------------------|-------------------------|-------------------------|
> | FDFI   | 11.0347                      | 12.7988                      | 15.0577                      | 6.8195                  | 7.9727                  | 9.0607                  |
> | DFI    | 11.4675                      | 13.4683                      | 16.9657                      | 6.8588                  | 8.2657                  | 9.3723                  |
> | ad-hoc | 12.0083                      | 13.1241                      | 17.8346                      | 6.8991                  | 8.2052                  | 9.1645                  |

---

> ### Author Response · Authors · 2025-11-20
> **Response to Reviewer rfpb (continued)**
>
> **[Q1] (Downstream tasks)**
> We thank the reviewer for this important question.
>
> Our primary motivation in developing FDFI is to mitigate the *correlation distortion* of classical Shapley- and SHAP-based attributions under dependent covariates. Consequently, any downstream task that currently relies on SHAP values can, in principle, benefit from replacing the underlying attribution with FDFI. This includes the broad range of use cases, such as global feature ranking and selection in tabular risk or clinical models, local explanations of individual predictions, monitoring and debugging of complex models, fairness and risk analyses, and domain-specific workflows for text (e.g., sentiment analysis, QA), images (e.g., image classification), and genomics. In these settings, FDFI offers advantages in terms of calibrated type-I error (uncertainty quantification for the estimated importance, which SHAP does not offer) under feature correlation, improved power to detect truly predictive variables, and more stable global rankings, while avoiding the combinatorial coalition enumeration that often limits the scalability of exact or approximate Shapley methods.
>
>
> Concretely, one downstream task we already demonstrated in Fig.4 (and MicroMass, Fig.E9) is *feature pruning*, where retaining the top-$k$ features ranked by FDFI preserves or improves predictive accuracy relative to ad-hoc clustering while removing low-impact variables. This importance-guided pruning is directly useful for scientific discovery; for instance, in biomarker identification, where compact yet predictive molecular signatures must be extracted from high-dimensional biological data. By eliminating redundant dimensions while preserving the FDFI-ranked signal, feature pruning naturally enables *model compression*, yielding smaller, faster, and more interpretable models with minimal loss of accuracy under inference-time or latency constraints.
>
> FDFI is also useful for *data collection and experimental design*, where the goal is to prioritize variables or sensors whose marginal utility is highest, thereby improving sample efficiency and reducing acquisition cost. For example, in regulatory genomics, feature importance has been used to guide experimental design and data acquisition by identifying which molecular layers most strongly influence gene expression, thus indicating which assays or conditions should be prioritized to maximize biological insight. Replacing correlation-distorted Shapley scores with FDFI in such pipelines yields more reliable prioritization when the underlying molecular measurements are strongly correlated.
>
> Finally, FDFI improves *model debugging and spurious-correlation detection* by flagging features whose apparent signal is driven primarily by correlation rather than genuine predictive content, addressing known pitfalls of permutation and Shapley/SHAP importance under dependence. For example, Fig.3 of Chen et al 2022 showed that in the NHANES mortality model, observational Shapley values assigned non-zero importance to gender simply because it was correlated with age under the chosen baseline distribution. By contrast, the proposed FDFI uses latent independent factors to disentangle such effects, providing more stable and interpretable importance scores and thereby reducing false positives in downstream scientific or decision-making workflows.
>
> In short, our method is designed as a drop-in replacement for Shapley-based attributions in downstream tasks such as feature pruning, model compression, experimental design, and model debugging, particularly in correlated, high-stakes, or resource-constrained regimes where accurate and statistically valid attributions are critical. Motivated by this question, we have added a short paragraph in the Discussion section of the revised manuscript to summarize these downstream benefits and the connection to Shapley-based use cases.
>
> **References**
>
> Chen, Hugh, Scott M. Lundberg, and Su-In Lee. "Explaining a series of models by propagating Shapley values." Nature communications 13.1 (2022): 4512.

---

> > ### Comment · Reviewer_rfpb · 2025-11-26
> >
> > I thank the authors for the reply and I maintain my score.

---

### Author Response · Authors · 2025-11-20

We thank all reviewers for your careful reading and constructive feedback.
In the revised manuscript, we have implemented a number of substantial changes to clarify the theoretical framework, strengthen the empirical evaluation, and improve exposition.

- **Theoretical clarification and positioning.** We clarified the role of the smoothness and complexity assumptions, emphasizing that M-smoothness (Assumption A1) is used only for the equivalence results in Section 2.1, while the semiparametric efficiency and inference theory rely on differentiability and standard empirical-process conditions (Assumptions A2–A4). We added intuitive summaries of these assumptions in the main text, expanded the discussion before Theorem 3.1 and Proposition 3.2, and provided an explicit derivation in Appendix A.3 showing how FDFI reduces to DFI under $\ell_2$ loss.

- **Algorithmic transparency and organization.** We introduced a concise Algorithm 1 in the main text that summarizes the full FDFI estimation and inference pipeline, while keeping full pseudocode in Appendix D.1. We also reorganized the presentation by moving high-level explanations of key assumptions and inference steps from the appendix into the main text and streamlining the experimental section to separate core results from auxiliary details.

- **Expanded empirical evaluation.** Beyond the original benchmarks, we added new simulation studies with more complex correlation structures and heavy-tailed covariates, as well as additional real-data experiments on 2 high-dimensional RNA-seq datasets (TCGA-PANCAN-HiSeq and Human Single-Cell RNA-Seq) and 4 large-scale datasets, demonstrating that FDFI remains effective in high-dimensional, strongly correlated regimes. For the cardiotocography case study, we now also report block-wise and case-wise attributions, illustrating how the Jacobian weights and latent scores combine to yield stable and clinically interpretable patterns.\
Below is a summary of all datasets used in the current manuscript:\
| Dataset                         | (n, d)        | Task | Variable type            | Domain      |
|---------------------------------|---------------|------|--------------------------|------------|
| Cardiotocography                | (2126, 21)    | Cls  | Continuous               | Medical    |
| Pima Indians Diabetes           | (768, 8)      | Cls  | Continuous               | Medical    |
| MicroMass                       | (571, 1300)   | Cls  | Discrete                 | Biological |
| Codon usage                     | (13028, 69)   | Cls  | Continuous & Discrete    | Biological |
| Default of Credit Card Clients  | (30000, 23)   | Cls  | Continuous & Discrete    | Commercial |
| Superconductivity Data          | (21263, 81)   | Reg  | Continuous               | Industrial |
| Video Transcoding               | (68784, 19)   | Reg  | Continuous & Discrete    | Industrial |
| TCGA-PANCAN-HiSeq              | (801, 20531)  | Cls  | Discrete                 | Biological |
| human single-cell RNA-seq       | (632, 23257)  | Cls  | Discrete                 | Biological |

- **Complexity, scope, and downstream use cases.** We added a detailed discussion of FDFI's computational complexity and its relationship to DFI, highlighting how the additional overhead is tied to greater flexibility (general differentiable losses and non-Gaussian covariates) and how the implementation exploits vectorization and parallelism. In the Discussion section, we expanded the description of downstream applications (feature pruning, model compression, experimental design, and model debugging) and positioned FDFI as a drop-in replacement for Shapley-based attributions in correlated settings. We also added an explicit discussion of potential failure modes in highly non-smooth or parity/XOR-type regimes and clarified that these non-local, combinatorial structures are an important direction for future extensions of flow-disentangled feature importance.

---

### Author Response · Authors · 2025-11-27

We sincerely thank the reviewers and the area chair for their time and constructive feedback. We appreciate the thoughtful comments and have addressed all points carefully in our responses and revisions. Thank you again for your valuable contributions to strengthening our work.

---

### Meta-Review · Area_Chair_Qmjv · 2025-12-13

**Summary:**

All of the reviewers replied to the authors after reading the authors' rebuttal, indicating that they're satisfied with the authors' responses. For this reason, I recommend accepting the paper.

**Reviewer Concerns:**

* Theoretical clarification and positioning: The authors clarified these points.

* Algorithmic transparency and organization: The authors addressed these points by providing a concise Algorithm 1 and reorganizing the presentation.

* Expanded empirical evaluation: The authors added new simulation studies with more complex correlation structures and heavy-tailed covariates.

* Complexity, scope, and downstream use cases: The authors addressed these points clearly.

**Reviewer Scores:**

* **Reviewer rfpb**: indicated that they maintain the score *6* after reading the rebuttal.

* **Reviewer 8E6r**: indicated that they raised the score to *6* after reading the rebuttal.

* **Reviewer akD3**: indicated that they main the score *8* after reading the rebuttal.

---

### Decision · Program_Chairs · 2026-01-26

Accept (Poster)